# Meltwater storage in low-density near-surface bare ice in the Greenland Ice Sheet ablation zone

Matthew G. Cooper[1], Laurence C. Smith[1], Asa K. Rennermalm[2], Clément Miège[3], Lincoln H Pitcher[1], Jonathan C. Ryan[1,4], Kang Yang[5,6], Sarah Cooley[1,7]

[1]Department of Geography, University of California-Los Angeles, CA 90095, USA
[2]Department of Geography, Rutgers, The State University of New Jersey, Piscataway, NJ 08854, USA
[3]Department of Geography, University of Utah, Salt Lake City, Utah, USA
[4]Institute at Brown for Environment and Society, Brown University, Providence, Rhode Island, USA
[5]School of Geographical and Oceanographic Sciences, Nanjing University, Nanjing 210023, China
[6]Joint Center for Global Change Studies, Beijing 100875, China
[7]Dept. of Earth, Environmental and Planetary Sciences, Brown University, Providence, RI, 02912

*Correspondence to*: Matthew G. Cooper (guycooper@ucla.edu)

**Abstract.** We document the density and hydrologic properties of bare, ablating ice in a mid-elevation (1215 m a.s.l.)
supraglacial internally drained catchment in the Kangerlussuaq sector of the western Greenland Ice Sheet. We find low-density (0.43–0.91 g cm$^{-3}$, µ=0.69 g cm$^{-3}$) ice to at least 1.1 m depth below the ice sheet surface. This near-surface, low-density ice consists of alternating layers of water-saturated, porous ice and clear solid ice lenses, overlain by a thin (<0.5 m), even lower density (0.33–0.56 g cm$^{-3}$, µ=0.45 g cm$^{-3}$) unsaturated weathering crust.  Ice density data from 10 shallow (0.9–1.1 m) ice cores along an 800 m transect suggest an average 14–18 cm of specific meltwater
storage within this low-density ice. Water saturation of this ice is confirmed through measurable water levels (1–29 cm above hole bottoms, µ=10 cm) in 84% of cryoconite holes and rapid refilling of 83% of 1m drilled holes sampled along the transect. These findings are consistent with descriptions of shallow, depth-limited aquifers on the weathered surface of glaciers worldwide and confirm the potential for substantial transient meltwater storage within porous low-density ice on the Greenland Ice Sheet ablation zone surface. A conservative estimate for the ∼63 km$^2$
supraglacial catchment yields 0.009–0.012 km$^3$ of liquid meltwater storage in near-surface, porous ice. Further work is required to determine if these findings are representative of broader areas of the Greenland Ice Sheet ablation zone, and to assess the implications for sub-seasonal mass balance processes, surface lowering observations from airborne and satellite altimetry, and supraglacial runoff processes.

## 1 Introduction

Each summer a vast hydrologic network of lakes and rivers forms on the surface of the western Greenland Ice Sheet ablation zone in response to surface melting (Chu, 2014; Smith et al., 2015). Evidence suggests that most or all of this

water is efficiently delivered via supraglacial rivers to moulins, crevasses, and, ultimately, to proglacial rivers and surrounding oceans (van As et al., 2017; Colgan et al., 2011; Lindbäck et al., 2015; Rennermalm et al., 2013; Smith et al., 2015). The assumption of efficient meltwater delivery is reflected in regional climate and surface mass balance models for Greenland that instantaneously credit ablation zone surface runoff to the ocean with no physical representation of hydrologic processes taking place on the bare ice surface (Smith et al., 2017). Field studies and satellite remote sensing, however, have found evidence of substantial meltwater runoff delays on daily to monthly timescales in the Greenland Ice Sheet ablation zone (van As et al., 2017; Karlstrom and Yang, 2016; Koenig et al., 2015; Lindbäck et al., 2015; Overeem et al., 2015; Rennermalm et al., 2013; Smith et al., 2017). Similar runoff delays are observed on valley glaciers elsewhere (Karlstrom et al., 2014; Munro, 1990), inferred to relate to the presence of a degraded, porous "weathering crust" (Müller and Keeler, 1969) on the bare ice surface of glaciers and ice sheets that stores meltwater, delaying its delivery to supraglacial channels via porous sub-surface flow (Irvine-Fynn et al., 2011; Karlstrom et al., 2014; Munro, 2011). The porous weathering crust may also provide a locus for internal and/or surficial refreezing of meltwater (Hoffman et al., 2014; Paterson, 1972; Willis et al., 2002). Together, hydrologic processes in the weathering crust are similar to those of meltwater transport, storage, and refreezing in snow and firn (Cox et al., 2015; Forster et al., 2014; Harper et al., 2012; Machguth et al., 2016). The presence of weathering crust in Greenland, however, has gone largely undocumented, and little is known about its effect on hydrologic processes in the bare ice ablation zone, where >85% of ice sheet surface meltwater runoff is generated (Machguth et al., 2016).

Weathering crusts are fractured, disintegrated, or "rotten" ice layers that form during the melt season on the thermally transient surface of ablating glaciers (Fountain and Walder, 1998; Irvine-Fynn et al., 2011; Müller and Keeler, 1969). In temperate ice, liquid water exists within an interconnected network of meltwater veins (Lliboutry, 1996; Mader, 1992; Nye and Frank, 1973). When glacier ice is exposed to water, these veins coarsen to the order of tenths of a millimetre in diameter, a process referred to as "rotting" (Nye, 1991). On bare ice surfaces exposed to solar radiation, this action is intensified by the transmission and absorption of solar radiation through the upper few meters of ice (Cook et al., 2016; Fountain and Walder, 1998; Irvine-Fynn and Edwards, 2014). Sub-surface radiative heating enhances melting along ice grain edges, further coarsening vein networks and disaggregating ice crystals, creating a layer of porous ice typically <2 m thick (Fig. 1) (Irvine-Fynn et al., 2011; Müller and Keeler, 1969).

Weathering crust formation reflects a balance between the vertical depth of sub-surface melting and the rate of ice surface lowering. This balance evolves in response to spatio-temporal changes in the surface energy balance during the melt season (Müller and Keeler, 1969). During clear-sky conditions, solar radiative heating promotes development and deepening of the weathering crust. The depth of sub-surface melting is typically limited to <2 m by the exponential attenuation of radiative heating with depth (Brandt and Warren, 1993). Conversely, during

exceptionally warm, windy, or cloudy conditions when surface melt rates are enhanced relative to sub-surface melting, the weathering crust may decay or be rapidly removed (Müller and Keeler, 1969). As the weathering crust develops, a shallow, depth-limited aquifer may establish in the near-surface porous ice (Fig. 1) (Irvine-Fynn and Edwards, 2014). At vertical depths where meltwater drains through the permeable weathering crust to seeps and

supraglacial channels, the near-surface ice density is reduced with no detectable change in glacier surface height (Hoffman et al., 2014; Müller and Keeler, 1969). Consequently, weathering crust ice density exhibits a characteristic non-linear increase from a very low-density (<0.5 g cm$^{-3}$) surface layer to a higher density (~0.90 g cm$^{-3}$) impermeable substrate (Cook et al., 2016; LaChapelle, 1959) (Fig. 1). As such, mass change during periods of weathering crust development or removal cannot be resolved solely from ice surface elevation changes, but also

requires knowledge of the sub-surface depth-density profile (Braithwaite et al., 1998; LaChapelle, 1959; Müller and Keeler, 1969; Munro, 1990).

Weathering crust is often enhanced by cryoconite, the biologically active dark sediment that preferentially absorbs solar radiation, locally enhancing melt in quasi-cylindrical holes that deepen into the weathering crust and brighten

the ice surface relative to dispersed or uniform debris-covered ice (Boggild et al., 2010). Cryoconite holes are coupled to the weathering crust via porous sub-surface water exchange (Cook et al., 2016) and surface flow that redistributes sediments, nutrients, and microbial cells between holes and the ice surface, potentially controlling their distribution and ecological structure (Edwards et al., 2011; Hodson et al., 2007; Hotaling et al., 2017; Takeuchi et al., 2000). Weathering crust hydrology, therefore, exerts a dynamic control on the "photic zone" where solar radiation, liquid

water, nutrients, and air provide habitat for a rich microbial community within the upper few meters of an ablating glacier (Fig. 1) (Irvine-Fynn and Edwards, 2014). Physical controls on these ecohydrologic interactions, however, have only recently been explored and remain poorly understood, especially the seasonal evolution of depth-variable ice density, permeability, water storage, and microbial mobility (Cook et al., 2016; Irvine-Fynn et al., 2012; Stevens et al., 2018).

Despite the hydrological and ecological implications of weathering crust for supraglacial processes, no studies have described the physical structure or documented sub-surface meltwater storage for the Greenland Ice Sheet weathering crust. When present, the weathering crust could provide a temporary storage reservoir, thus modulating meltwater delivery to supraglacial channels, crevasses, and englacial hydrologic systems (Karlstrom et al., 2014). In

addition, because mass may be removed from the weathering crust without detectable change in glacier surface height, the growth and decay of the crust may confound estimates of sub-seasonal surface mass balance made from ice surface elevation change or surface energy balance models that neglect its presence (van den Broeke et al., 2008; Munro, 1990). Weathering crust structure and hydrologic storage is therefore an important but understudied

component of the Greenland Ice Sheet bare ice ablation zone. The purpose of this study is to describe the physical structure and hydrologic storage of the weathering crust in a mid-elevation Greenland Ice Sheet supraglacial catchment. We provide an initial set of measurements of near-surface ice density, porosity, water saturation, and water table height, and use these data to estimate meltwater storage within the weathering crust. To illustrate the implications of our findings, we extrapolate this storage estimate across the study catchment for comparison with proglacial meltwater runoff volumes. Finally, we discuss broader implications of the findings for ablation zone hydrology and surface mass balance processes to guide future work.

## 2 Data and methods

The data presented in this study were collected during a 6–14 July 2016 field campaign in the middle ablation zone (67.049$^o$ N, 49.022$^o$ W, 1215 m a.s.l.), in the Kangerlussuaq sector of the Greenland Ice Sheet. We measured near-surface ice density, effective porosity, presence/absence of sub-surface water saturation, cryoconite hole depth, cryoconite hole water levels, and ice surface topography. Measurements were made along an 800 m transect in a 63.1 km$^2$ moulin-terminating supraglacial river catchment in the bare ice ablation zone (Fig. 2). We supplement these transect measurements with daily records of meteorological variables recorded by the nearby PROMICE/GAP KAN-M automatic weather station (AWS) ([www.promice.org](www.promice.org)) (van As et al., 2017). KAN-M is located ~8.3 km ENE of our field site at ~1270 m a.s.l. and is the closest AWS to our study site (Fig. 2).

### 2.1 Density and stratigraphy of near-surface ice

At 80 m intervals along the 800 m transect (Fig. 2), shallow ice cores 0.9–1.1 m deep were collected with a 7.25 cm diameter Kovacs Mark III coring system (www.kovacsicedrillingequipment.com). Cores were collected and processed adjacent to the core sites on 11 July (#4–10; Fig. 2) and 12 July (#1–3; Fig. 2) between 14:00 and 21:00 local time. Core stratigraphy observations recorded in field notes include the presence of liquid water, ice lenses, and air bubbles. Natural breaks were used to separate the cores into individual segments. Each segment's length and diameter were measured to the nearest 0.1 cm with a calliper and weighed to the nearest 0.1 g on an Acculab digital scale to determine the ice density of each segment. The natural break segmentation yielded a 13 ± 6 cm mean sampling interval. Individual segments ranged from 3 to 40 cm in length.

At six sites (core #1, 2, 4, 5, 9, and 10), the upper 14–30 cm of ice lacked sufficient cohesion for intact removal with the coring system. To obtain density measurements for this material, ice samples were removed adjacent to the core sites with a Snowmetrics© (www.snowmetrics.com) 1000 cm$^3$ wedge-type steel snow density sampler (Fig. 3). In typical usage, the snow density sampler is inserted horizontally into the sidewall of a snow pit to obtain undisturbed

snow samples. For this study, ice samples were obtained by inserting the 20 cm sampler vertically downward into the ice. To our knowledge, this instrument has not been used for ice density studies on weathering crust but was highly effective for our purposes. These measurements provide bulk density estimates for the upper 20 cm for the six aforementioned sites, however the density measurements may be more representative of the uppermost ~6 cm of ice

because of the shape of the sampler (see Fig. 3). Missing data between 20–30 cm depth for cores #1, 4, 5, and 9 were gap filled with linear interpolation. Together, the shallow ice core and density sampler measurements provide depth-density profiles to depths ranging from 0.9–1.1 m. The nominal 1 m coring depth was selected based on the expectation that weathered ice would not extend below the 1 m depth of the drill barrel. For additional context, two 1.8 m cores were extracted but ice density measurements were not undertaken, these cores are described further in

Sect. 3.3.

Density measurement uncertainty cannot be quantified with known accuracy as each ice core segment was unique in size and shape. Based on visual inspection, we consider 1.5 cm (~10%) uncertainty in ice core segment length to be conservative. This 10% measurement uncertainty primarily accounts for loss of material at the irregular ends of the

ice core segments, which would tend to result in overestimated volume and underestimated ice density. Additionally, it is possible some interstitial meltwater remained in the ice cores when weighed, resulting in overestimated mass and ice density. The cores were held vertically and drained when extracted, and drainage continued prior to analysis and weighing adjacent to the core sites. Nevertheless, some interstitial meltwater likely remained. Estimates of depth-dependent glacier ice water content range from 0–9% though 15 of 18 independent estimates range from 0%

to less than <3.4% (Pettersson et al., 2004). Such water retention errors would tend to cancel with overestimated volume errors, though to an unknown extent. We consider both sources of error to be poorly constrained within the ±10% limits, which we consider sufficiently conservative without giving undue confidence to either the measurements or the error estimate. This uncertainty is incorporated into calculations of density and porosity, propagating into ±14% specific water storage uncertainty (see Sect. 2.2 and Sect. 2.4).

**2.2 Effective porosity of near-surface ice**

The porosity of the near-surface ice was examined to determine the liquid meltwater storage capacity of the study area weathering crust. In theory, the total porosity of a solid material is the ratio of pore space volume to total volume and is calculated from the ratio of measured density to pure material density (Dingman, 2002):

$$\phi_T = 1 - \frac{\rho_M}{\rho_T} \geq \phi_{eff}, \tag{1}$$

where $\phi_T$ (-) is total porosity, $\rho_M$ (g cm$^{-3}$) is measured density, $\rho_T$ is solid material density (0.917 g cm$^{-3}$ for pure ice), and $\phi_{eff}$ (-) is effective porosity, or the porosity effectively available for water storage. Because glacier ice contains

closed air bubbles that are unavailable for water storage, the $\phi_{eff}$ can be less, but not greater than $\phi_T$, and cannot be calculated directly from ice density. Instead, $\phi_{eff}$ must be estimated by measuring the ratio of interconnected pore volume to total ice volume.

To measure $\phi_{eff}$ we used the 1000 cm³ weathering crust ice samples extracted with the snow sampler described above. Twenty-five samples were collected in total, one at each core site and fifteen additional sites at random along the transect. Samples were immediately weighed to determine $\rho_M$. Liquid water sourced from nearby flowing rills was then applied to the levelled ice-filled sampler until the water level was coincident with the ice surface (i.e., until the interconnected pore space was filled with water). $\phi_{eff}$ was computed as the ratio of the water volume required to

fill the sample to the 1000 cm³ ice sample volume. We restricted our measurements of $\phi_{eff}$ to ice sampled from assumed dry weathering crust, but it was not possible to control for the effect of residual liquid water content. Air bubbles and ice crystals were observed for signs of melt and none were observed.

      To estimate $\phi_{eff}$ throughout the shallow ice core samples (where $\phi_{eff}$ was not measured) an "error-in-variables"

(EIV) linear model (York, 1968) was computed between coincident point measurements of $\phi_{eff}$ and $\rho_M$ obtained with the snow sampler. EIV refers to a general class of methods for fitting a straight line to experimental data when measurement errors are present in both the independent and dependent variables. The method has been widely applied in geophysical research when measurement errors are considered important (Brutsaert and Lopez, 1998; York, 1968). The important feature is that EIV regression accounts for error in both the independent and dependent

variables when determining the slope and intercept of the straight line. The model is identical in form to a standard ordinary least squares regression but contains additional error terms:

$$\hat{\phi}_{eff}^* = \alpha + \beta \cdot (\rho^* + \eta) + \varepsilon, \tag{2}$$

where $\phi_{eff}^*$ and $\rho^*$ are the 'true' but unobserved effective porosity and ice density, $\hat{\phi}_{eff}^*$ is the EIV estimate of effective porosity, $\eta$ and $\varepsilon$ are the measurement errors (10%), and $\alpha$ and $\beta$ are the intercept and slope, respectively.

The exact solution procedure is described in York et al., (2004). The $\alpha$ and $\beta$ estimates are then applied to the shallow ice core $\rho_M$ to estimate $\phi_{eff}$ for each shallow ice core segment.

**2.3 Depth to liquid water saturation**

At 8 m intervals along the 800 m transect, the presence/absence of liquid water saturation within the weathering crust, the depth of cryoconite holes, and the depth to water within cryoconite holes were measured with respect to

the ice sheet surface. First, the presence/absence of liquid water saturation was assessed by drilling a 1 m deep hole into the weathering crust with a 5 cm diameter Kovacs auger. The drilled holes were monitored for liquid water

refilling within 30 minutes as an indication of sub-surface water saturation. Second, the nearest cryoconite hole within a 1 m radius of each measurement interval was identified and the total depth of each hole and the depth to water in each hole below the surface were measured. The height of water in each hole is calculated as the difference between the depth of the hole and the depth to water. The depth to water in the holes is used as an estimate of the depth to liquid water saturation (i.e. the water table height). Absence of cryoconite holes was noted if none were present within a 1 m radius of the 8 m measurement interval.

As an additional qualitative check on the weathering crust structure, a Snowmetrics© steel pointed depth probe was forced downward adjacent to each 1 m drilled hole until impenetrable ice was encountered. The expectation was that these measurements would approximate the depth to the shoulder of the sub-surface density profile, roughly corresponding to the depth of rotten unsaturated ice as per Fig. (1) in Müller and Keeler, (1969). Initial field observations confirmed the upper few tens of centimetres of ice was composed of weakly-bonded, coarse-grained ice that was easily removed with a flat bladed shovel and penetrated with the depth probe. The depth probe measurements are used as a qualitative description of the weathering crust structure in Sect. 3.3.

**2.4 Estimating water storage in the weathering crust**

The total volumetric water storage $S$ in the weathering crust is defined as:

$$S = S_P + S_{CH} + S_{cap}, \tag{3}$$

where $S_P$ is free-draining liquid water storage within the weathering crust ice matrix, $S_{CH}$ is liquid water storage in cryoconite holes, and $S_{cap}$ is irreducible liquid water held under capillary tension within the weathering crust. The focus of this work is $S_P$, which we estimate with the following relationship:

$$S_P = \phi_{eff} \cdot D_P, \tag{4}$$

where $\phi_{eff}$ is the effective porosity of the saturated porous ice within the weathering crust and $D_P$ is the thickness of saturated porous ice. Eq. (4) is applied to each segment of porous ice in the extracted ice cores, where $\phi_{eff}$ is calculated from the segment's $\rho_M$ (Eq. 2), and $D_P$ is the measured thickness of each segment. We exclude the thickness of unsaturated ice in each core, as estimated by the average depth to water in cryoconite holes measured adjacent to the core sites, which we show is relatively constant along the transect (Sect. 3.3). The ±10% measurement uncertainty for $D_P$ and $\phi_{eff}$ ($\eta$ and $\varepsilon$, Eq. 2) propagate into ±14% uncertainty for $S_P$:

$$\Delta S_P = \sqrt{\eta^2 + \varepsilon^2} = 14\% \tag{5}$$

The segment $S_P$ ($\pm\Delta S_P$) values are then summed across each core and reported as lower and upper values for specific storage (cm).

Finally, for illustrative purposes we scale our $S_P$ estimate to the study catchment by multiplying the lower and upper values for $S_P$ estimated from the shallow ice cores by the bare ice surface area of the study catchment (63.1 km²). In terms of total water storage, this calculation is conservative since it assumes there is no water storage below the ~1 m depth measured with our field equipment and excludes storage within cryoconite holes and unsaturated storage. However, it also assumes the ice density, porosity, and saturation conditions measured along the transect are representative of conditions across the entire catchment. Recognizing this uncertainty, we caution that it is meant for illustrative purposes.

## 3 Results

### 3.1 Density and stratigraphy of near-surface ice

Throughout the study area, the ice sheet surface was characterized by a layer of coarse-grained, weakly-bonded ice, a few tens of centimetres thick (Fig. 3). Bulk $\rho_M$ of this material measured to 20 cm depth with the snow sampler is 0.45 ± 0.05 g cm⁻³, and ranges from 0.33–0.56 g cm⁻³. This is much lower than typical glacier ice densities of 0.83–0.90 g cm⁻³ (Cuffey and Paterson, 2010), but is consistent with previous findings of ice densities < 0.50 g cm⁻³ in the upper few tens of centimetres of weathering crust (Müller and Keeler, 1969; Schuster, 2001). Suggestive of its weak bonding, this material was easily removed with a flat bladed shovel, penetrated with the depth probe, and often deformed or collapsed slightly underfoot. Free draining liquid water was not observed in the extracted ice samples and there was no sub-surface water table observed within this upper weathering crust layer, for example when penetrated with the depth probe or when the material was removed with the shovel.

Sub-surface $\rho_M$ averaged across the ~1 m depth from the shallow ice cores is 0.69 ± 0.10 g cm⁻³, and ranges from 0.43–0.91 g cm⁻³. In most cores, $\rho_M$ steadily increased with depth but remained less than solid ice density across the entire depth profile, suggesting substantial sub-surface ablation across the ~1 m depth sampled by the cores (Fig. 4). Sampling resolution limited our ability to resolve the depth-density profile in the upper few tens of centimetres, but the observed profiles are generally consistent with the expected non-linear increase in weathering crust ice density from the upper few tens of centimetres to unweathered glacier ice across the upper 1–2 m (Irvine-Fynn et al., 2011; Müller and Keeler, 1969) (Fig. 1). However, densities less than <0.50 g cm⁻³ were found at depths of 50 cm, 40 cm, and 92 cm at core sites 2, 5, and 8, respectively, and density variability was twice as large across the depth of the shallow ice cores than the upper 20 cm (0.10 g cm⁻³ and 0.05 g cm⁻³, respectively).

The source of this density variability likely corresponds to core stratigraphy. While coring, alternating weak and resistant layers were qualitatively observed based on the resistance to downward motion. This structure was confirmed by the presence of alternating layers of coarse-grained (> 1 cm), weathered ice and clear, solid ice lenses in all cores. The ice lenses were readily identified in the core stratigraphy and removed intact from the granular, friable ice between lenses (Fig. 5). The ice lenses contained visible closed air bubbles trapped in clear solid ice. Subtle evidence of internal melting along coarse grain edges was visible in some ice lenses, but most were solid with minimal or no apparent evidence of weathering. Densities of these lenses were not measured in the field but based on their solid structure are estimated to be in the range of typical glacier ice densities (e.g. 0.83–0.90 g cm$^{-3}$; Cuffey and Paterson, 2010).

Previous analyses of weathering crusts have not reported ice structure, therefore the pattern we find of alternating coarse-grained, weathered ice and clear, solid ice cannot be compared to previous studies (e.g. Hoffman et al., 2014; Müller and Keeler, 1969; Schuster, 2001). Though refrozen meltwater lenses are found in firn at elevations above the study area (Cox et al., 2015; Machguth et al., 2016), refrozen meltwater lenses are unlikely in a bare ice, ablating weathering crust (Schuster, 2001). Rather, the observed stratigraphy likely reflects differential weathering of the underlying structural ice fabric (Hudleston, 2015). Surface expression of differential weathering is visible as contrasting dark and light areas along the transect (Fig. 2), similar to kilometre-scale foliated bands associated with outcropping of stratified impurities in the study region (Wientjes et al., 2012). At the scale of the shallow ice cores, stratified distributions of crystal size and shape, bubble elongation and distribution, and impurity content with depth could each influence rates of sub-surface radiative heating (Brandt and Warren, 1993; Liston et al., 1999) and hence could promote differential weathering of centimetre-scale foliated ice layers at depth (Hudleston, 2015). Meltwater advection along micro-seams and cracks, or along foliated planes with enhanced permeability (Wakahama et al., 1973) could provide an additional differential heat source at depth, either via enhanced rotting of temperate ice (Nye, 1991) or, if transported to cold ice, via meltwater refreezing. The ice lenses, then, may represent structural resistance to weathering, and/or result from heterogeneity in sub-surface flow paths that promote differential weathering of sub-surface ice. We would thus expect lenses to be localized features, which helps explain the lack of consistent stratigraphy among cores. Mechanism aside, the $\rho_M$ values reported in Fig. 4 were calculated from the mass of each ice core segment measured prior to removing the ice lenses, and therefore represent the bulk density of each segment (i.e. weathered ice + lens ice). These features are discussed further in Sect. 4.

### 3.2 Measured and estimated effective porosity

Effective porosity $\phi_{eff}$ measured with the snow sampler is 0.44 ± 0.05 and ranges from 0.33–0.56. Measured values were generally smaller than the theoretical upper bound total porosity ($\phi_T$) calculated from $\rho_M$ (Fig. 6), likely owing to observed closed air bubbles in the porous ice grains that decrease the density without increasing the porosity. This result suggests our measurement technique was accurate, as data points above the dashed line would be physically implausible. A significant linear relationship was found ($\phi_{eff} = -0.97\rho_M + 0.89$; $r^2$ = 0.53, RMSE = 0.03), and was used to predict $\phi_{eff}$ from the shallow ice core $\rho_M$ (Fig. 4, top axis). Predicted $\phi_{eff}$ averaged across all core segments is 0.22 ± 0.11 and ranges from 0.002–0.47. Though lower on average (and more variable) than the range of $\phi_{eff}$ measured with the snow sampler in the upper 20 cm of crust, this range suggests substantial porosity across the ~1 m depth sampled with the shallow ice corer. However, the narrow range of $\rho_M$, and structural differences in the ice sampled with the corer, are sources of uncertainty when extrapolating outside of the measurement range (i.e. $\phi_{eff}$ < 0.35 and $\phi_{eff}$ > 0.55 have greater uncertainty).

### 3.3 Evidence of saturation from drilled holes and cryoconite holes

The ice surface topography along the study transect was highly variable across short spatial scales (<10 m) (Fig. 7). Qualitatively, the surface was characterized by hummocks and hollows separated by shallow rills (often flowing) and pitted cryoconite deposits. Water-filled cryoconite holes were ubiquitous across the study area surface, though variability in cryoconite hole water levels and spatial coverage was observed. For example, at 14 of the 100 measurement locations no cryoconite holes were present within the nominal 1 m observation radius and at 9 locations all cryoconite holes within the 1 m radius were dry at the time of observation. At the remaining 77 locations cryoconite holes contained measurable water levels. Cryoconite holes were 25.2 ± 11.4 cm deep and water levels were 15.5 ± 7.8 cm below the ice sheet surface, equivalent to water heights of 9.7 ± 7.8 cm above hole bottoms (Fig. 7b). The height of water in these holes likely varied diurnally and could have steadily drained or filled during the study period (Cook et al., 2016), thus the 15.5 cm average depth to water likely represents a snapshot of the transient water table surface. As such, the presence/absence of water in cryoconite holes may have also varied during the study period. With respect to distance along the transect, there was a trend toward shallower holes (-0.012 cm m$^{-1}$, p<0.005) but no trend in depth to water below the ice sheet surface. Rather, cryoconite hole water levels generally mirrored the 8 m scale topographic variability (Fig. 7b). We measured the water level in a single hole at each 8 m interval and thus cannot quantify sub-8 m scale water table height variability, but these measurements suggest the sub-surface water table drains to seeps and supraglacial channels at <8 m spatial scale.

In 83 of 100 drilled 1 m holes, water from surrounding ice refilled the hole within the nominal 30 minute post-drilling observation period. Refilling rates were not systematically measured but were observed to vary from nearly instantaneous refilling before the auger was removed, to relatively slow (and incomplete) refilling over the 30 minute observation period, suggesting substantial variability over short spatial scales. In addition to this rapid refilling and the widespread presence of water-filled cryoconite holes, all but one of the ten shallow ice core boreholes were observed filling with water during the post-drilling period, though the equilibrium height of water in these holes was not measured. Collectively, these measurements suggest the ice was saturated across the entire 800 m transect to a depth of at least 1 m, albeit with substantial spatial variability in refilling rates.

Based on these observations, we characterize the near-surface ice as composed of two continuous layers with varying thicknesses. The upper layer consisted of low-density (0.33–0.56 g cm$^{-3}$), unsaturated weathering crust ice with relatively uniform crystal structure and no marked stratigraphy. This ice was readily probed, easily removed with a shovel, and often deformed or collapsed under foot. This layer was penetrated to 11.3 ± 5.8 cm (maximum 49 cm) with the depth probe. Beneath this layer was a higher density (0.43–0.91 g cm$^{-3}$), saturated ice layer that we could not excavate with the shovel nor penetrate with the depth probe. The transition between layers was marked by a distinct increase in material strength across a short (~4 cm) distance below which the shovel and depth probe could not penetrate. Though inferred from the shovel and depth probe, this transition likely marks the non-linear increase in density on the shoulder of the theoretical depth-density curve (Fig. 1). The transition roughly coincides with the 15.5 cm average depth to water measured in cryoconite holes, suggesting a possible link between ice density and water table height.

The vertical structure of the higher density, saturated ice was highly variable, consisting of alternating layers of coarse-grained, porous ice and clear, solid ice lenses. The thickness of the saturated ice layer could not be definitively determined with the drilling equipment. However, at two locations shallow ice cores 1.8 m deep were extracted. The densities of these cores were not measured, but at both sites the ice cores consisted of coarse-grained, porous ice alternating with clear, solid ice lenses across their entire depth. There were no qualitative differences between the ice in these cores and the ice presented in Fig. 4 and Fig. 5. At one of these two sites, weathered ice persisted to 1.8 m depth. At the other site, a 20 cm thick segment of solid ice was found between 1.6 and 1.8 m, possibly marking the transition to cold, solid, impermeable ice at this location.

**3.4 Meltwater storage in the near-surface ice and the seasonal context**

Averaged across the 94 cm mean depth of the 10 shallow ice cores, $S_P$ is 14–18 cm (Table 1). The average $\rho_M$ and $\phi_{eff}$ are 0.69 g cm$^{-3}$ and 0.22, respectively. As each ice core was unique in terms of sampling interval, the $\rho_M$ and $\phi_{eff}$

are depth-weighted mean values whereas $S_P$ is summed across each core. $S_P$ is lower in cores 1–3 owing to higher sub-surface $\rho_M$ and hence lower $\phi_{eff}$ below ~50–60 cm depth, whereas $S_P$ in cores 4–10 consistently ranges between 14 and 21 cm, owing to lower $\rho_M$ across their entire depth. While these estimates suggest substantial $S_P$ at the time of observation, they should be considered minimum bounds (within ±14% uncertainty) as they do not include the

potential for additional liquid meltwater storage below the measured ice core depths. The methods used in this study did not yield a definitive bound on the thickness of the saturated ice layer, nor the variation in porosity with depth below the range of the shallow boreholes, and thus deeper water storage cannot be ruled out.

Given the transient nature of the weathering crust, it is important to place these findings in a seasonal context.

Antecedent meteorology such as the timing of snowmelt, rainfall, and prevalence of shortwave radiation, would each influence weathering crust growth and decay. Albedo data recorded at the KAN-M automatic weather station (AWS) indicate the spring snow cover melted out on ~8 June, followed by two ephemeral snowfall events on ~16 June and ~25 June (Fig. 8, vertical grey bars). These dates correspond closely to the ~21 June snow disappearance date reported for the Kangerlussuaq region in Tedstone et al. (2017), based on data from the Modèle Atmosphérique

Régional (MAR) regional climate model (Fettweis et al., 2017). AWS data indicate the ice surface was actively melting prior to 12 July in response to positive air temperatures (>0°C) and positive net shortwave radiation (Fig. 8). Ice surface ablation rates averaged 1.85 cm d$^{-1}$ during this period but were relatively low compared to the peak daily ablation rates (>5 cm d$^{-1}$) recorded between 15 July and 01 August, suggesting conditions were favourable for weathering crust development. AWS data indicate ~74 cm of cumulative ice surface ablation occurred prior to

collection of the shallow ice cores on 11–12 July, equivalent to 66.6 cm water equivalent assuming solid ice density of ~0.90 g cm$^{-3}$. The inferred 14–18 cm $S_P$ is therefore equivalent to ~21–27% of the cumulative seasonal ice surface ablation recorded prior to 12 July, or ~11–14% of the ~1.25 m average annual surface ablation at KAN-M (van As et al., 2017).

Further, MAR data suggests conditions during summer 2016 favoured weathering crust growth in the study region. These include below average cloud cover and rainfall, and above average downward shortwave radiation (e.g. compare to 2000–2016 period, Fig. 1–4 in Tedstone et al., 2017). These meteorological conditions suggest that the presence of a well-developed weathering crust in the study area at the time of observation is not surprising, though inferring a likely thickness is not possible without a physical model for weathering crust development. The AWS data

presented in Fig. 8 provide context for our study, but a detailed investigation of weathering crust formation is well beyond our scope here. Nevertheless, the >1.6 m thickness of weathered ice we find is perhaps surprising given the ephemeral snow cover and ~26 June snow disappearance date suggested by the AWS albedo data. These data suggest the conditions we document developed over a relatively short period of exposure to solar radiation, or persisted

during the previous winter, further suggesting structural controls unrelated to penetration of shortwave radiation may underlie the observed weathering crust structure.

## 4 Discussion

We have presented measurements of near-surface ice density which, to our knowledge, provides the first
characterization of the structure and hydrologic storage of a bare ice weathering crust in the Greenland Ice Sheet ablation zone. These data suggest 14–18 cm of liquid meltwater was stored within porous, low-density ice at the time of observation, and that substantial sub-surface melting may occur in the Greenland Ice Sheet bare ice ablation zone. Together, these findings suggest hydrologic processes in the bare ice ablation zone are affected by porous ice, and that surface lowering measurements may not accurately quantify total mass loss during periods of weathering crust
growth and decay in the Greenland Ice Sheet ablation zone.

### 4.1 Weathering crust structure and hydrologic storage

Water storage in the weathering crust has been reported (Irvine-Fynn et al., 2011; Larson, 1978) but is generally not considered a significant component of water storage in supraglacial environments, owing to its transient nature (Fountain and Walder, 1998; Jansson et al., 2003; Müller and Keeler, 1969). While more work is required to
determine the spatial extent and seasonal evolution of the conditions found in this investigation, our documentation of a saturated weathering crust storing up to 18 cm of liquid meltwater supports the possibility of a substantial transient reservoir in Greenland's bare ice ablation zone, consistent with observations of weathering crust for supraglacial environments worldwide (Irvine-Fynn, 2008; Larson, 1978; Munro, 1990). Though a snapshot characterization, the weathering crust structure presented in Fig. 7 is consistent with conceptual models of the near-
surface weathering crust-cryoconite hole hydrologic system (e.g. Fig. 1; Irvine-Fynn and Edwards, 2014; Müller and Keeler, 1969) and confirms this system is present in the Greenland Ice Sheet ablation zone. The ubiquity of water-filled cryoconite holes, the rapid refilling of drilled holes with liquid water, and the excavation of saturated ice cores to depths >1.6 m suggests the study area weathering crust acts as a depth-limited aquifer (Irvine-Fynn et al., 2011), storing meltwater in the seasonally-temperate near-surface ice and likely delaying the delivery of meltwater to
supraglacial streams and rivers via saturated sub-surface flow (Irvine-Fynn et al., 2011; Karlstrom et al., 2014; Munro, 2011).

In addition to meltwater storage, we describe the structure of the weathering crust. We find a pattern of porous, granular ice alternating with solid ice lenses in the upper 1-2 m of weathering crust in the study area, rather than a
homogeneous rotten near-surface ice layer (e.g. Müller and Keeler, 1969). Given the rapidly ablating ice surface prior

to the study, we posit the solid ice lenses are emergent structural features, as refrozen meltwater is unlikely in an ablating weathering crust (Schuster, 2001). Though beyond the scope of the data collected in this study, we hypothesize two mechanisms to explain the observed stratigraphy. First, stratified distributions of crystal size and shape, bubble elongation and distribution, and impurity content with depth could influence rates of sub-surface radiative heating (Brandt and Warren, 1993; Liston et al., 1999). The ice lenses may then represent optically transparent ice layers with larger crystal size, lower air bubble content, or lower impurity content. The optical properties of these layers may reduce absorption of shortwave radiation, substantially reducing internal melting relative to optically opaque layers. Second, meltwater advection along micro seams, cracks, or foliated ice layers with enhanced permeability may promote differential melting via sensible and frictional heat transfer (Hambrey, 1977; Hambrey and Lawson, 2000; Wakahama et al., 1973). Therefore, underlying structural features such as foliation, cracks, and fractures caused by thermal expansion (Sanderson, 1978) may be accentuated by differential radiative heating, enhanced "rotting" by meltwater along preferential flow paths, or heating due to meltwater refreezing. Together, these suggest weathering crust formation in the study area may be more complicated than previous descriptions of a process driven solely by solar radiative heating (Hoffman et al., 2014; Müller and Keeler, 1969), and suggest meltwater dynamics and ice structure may be important controls on weathering crust development.

Though we interpret the lenses as structural features, there is evidence that internal refreezing of meltwater occurs in weathering crust on the Dry Valley glaciers in Antarctica (Hoffman et al., 2014). Although the climatic context is different, this raises the possibility of meltwater refreezing within the weathering crust ice matrix in Greenland. If so, refreezing would represent a heat source within near-surface ice, and a possible sink for meltwater retention (Pfeffer et al., 1991). Though detailed energy balance studies suggest internal refreezing is negligible in near-surface porous ice on alpine glaciers in the Canadian Rockies (Paterson, 1972; Schuster, 2001), such analyses have not been performed for the Greenland Ice Sheet ablation zone. Regardless of internal refreezing at depth, we frequently observed night-time refreezing of meltwater at the surface of cryoconite holes and water tracks in the study area (Fig. 9), though the magnitude of this refreezing was not studied. In addition to careful observation of sub-surface ice structure, future work should determine if internally refrozen meltwater occurs within weathering crust in the Greenland Ice Sheet ablation zone, especially during seasonal transitions from temperate to cold near-surface ice.

## 4.2 Estimating meltwater storage of the study catchment weathering crust

While extrapolating these local scale findings to broader areas of the Greenland Ice Sheet is not justified presently, it is illustrative to consider the potential meltwater storage volume of the weathering crust in our study catchment. For example, if we assume our shallow ice core data are broadly representative of conditions across its 63 km² area, multiplying the lower and upper estimate of $S_P$ (Table 1) by the bare ice surface area of the study catchment yields

0.009–0.012 km$^3$ of meltwater storage. To put these numbers in perspective, one hour of peak discharge measured at the Watson River in Kangerlussuaq during the July 2012 record melt event (Nghiem et al., 2012) was equivalent to 0.0115 km$^3$ (van As et al., 2017). Our study catchment is equivalent to ~2% of the ~2800 km$^2$ ablation zone contributing area draining to the Watson River (Lindbäck et al., 2015). Thus, while our 800 m shallow ice core survey

may not be representative of ice density or porosity more widely over the Greenland Ice Sheet ablation zone, even relatively small areas of weathering crust have the potential to buffer large volumes of supraglacial meltwater, potentially delaying its delivery to en-, sub- and proglacial systems. Future work should seek to identify the underlying meteorological controls on weathering crust development in the Greenland Ice Sheet ablation zone to determine the likely spatial extent of the conditions we document.

**4.3 Implications of weathering crust for surface mass balance processes**

Our findings of low-density, saturated weathering crust in the Greenland Ice Sheet ablation zone have at least three implications for Greenland Ice Sheet surface mass balance (SMB). First, sub-surface meltwater generation within the weathering crust does not materially lower the ice surface (Braithwaite et al., 1998; Müller and Keeler, 1969; Munro, 2011). Lateral drainage of internal meltwater through the permeable weathering crust to supraglacial channels

reduces weathering crust ice density, by removing mass with no detectable change in surface height. As a result, mass change during periods of weathering crust development may be underpredicted or, during periods of weathering crust removal, overpredicted, if determined solely from ice surface elevation changes (Braithwaite et al., 1998; LaChapelle, 1959; Müller and Keeler, 1969). In the Kangerlussuaq region of the southwest Greenland Ice Sheet ablation zone, penetration of shortwave radiation into near-surface ice is estimated to generate 20–30% of total

summertime melt, suggesting ice surface elevation change measurements may not be reliable for short-term model validation in this region unless sub-surface melt is accounted for (van den Broeke et al., 2008; Munro, 1990).

Second, the timing, magnitude, and location of meltwater delivery to the englacial system is powerfully altered by surface hydrologic processes operating on the Greenland Ice Sheet bare ice surface (Smith et al., 2017). In addition to

catchment size and shape, transient water storage in the weathering crust has been inferred to attenuate the timing of meltwater delivery to englacial and proglacial hydrologic systems (Karlstrom et al., 2014; Munro, 2011). Typical flow velocities of 0.4–2.6 m s$^{-1}$ in supraglacial meltwater channels on the Greenland Ice Sheet surface are 3–5 orders of magnitude greater than hydraulic conductivity estimates for permeable ice (Cook et al., 2016; Gleason et al., 2016; Karlstrom et al., 2014; Wakahama et al., 1973). Thus, porous sub-surface meltwater flow may modulate delivery of

surface meltwater to supraglacial channels, which in turn deliver meltwater to englacial and subglacial systems. Yet, the Greenland Ice Sheet weathering crust hydraulic conductivity has only recently been investigated (Stevens et al.,

2018) and its effect on meltwater delivery to the en-, sub-, and proglacial hydrologic system is poorly understood (Munro, 1990, 2011; Smith et al., 2017).

Finally, the weathering crust provides a substrate for retention of impurities, cryoconite, and microbial communities
that influence the Greenland Ice Sheet ablation zone surface albedo (Boggild et al., 2010; Lutz et al., 2014; Ryan et al., 2017; Yallop et al., 2012). Cryoconite deposits locally enhance melt, forming quasi-cylindrical melt holes that deepen into the weathering crust (Gribbon, 1979), likely reducing their direct effect on mesoscale ice albedo patterns in southwest Greenland (Ryan et al., 2016; Tedstone et al., 2017). Conversely, interstitial water within the weathering crust, such as that documented in this study, provides abundant habitat for microalgae and cyanobacteria (Irvine-
Fynn and Edwards, 2014), which reduce ice surface albedo (Yallop et al., 2012). Sub-surface water exchange may further redistribute soluble impurities and microbes between the permeable weathering crust and cryoconite holes (Cook et al., 2016), while channel invasion of cryoconite holes during periods of weathering crust removal may disperse cryoconite sediments and microbes across the ice surface (Hodson et al., 2007; Takeuchi et al., 2000). Thus, while it has not been confirmed, weathering crust hydrology, in addition to its growth and removal, could modulate
the distribution of impurities and microbial communities on the Greenland Ice Sheet ablation zone surface, and hence could influence surface albedo patterns.

Underpinning each of these implications of weathering crust, however, is the transient nature of its growth and decay. Our study provides a snapshot characterization of what appears to be a deeply developed weathering crust,
approximately midway through a summer characterized by below average cloud cover, albedo, rainfall, and spring snow depth, earlier than average snow disappearance, and above average downward shortwave radiation (e.g. compare to Fig. 1–4 in Tedstone et al., 2017). These conditions suggest abundant time for weathering crust development, and lack of conditions conducive to its removal or decay. Interannual variability in these conditions is substantial, and the conditions we document may not be representative of normal conditions. The net seasonal effect
of weathering crust processes on Greenland Ice Sheet ablation zone hydrology and mass balance remains poorly understood and should form the basis for future work.

**5 Conclusion**

This study suggests presence of a water-saturated weathering crust at least 1 m thick on the bare ice surface of the Greenland Ice Sheet ablation zone. The observed characteristics of this weathering crust are similar to those
described for supraglacial environments worldwide (Cook et al., 2016; Hoffman et al., 2014; Irvine-Fynn and Edwards, 2014; Karlstrom et al., 2014; Larson, 1978; Müller and Keeler, 1969; Munro, 2011). Namely, the weathering

crust acts as a depth-limited aquifer (Irvine-Fynn et al., 2011), storing liquid meltwater and likely slowing its transport to supraglacial streams via porous sub-surface flow (Cook et al., 2016; Karlstrom et al., 2014). Our empirical relationship ($\phi_{eff} = -0.97\rho_M + 0.89$) between measured ice density and measured ice porosity at the study field site suggests 14–18 cm of meltwater storage within weathering crust at our study site. If these findings are representative of broader areas of the Greenland Ice Sheet ablation zone, they suggest the potential for substantial sub-seasonal meltwater storage within porous low-density ice on the Greenland Ice Sheet ablation zone bare ice surface. Future work should examine how spatio-temporal changes in the surface energy balance and underlying ice structure control weathering crust development, and quantify potential errors in sub-seasonal mass balance and surface elevation change estimates derived from surface energy balance models and altimetry, as most currently neglect removal of mass due to sub-surface melting in the bare ice ablation zone.

**Data availability**

All field data is available upon request from the corresponding author and is currently in preparation for submission to an online open access data repository.

**Author contribution**

M.G. Cooper, L.C. Smith, and A.K. Rennermalm designed the experiment. M.G. Cooper, C. Miège, A.K. Rennermalm, L. H Pitcher, J. Ryan, and S. Cooley collected the field data. J. Ryan assisted with Unmanned Aerial System image processing. M.G. Cooper performed the data analysis. M.G. Cooper wrote the manuscript with contributions from all authors. The authors declare they have no conflict of interest.

**Acknowledgements**

This project was funded by the NASA Cryosphere Program grant NNX14AH93G (P.I. Laurence C. Smith) managed by Dr. Thomas P. Wagner. We thank Professor Robert Hawley of Dartmouth University for the generous lending of the shallow ice corer. We thank Polar Field Services for their field support, Charlie Kershner (George Mason University), Brandon Overstreet (University of Wyoming), Sasha Leidman (Rutgers University), and Rohi Muthyala (Rutgers University) for their field work assistance. Data from the Programme for Monitoring of the Greenland Ice Sheet (PROMICE) and the Greenland Analogue Project (GAP) were provided by the Geological Survey of Denmark and Greenland (GEUS) at http://www.promice.dk.

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

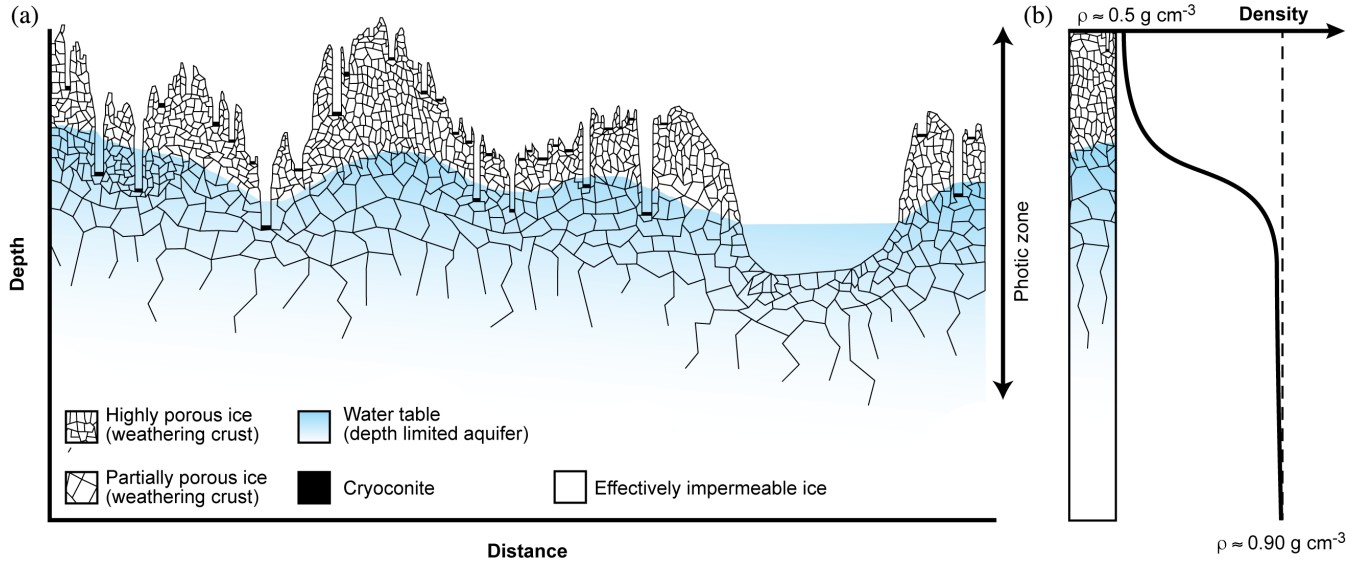

Fig. 1. (a) Conceptual diagram of weathering crust structure, highlighting the porous ice layers, cryoconite holes, and
saturated water table (adapted from Irvine-Fynn and Edwards, 2014 and Müller and Keeler, 1969). (b) Theoretical sub-
surface depth-density profile showing the non-linear increase in ice density from the highly porous, low density near-
surface ice to higher-density, unweathered glacier ice (adapted from LaChapelle, 1959).

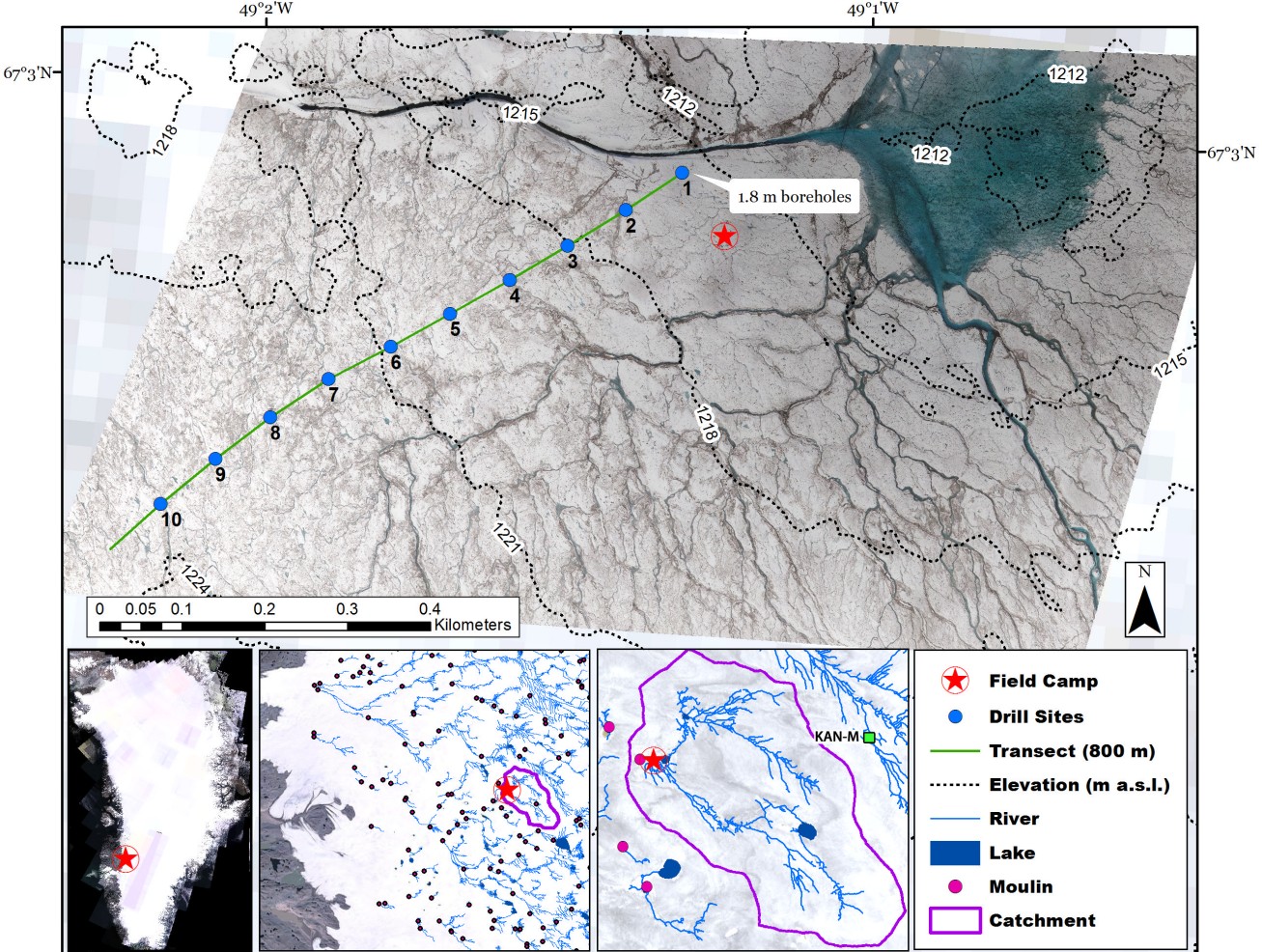

**Fig. 2.** Ortho-rectified image mosaic of the study area at 6 cm ground resolution from RGB camera imagery collected 10 July 2016 on board a quad-copter drone. Background 30 m Landsat image collected same day. Shallow ice cores extracted at 80 m intervals (blue circles) along the 800 m transect provide ice density measurements to depths of 1.1 m, with two additional shallow ice cores extracted to 1.8 m depth at interval 1. Insets (below) show the 63.1 km² supraglacial catchment extent (magenta outline), as delineated from WorldView satellite stereo-photogrammetric digital elevation model topography, and supraglacial river and moulin locations derived from Landsat 8 imagery (Yang and Smith, 2016).

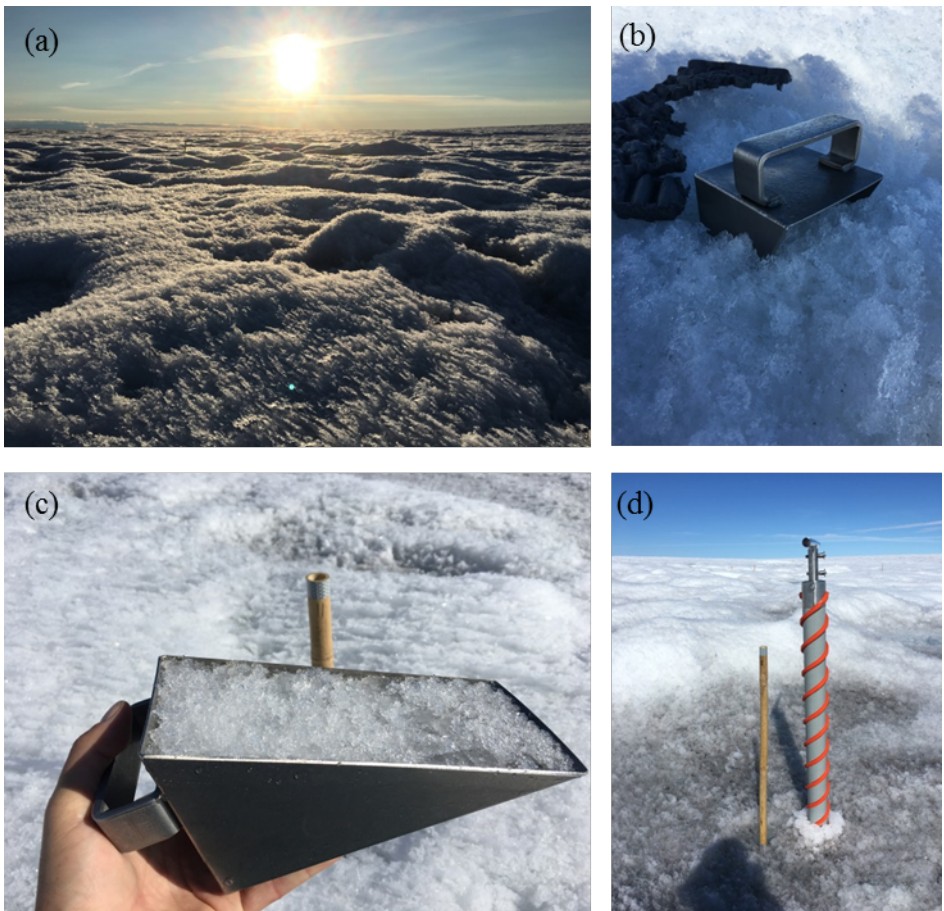

Fig. 3. (a) A surface weathering crust was pervasive throughout the study area, characterized by small scale topographic variability and cryoconite holes. (b-c) A 1000 cm³ steel snow density sampler was vertically inserted into the upper 20 cm weathered ice. (d) A shallow ice corer was used to obtain ice samples to depths of 1.8 m.

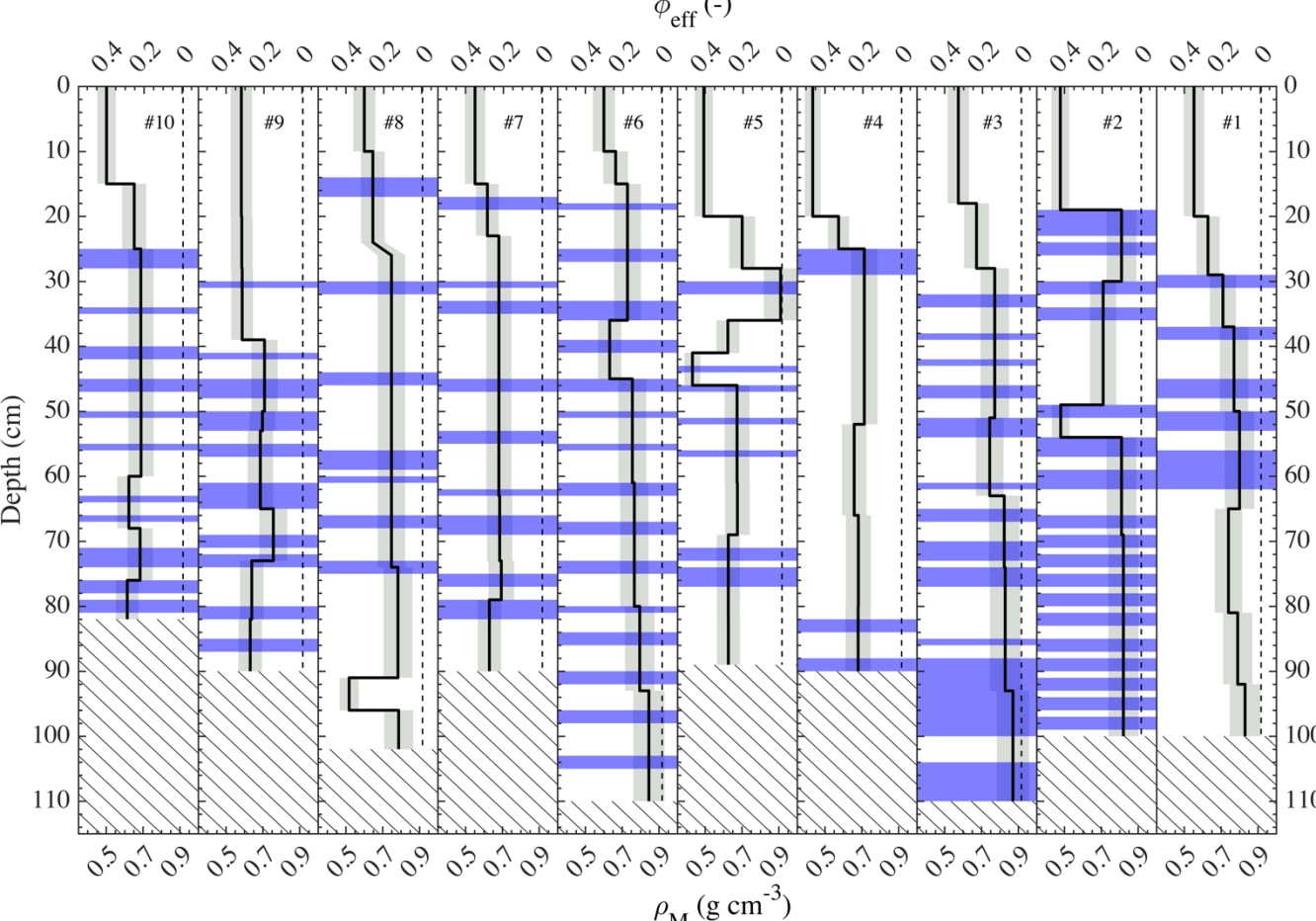

Fig. 4. Sub-surface measured ice density ($\rho_M$) and corresponding calculated effective porosity ($\phi_{eff}$), and stratigraphy profiles from 10 shallow ice cores (#10-1, left to right) extracted at 80 m intervals along the study transect (see Fig. 2 for ice core locations). Horizontal blue shading represents solid ice layers. Vertical dashed line at solid ice density 0.917 g cm$^{-3}$. Assumed ±10% measurement uncertainty represented by shaded grey bars. Hatched areas are no data.

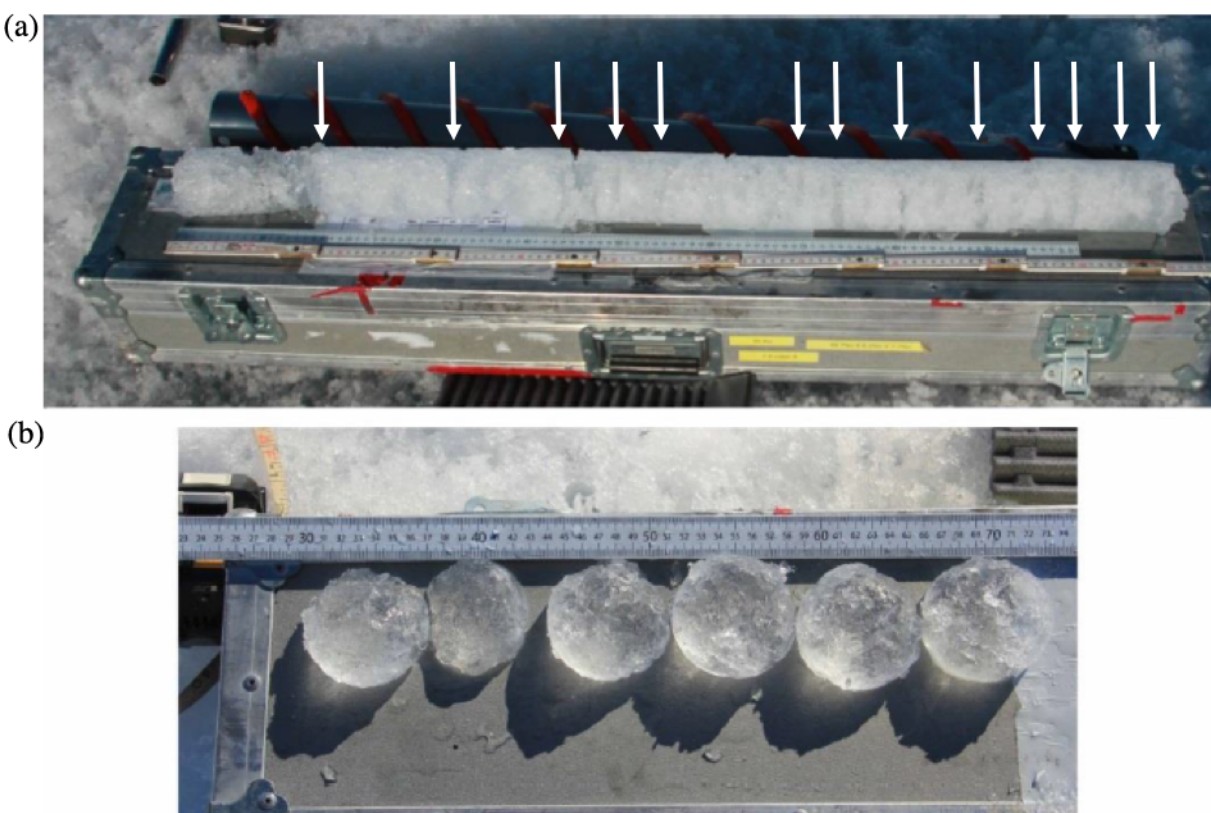

**Fig. 5. (a)** Typical near-surface shallow ice core (core #6) prior to in situ analysis of density and stratigraphy. Clear, solid ice lenses alternate with granular, fractured ice. Approximate locations of ice lenses noted with white arrows (not all lenses are clearly visible). **(b)** Ice lenses removed and confirmed after completed core analysis (core #1).

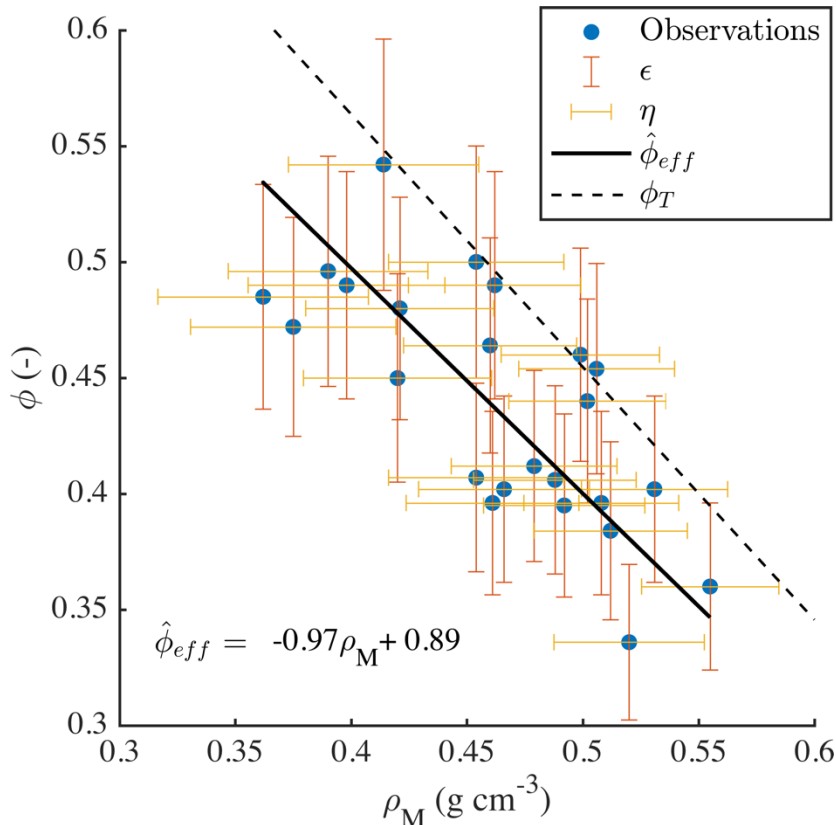

Fig. 6. Linear relationship ($\hat{\phi}_{eff}$, solid line) between measured ice density ($\rho_M$) and effective porosity ($\phi_{eff}$) and assumed ±10% measurement error (whiskers). Dashed line is theoretical upper limit where effective porosity equals total porosity (i.e. $\phi_T = \rho_M/\rho_T$).

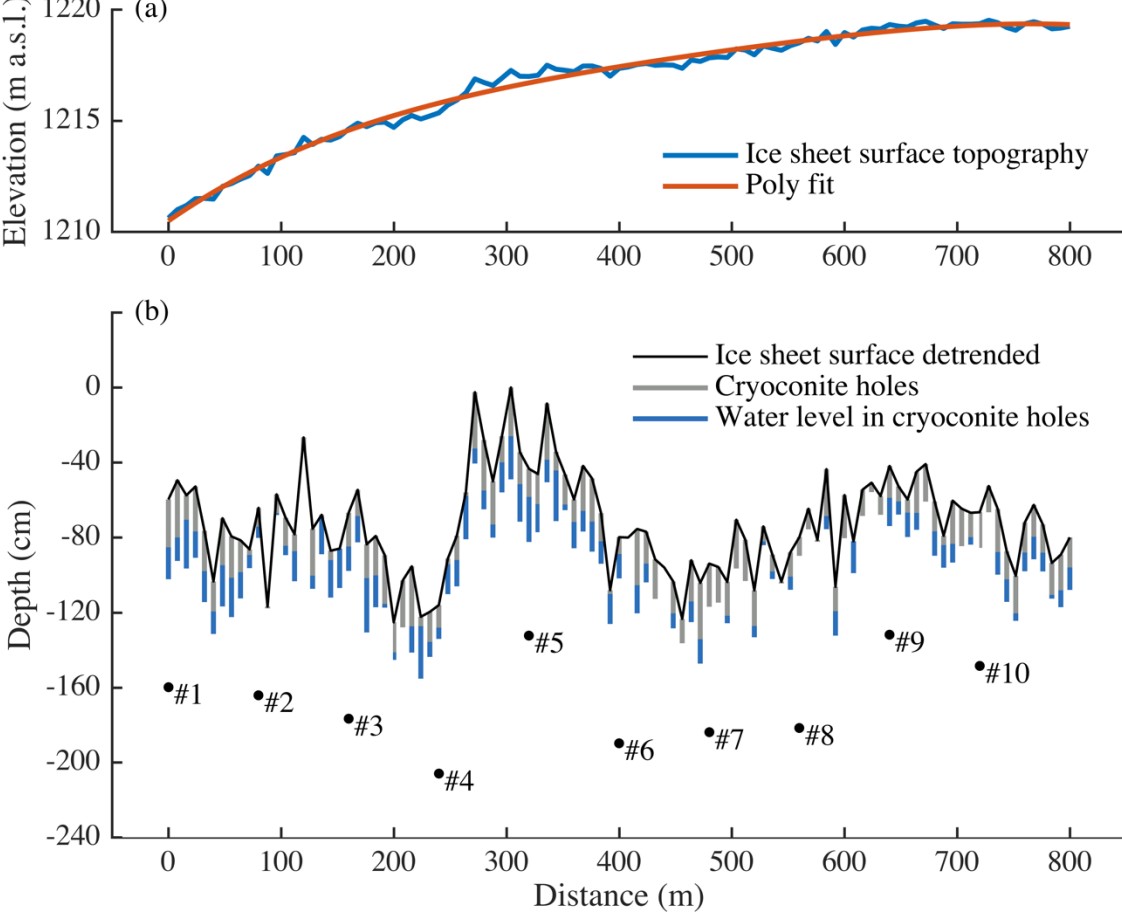

**Fig. 7. (a)** Ice sheet surface topography along the 800 m study transect extracted from a 6 cm resolution stereo-photogrammetric digital elevation model derived from RGB imagery collected 10 July 2016 from a quad-copter drone and the 2nd-order polynomial best fit. **(b)** Ice sheet surface topography detrended with the polynomial best fit, cryoconite hole depths (vertical grey bars), and cryoconite hole water levels (vertical blue bars) sampled along the 800 m study transect, adjusted to a common vertical reference. Locations of the 10 shallow boreholes and their depth relative to the detrended surface are labelled #1-10.

**Table 1: Shallow ice core depth, mean core density, mean core porosity, and specific storage depth ($S_P$), for each shallow ice core.**

| Core | Ice Core Depth (cm) | Mean Core Density (g cm$^{-3}$) | Mean Core Porosity (-) | $S_P$ (cm) |
|---|---|---|---|---|
| 1 | 100 | 0.72 | 0.19 | 12 – 16 |
| 2 | 100 | 0.72 | 0.19 | 11 – 15 |
| 3 | 100 | 0.76 | 0.15 | 10 – 13 |
| 4 | 90 | 0.63 | 0.28 | 15 – 21 |
| 5 | 89 | 0.63 | 0.27 | 16 – 21 |
| 6 | 97 | 0.74 | 0.17 | 15 – 20 |
| 7 | 90 | 0.65 | 0.26 | 15 – 20 |
| 8 | 102 | 0.72 | 0.19 | 15 – 20 |
| 9 | 90 | 0.64 | 0.26 | 16 – 21 |
| 10 | 82 | 0.64 | 0.27 | 14 – 18 |
| **Average:** | **94** | **0.69** | **0.22** | **14 – 18** |

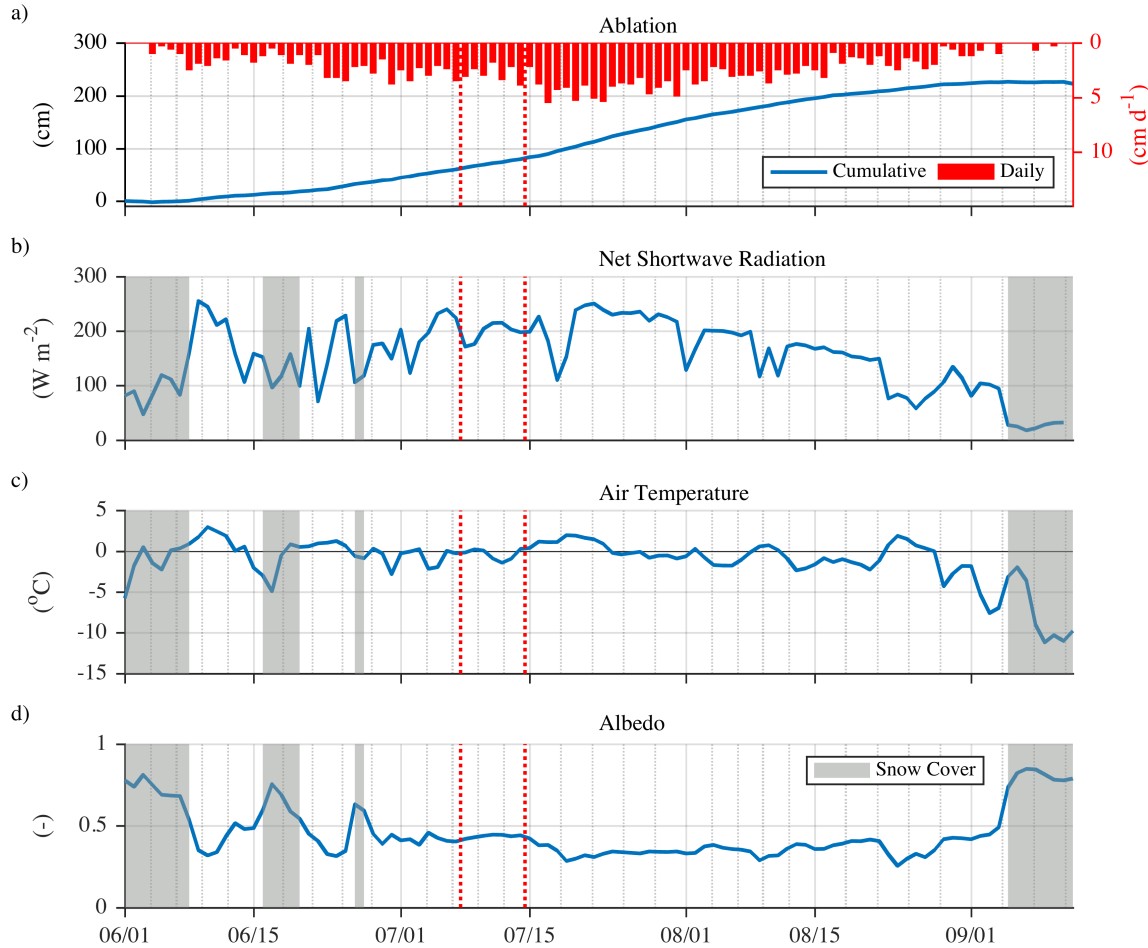

**Fig. 8. Meteorological records of a) daily and cumulative ice surface ablation, b) net shortwave radiation, c) 2 m air temperature, and d) albedo, for the period 01 June 2016 – 15 September 2016. Data were collected by the PROMICE/GAP KAN-M automatic weather station (for location see Fig. 2). Vertical grey shaded bars indicate time periods when albedo was greater than >0.5, indicating snow cover was likely present. Albedo is calculated from the ratio of outgoing solar radiation to incoming solar radiation measured ~2 m above the ice sheet surface at the KAN-M station (www.promice.org).**

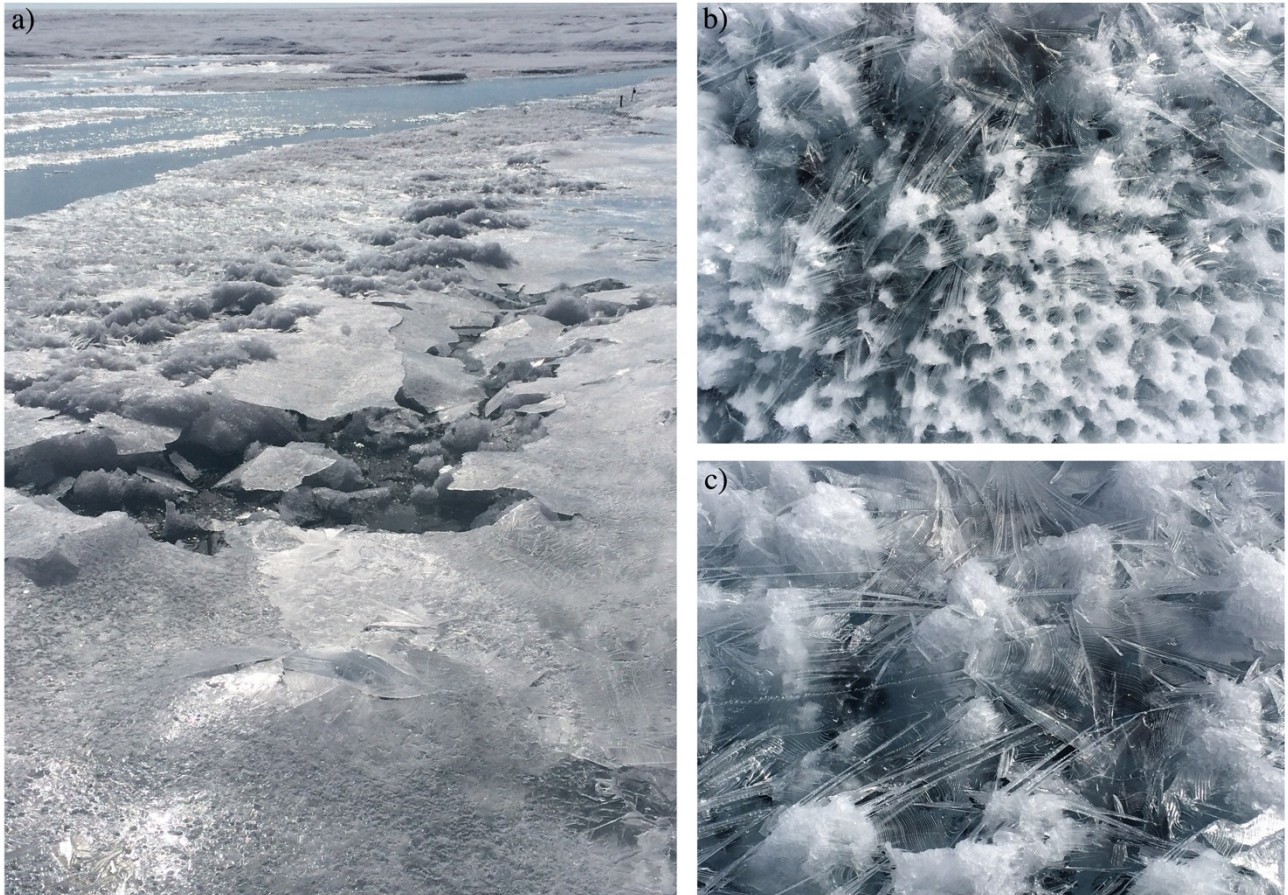

**Fig. 9. Night-time refreezing of meltwater at the surface of a) water tracks (~10 m length scale), and b) cryoconite holes (~0.1 – 1 m length scale) was frequently observed during the field study. Photos were collected by the first author during the 6 July – 12 July 2016 field campaign between 04:00 and 07:00 local time.**

