# Peer review of "Meltwater storage in low-density near-surface bare ice in the Greenland Ice Sheet ablation zone"

_The Cryosphere, 2017_

## Referee Comment (RC1) · Anonymous Referee #1 · 27 Jul 2017

The paper presents the results of a recent field survey conducted to assess the potential for subsurface meltwater storage in porous ice in the ablation zone of the Greenland ice sheet. Focusing on a small, internally draining, hydrological catchment in the much-studied South West, the authors find that the subsurface 'reservoir' consists of two layers. These consist of a thin, light, unsaturated layer atop a thicker ($\sim$1 m), denser, saturated layer. Between them, these layers provide storage potential for up to 20 cm of meltwater which, integrated over the catchment, is the equivalent of one hour's proglacial discharge from this sector.

The methods employed in the study are sensible, and their results are very interesting,

however the manuscript is a little confused in places and I would have liked to see the field data discussed in more detail. That said, this is a good paper and I expect will make a solid contribution to the literature subject to editing as follows.

Major comments 1. The paper is lacking in analysis of spatial variability along the transect studied. For example Fig 3 reveals that ice lenses are not always common between adjacent cores. Further investigation into these features and why they arise would be both interesting and serve to strengthen the manuscript.

2. While you do not have seasonal data, you should contextualise your findings in terms of the seasonal cycle. For example what is the ablation rate in this location? Is your weathering crust likely to be the product of one, or more melt seasons? You attribute ice lenses to remnant glacier ice but to me it seems that these are evidence of meltwater refreezing at depth. Can you correlate the incidences of ice lenses to e.g. the annual cycle or specific weather events?

3. While it is good that you discuss the implications of your findings for SMB and surface hydrology studies, I would like to see a bit more qualitative information here. For example, what would the difference in ice mass be in your catchment a) at the density of ice and b) when you account for sub-surface porosity? Similarly, hydrological studies use estimates of snow permeability for water routing according to Darcy's Law. Can you provide an updated estimate of sub-surface permeability in your study area for future use by such studies?

Minor comments: Page 1 Line 29-30: I'm not sure that meltwater throughput 'reinforces concernes about ... sea level rise' Line 29: Greenland ice sheet not Greenland Ice Sheet.

Page 2: Line 3: Mention that these models assume that runoff is instantaneously lost to sea here. Line 8: Sentence structure is odd here; implies that the stored meltwater is the substrate. Is that what you mean? Line 16: Maybe mention melting due to friction from the flow of meltwater. Line 31: 'could potentially' rather than 'would'

[Figure]

Page 3: Line 2: logical disconnect here, add an explanatory line. Line 6: phrasing of 'near surface ablating' seems strange. Line 12: Add melt zones onto map in Fig 1 Line 16: delete 'study area' Line 27: example of logical jump; you assign and uncertainty then say where you got it. It would be better to say, 'we consider 1.3 cm (10%) accuracy to be conservative'. Line 29: just measurement uncertainty or a combination of measurement and instrument uncertainty?

Page 4 Line 12: What determined the maximum depth? Line 28: be consistent with units; you used cm3 before.

Page 5 Line 6: delete 'these' Line 7: justify this given the difference in structure between the two layers. Line 20: nearest cryoconite hole? Nearest x cryoconite holes? Line 27: another logical jump re: transition! Line 30: to what height? It would have been good to measure the water table at the drilled holes as well as at Cryoconite holes.

Page 8 Line 8: Impermeable yes, but how continuous? Are these highly localised features?

Page 9 Line 7: I think the value below snow surface would be a better one to quote. Line 10: Again, to what level? Completely full? Line 17: did you see any evidence of flow in any of the holes? Line 25: quantify 'often' Line 29: Ok so the water table is ~20 cm below the surface yes? Which is consistent with your statement that the bottom layer, from 20cm down is saturated.

Page 10: Line 29-30: This is speculative without seasonal data. If the acquifer is perennially saturated then this is not necessarily the case. Fig 3, reorient so #10 is on the left as in fig1. Fig 6 b add core depth.

---

## Referee Comment (RC2) · Anonymous Referee #2 · 29 Aug 2017

This paper presents findings from a field campaign on the Western margin of the Greenland ice sheet concerning the nature of the 'weathering crust' on the bare ice in the ablation zone. The paper provides measurements from shallow ice cores of ice density and corresponding porosity as well as water content, finding surprisingly low density ice down to at least 1 meter depth. It is pointed out that the presence of this weathering crust means that (subsurface) melt does not necessarily correspond to 'surface' lowering, as might be measured by satellite altimetry.

I think this is an interesting set of measurements, and is a valuable contribution to understanding the supraglacial hydrology of the Greenland ice sheet. Meltwater storage

in the percolation zone of the Greenland ice sheet (in firn) has been well documented over the last few years, and this study suggests that a non-negligible amount of water storage and transport may occur beneath the apparent ice surface in the ablation zone too.

The paper is well written and the figures are mostly clear. I have one main comment and a number of minor comments, mostly seeking clarification.

Main comment

The measurements represent a snap-shot of the weathering crust in mid-July 2016 and it is not clear how this relates to the behaviour over the course of the melt season. I appreciate that the field campaign was limited in length so it may not be known how the crust itself evolves, but I think there needs to be more discussion of the setting for these measurement. In particular, what is the annual ablation rate in this region? At what stage of the melt season are these taken (i.e. roughly how much melting has already occurred here)? What is the ice temperature in this region? These are important issues in understanding how reflective these results are of wider spatial scales but also larger time-scales.

In particular, I think there could be more discussion of how the inferred stored water thickness (15-22cm) relates to the amount of melt that has so far been produced this season (roughly what fraction of it is this), and how the depth of the porous ice compares with the amount that melts each year. Eg. are the ice lenses that form the result of recent refreezing (i.e. earlier the same year) or some earlier time?

There could then be some discussion of how the porous ice evolves over the course of the year. Presumably all the water freezes again in winter? In which case the ice is mostly solid at the start of the melt season (except perhaps the unsaturated surface layer, which seems to have a similar porosity to snow)? If the saturated subsurface water doesn't ever run-off, it may simply be changing the quantity of runoff rather than delaying it.

Specific comments

Why are the findings frequently referred to as 'preliminary'? What are they preliminary too? If they are really preliminary, it begs the question why they are being published. I'd suggest that if the authors think the results are worth publishing they should not refer to them as preliminary (which does not preclude doing more work on the topic).

Page 1, line 29 - why does the routing of surface water to the ocean 'reinforce concerns' about contribution to global sea level rise? Isn't such melting part of the 'normal' operating cycle of an ice sheet?

Section 2.1 - it is not clear from this description how liquid water in the core is dealt with. Is it allowed to drain out? Presumably there is still quite a lot of water trapped in the core samples (due to capillary forces) and this contributes to the measured mass?

page 4, line 9 - this is a bit awkward wording, since this statement presumably assumes that the density is uniform (independent of depth). Perhaps better just to say that the geometry of the sampler means that the near-surface ice is disproportionately weighted in this average, rather than quantifying the 'center of mass'.

Section 2.3 - there is some confusion here about the 'unsaturated weathering crust depth', and how it relates to how penetrable the ice is. Reading further, it seems that the 'water table' (which I would interpret as the unsaturated depth) roughly coincides with a change in the strength of the ice that is presumably what the depth probe is detecting. It does not seem obvious to me why these two surfaces (the impenetrable ice surface and the water table) should happen to coincide - perhaps the presence of air in the pores above this allows the surface ice to 'rot' more rapidly. Or perhaps the permeability of the upper layer is sufficiently large that water in this layer readily runs off horizontally keeping it unsaturated.

Perhaps the qualitative description of the surface given on page 9, line 20, could be moved forward to the method section to help explain these issues. In any case it would
[Figure]

Interactive
comment

help to be clearer precisely what is meant by unsaturated - does this mean there is no liquid water, only residually-trapped water, or that water does not fill the pore space (all of which are different)?

Page 6, line 5 - why is refrozen meltwater included as water storage? If it has refrozen it is ice again and should be be thought of as storage (it requires melting again - with the associated energy implications - before it could run off).

Page 6, line 19 - is the 'potential' liquid storage capacity not just the effective porosity multiplied by depth and total area (i.e. including the currently unsaturated pore space too)?

Page 8, line 7 - there seems to be some subjectivity involved here. Why is estimating the value wrong in one direction deemed 'not problematic'? If the densities were measured including the ice lenses, would it not make sense to use the the volume including the ice lenses when converting to water content using the effective porosity? Or otherwise use a solid ice density of the ice lens to infer the density of the non-ice lens part of each segment?

Page 10, line 4 - the drill did not go below 1.8m for fear of freezing. Did you make any measurements of temperature in the porous crust? It would be helpful to know if the ice is all at the bulk melting temperature or if it goes below this at depth.

Page 10, line 20 - what is meant by a storage 'rate'? I could not work out what this number means.

Page 11, line 15 - the 'lower and upper mean' is a strange concept; perhaps the 'mean lower and upper values' would be better wording.

Page 11, line 21 (and conclusions) - why are the results not considered representative of the rest of the ablation zone? I understand the desire for caution given that this is only one location, but without other evidence (perhaps you have it?) wouldn't the default assumption be that the results do apply more widely? What do you think is special

about your field site that means the results would not apply more widely? Perhaps you could just say 'We do not know whether these findings represent typical conditions....' rather than 'not proposing' it.

---

## Referee Comment (RC3) · Anonymous Referee #3 · 8 Sep 2017

This paper reports on the findings from a field campaign on the ablation zone in south-western Greenland. The focus of the paper is the so-called "weathering crust" that characterises glacier and ice sheet surfaces, and its potential hydrological storage role. The authors use a set of shallow ice cores (n=10) to describe the variability in near-surface ice density over depths of < 2 m. From these observations, the authors explore and effective porosity of the near-surface ice, and examine a potential water storage based on observations of a water table evident within the weathering crust. A specific storage of ∼0.2 m is derived, suggesting that at the time of observations a water volume equivalent to 1 hour's worth of discharge from the local supraglacial catchment was essentially stored within the weathering crust.

[Figure]

The findings are a useful demonstration that this weathering crust exists on the Greenland Ice Sheet, and provides a sensible trigger for future work assessing the supraglacial drainage system and its functionality. Although some recent focus in Greenland has included the firn aquifer at higher elevations, it is an interesting insight to an overlooked hydrology of the ablating bare ice sector of the ice sheet. The growing recognition of the supraglacial realm as an ecosystem, and the potential importance of water storage on biogeochemical cycling at the ice sheet surface ensures this is a timely contribution and serves as a useful benchmark in this type of work.

Overall, the paper is well written, sensibly referenced, and the figures are clear. The methods are intelligible and could be repeated, and the calculations utilised are sensible within the limits of the data available/presented.

However, major comments would include:

A stronger description of how the weathering crust forms, and the subtlety of its growth and decay would be beneficial both in the introduction and in the later discussion. Specifically, would you expect a deep weathering crust at the time of your observations? Does the timing of snow melt, dominance of shortwave radiation, absence of rainfall give reason to consider the weathering crust (and ice lenses) you describe? Is this a glacier ice weathering crust or one that perhaps is superimposed ice derived from snow and refrozen lenses forming therein? If this is glacier ice, then you should at least mention ice structure in addition to refreezing processes (particularly given the evidence of marked structure in the locality).

A clearer emphasis regarding the results being a snapshot which reveals something about the supraglacial hydrology of the Greenland Ice Sheet could be beneficial. In your discussion, albeit subjectively, are you able to comment on the likelihood of greater or less storage to be seen at other times of the summer, is this a seasonally progressive hydrological feature or is the observation just that, a discrete observation – there are climate records for the locality which might allow some extrapolation of these ideas.

Appreciably, it is not possible to go beyond perhaps a statement on this, given the limitations of the data set, but It would be helpful.

A slightly strengthened assessment of the uncertainties would be important I feel – this should include an assessment of the water content of the cores themselves as it is not clear if all interstitial water was removed prior to evaluating mass for density estimates. Temperate ice can have interstitial water content of up to ∼10% (see Petterson et al., 2004, JGR), and certainly the saturated lower-most ice in the developing weathering crust may exhibit such water content if this is a seasonally temperate ice layer. Can you perhaps try to assess uncertainties associated with this water content, and the resultant impact on other estimates presented here. The 10% and 10% quoted seemed a little arbitrary when slightly more detailed and thorough approaches could have been taken. Furthermore, can you account for the ablation of the ice surface if cores were not all takes on one day – can the core profile figure be corrected/adjusted for surface ablation – making crude assumption that ablation over transect broadly similar, or using a point estimate from the energy balance? Correcting for the 7 days ablation period might be informative and aid inferences – such that for example, ice lenses may be better aligned perhaps if representative of refreezing events or local thermal conditions.

Could more analysis of the data presented in Figure 6 be made available here? There are opportunities to examine patterns over elevation (small range though that is) and in relation to the detrended surface and 'roughness'. Similarly, it would be interesting to see if there is from the profiles (e.g. what are the patterns of phi-eff at, say, 33cm and 87cm depth, where it looks weathering crust (not ice lens) data is available across all cores – assuming these positions remain if adjusted for ablation over the 7 day sampling) – is there anything to be gained from a slightly deeper examination of the density and porosity data over depth and along the transect?

In places the writing style became less clear, or seemed to have a slightly reduced scientific quality. Similarly, a couple of key references seemed to be absent or choices of references seems a little misplaced, while in other places there was a proliferation

of sources when perhaps just one or two examples would suffice. Some further editing and subtle revisions would likely be beneficial, to strengthen this paper, in addition to perhaps examining a few more relevant publications that would be of help in supporting these results and findings and their significance.

Minor comments and suggestions (some touching on points above) would include:

P1: L1: Suggest hyphenate "near-surface" throughout. (There are some variations, e.g. P3 L13 and L15). L2: "Greenland Ice Sheet", as used throughout the manuscript. It is refreshing to see authors correctly use the appropriate capitalisation for proper nouns (it shouldn't be the Greenland ice sheet, given it is a specific location and entity) and at times I wish publishers would adhere to grammatical correctness – but that is another discussion altogether. L2: "Meltwater storage in low-density near-surface bare ice in the ablation zone of the Greenland Ice Sheet" might read a little better perhaps? L16: suggest refer to this as "specific storage". L17: clarify the water level is depth from the surface or from the base of the auger holes, and is "recharge" a more preferable term than infilling (given this is a hydrology paper). L18: "These observations are consistent. . ." given you present results and discuss them. Analysis might be provisional with clear directions to follow, but don't negate the potential utility of these observations. L21: "supraglacial catchment" L25: a longer opening paragraph would be stronger as an opening. Can there be a clearer link from mass balance or ablation to runoff models for Greenland, and the assumptions regarding the efficient delivery. The sea level aspect here seems misplaced, as the study looks at in-season delays or reductions in discharge. Surely, noting the assumed efficient drainage is now being examined more closely with reference to the firn aquifer and so on would allow for a stronger introduction paragraph here. S L28: what is a "terminal moulin"? And not all runoff goes to moulins – there are supraglacial routes to proglacial regions, and lakes and crevasses. Suggest more circumspect and/clarified text here.

P2: L5: cite Muller and Keeler (1969) for the introduction of the term "weathering crust". Might an additional diagram be helpful here to conceptually illustrate what you are focused on here for the less familiar reader? L6: Fountain and Walder (1998) also note minimal delay to supraglacial runoff and so text and citation here, given phrasing, might be slightly inappropriate. Suggest check the source again. L9: what is a "seasonally temperate glacier"? Poor terminology, please revise. Seasonally temperate surface ice perhaps, but glacier thermal regime is a very different thing. L11: surely the depth is ice-type dependent, and stating "~2m" is not strictly correct. Consider rephrasing (see Cook et al., 2016). It was also surprising that at no point is Munro (1990, AAAR) cited here, a source confirming the subsurface melt and bulk ice density variations leading to uncertainty in runoff volumes at Peyto Glacier. Suggest consideration of this source, especially with regard L19. L17: doesn't lateral meltwater motion result in sensible and frictional heat transfers, contributing to further removal of ice mass. Also suggest clarification over the vertical extension of the weathering crust, and how this influences mass for any given vertical position. The process described by Muller and Keeler (1969) is a little more complex than perhaps is given credence here, and perhaps a more careful description could be afforded. See also Cook et al., 2016. L22: The opening of this paragraph is not entirely appropriate, the structure and the content seems slightly superficial and/or repetitive (e.g. mention of delay in runoff is already in L6). Suggest revisiting this text through to L26 and P3. L22: Is subsurface melting in Antarctic contexts the same as the definition provided of weathering crusts on "temperate ice" (see L9)? Strongly advise some differentiation between subsurface melting and weathering crust terminology. This sentence could be removed at no loss to the paper. L25: Slightly unconvincing use of the literature here: some references focus on cryoconite holes, others on the weathering crust as a habitat. Recommend revisiting, with perhaps consideration of recent messages regarding glacier ecohydrology (e.g. Dubnick et al., 2017a,b, JGR and Hydro Proc.; Hotaling et al., 2017, Env Mic.; Milner et al., 2017, PNAS). Yes, the weathering crust is a substrate for cryoconite holes (see Muller and Keeler's 1969 diagram), but the focus here should be the hydrological aspects and for example disturbance to cryoconite holes that might influence their ecology (Edwards et al., 2011, ISME J; Mieczan et al., 2013, PPR) or distribution

(e.g. Hodson et al., 2007, JGR). Then develop the 'undescribed in Greenland to date' message and the guide of what is to follow (subsequent paragraph). If you do touch on the biogeochemical cycling aspects, it might be helpful to touch on these again in the discussion section.

P3: L1: Remove "In sum" L2-L4: Consider revisiting (see L25 above), and bringing in energy balance and ablation (see again Munro, 1990, AAAR) and describing the reasons for weathering crust relevance. Then have a single paragraph giving the justification for the study in Greenland. I just found these two paragraphs jump around a little and felt that a more logical progression through material could be achieved. L20: delete "mechanical" – not necessary. Be consistent with hyperlinks/formatting if used for www sources. L21: "drilling" in glaciology typically implies more than shallow coring, might just talking about "coring" and "core sites" be sufficient? (e.g. P4 L1 "core sites" seems more appropriate). L23: issue of mass for calculation of density is relevant here. Are you measuring water and ice? If so, are not the estimates of density in error. This issue needs to be addressed and accounted for; ice density has to be properly estimated given the depth variable water content. L29: Suggest "This uncertainty is incorporated into calculations of ice porosity and water content (see Sections 2.2 and 2.4)". In places, as here, writing clarity and conciseness could be tightened up.

P4: L8-9: Clarify the relevance of the centre of mass, if you are using the method to estimate the upper 14-30cm ice, just indicate that the upper 20cm is used, but the sampler geometry results in bias toward the uppermost ice and so leads to an underestimate of density. This just seems to be introducing terms which could be seen as confusing. L11: Issue of water content in core sections and uncertainties in density measurements remains problematic. L12-15: this could be condensed: e.g. "for context, two 1.8m cores were extracted but ice density measurements were not undertaken, these cores are described further in Section 3.3". L15: Estimates of porosity will be affected by ice core sections that were weighed still containing interstitial water. Removal of this water is not a trivial problem, as exposing the core to positive air temperatures will

initiate further melt, and methods of forcing water out via centrifugal force is similarly challenging. An estimate of uncertainty is needed here, and this needs to be incorporated into the data derived from these potentially erroneous ice mass measurements. L20: rationale for the change in ( ) to [ ]? L25: might combining the equations here to A = B > C (as Eq 1) seem a neater and more consistent presentation of the equations? Would allow a slightly smoother explanation. L30: does the time-frame and temperature of the water present any issue here? Given the thermal potential of supraglacial water (which I presume was used?), could you estimate and mass loss (or confirm this is negligible). The size of the weathering crust crystals might be important here – were the samples used representative of the upper 20cm for all sites?

P5: L9: while it is good practice to cite, do we need more than one of two examples here? Just considering journal space. L19: "8m intervals" ? L20 &P6 L2: were cryoconite holes ubiquitous features, or did you measure those proximate to the sample point? Clarify here. If one hole was measured – is this representative of local water table – might measuring 4 holes in at each site have provided more robust estimates? L21: The steel rod measurements are not entirely convincing, can you justify this a little more clearly. Furthermore, as above, a conceptual model might help here. Perhaps you need to consider the density decay curve (LaChappelle, 1959) and clarify your reasoning here, or use some alternative term in terms of a qualitative measure of "weathering crust mechanical resistivity" to the steel probe to indicate perhaps the shoulder on the density decay curve? There is also the issue of capillary draw in the weathering crust, are you able to confirm the water table in the crust is the same as that in the ice matrix? Does the water table truly exist as a broadly consistent level? If not, is this an uncertainty you can at least note if not estimate.

P6: L5: Refrozen water while a component of storage in an overarching sense, is not the liquid storage, and is likely to be a proportion of the total available liquid water following a freezing event or water drainage to a cold front in the ice. If you are talking about liquid "water storage" then surely it is a negative value/term in that it is water

lost to freezing? I'd also caution here given the inference is that the ice lenses are refrozen water – which may or may not be the case (see comment below regarding ice structure) and so a clearer definition of water storage might be helpful here. L10: see L19, but are you exploring a total storage potential or just the saturated ice. Can you remove "saturated" here, and discuss both the observed water storage volume and the potential storage volume? L11: do you not just "extract" cores, rather than excavate them? L19: I think you need to better define the "potential total storage volume" (ie. the entire weathering crust) vs. the estimated snapshot of water storage yielded by your observations – given the weathering crust storage potential will be time-variant given the nature of the weathering crusts formation. You could then discuss the total storage potential (under the conditions at the time of measurement), and the proportion of that which did indeed hold liquid water (the stored volume), and compare those to melt production or runoff volumes.

P7: L24: "metre rule" ?? L32: Given the strong surface expression of structural features in the sector of the ice sheet studies here, you might give an indication that ice structure might underlie this (and perhaps cite papers that note the strong evidence of structural glaciology at the locality, and its theoretical background – for example Hambrey et al, 2000, Geol Soc; Hudleston, 2015, J Struc Geol.). You could at least provide a hypothesis here as a potential guide for future work. Given ice lenses are discussed, and given the ice sheet surface is ablating, these lens features must be emergent – and while it is possible refreezing of meltwater may contribute, do ice temperatures or meteorological conditions support this given the prevalence of these lens features? Were the lenses truly horizontal in formation or exhibit slight orientation?

P8: L3: "The reported pM values therefore" ? Missing word. L4: perhaps use "lens ice" to clarify your meaning here. L6: surely if lenses are refrozen water their density will likely approach that of pure ice, and if structural features, their persistence would suggest higher glacier ice density values. As such, can you not include and quantify potential uncertainties here? L12: You have two data points above the theoretical limit,

so "consistently smaller" is not strictly correct. You also refer to Figure 5 here three times in as many lines – consider using just one reference to the graphic. L15-16: "unphysical" ? Perhaps use "physically implausible". L15 & L18: There seems to be an overemphasis on the quality of the data here, please recall the equation used to derive the porosity value means there is circularity here – the porosity is a function of the two densities. Avoid overstating something here. You can simply report the observed relationship, the fact that how robust this is beyond the bounds of observations is equivocal, and that the relationship was used to estimate porosity. L28: This section seems a little less flowing than others, and is characterised by short paragraphs. Can the core holes and the water levels noted in these be described further? The first paragraph and third surely belong together? But there is repetition here. Consider revisiting this section.

P9: L5: so do dry holes indicate the water table is more complex and not a level surface? L7: not sure you need to use caption detail in the figure reference here. L11: I think you need to define where the ice is saturated – it isn't the full depth of the weathering crust, or is it? Just feel a little more clarity in needed here to ensure the observations and inferences are clearly described. L21: You don't really have a handle on the "transient" nature of the weathering crust here – yes, you can conceptualise this as a two-layered feature. But although you show spatial variability, you have no detail on temporal change. I would focus on the message relating to the snapshot of water storage – and the volume that represents. And only in your discussion, mention the processes of weathering crust formation and how this would mean the depths of the porous and saturated ice would potentially vary. L34: further evidence for structural controls on the ice crystallography?

P10: L4: repetition of freezing leading to cessation of coring from method section, unhelpful here as a result section. L6: It would be nice to see a little more result reporting here – not solely the reference to the table and the mean for all sites. Perhaps expand a little. L13: Just wondered if a clearer summary section leading to discussion might be helpful – in following with the results. For example, open with the lacking recog-

nition of the weathering crust, and how here, observations of ice density revealed X Y and Z on a portion of the Greenland Ice Sheet, then move to how the two layered structure matches previous work, and how density and water storage values compare to the other limited reports. L14: Cite the Larson reports and Irvine-Fynn et al review that discuss near-surface surface storage here. Would citing the Jansson et al (2003, J Hydro) review also be useful here? L15: why the specifics on polythermal ice sheets here? The references cited discuss temperate glaciers and a polythermal glacier, respectively. L21: "stagnating" ??

P11: L1: Recall the reports you cite simply modelled water storage via water budgets – and so you can't compare core observations to hydrological models. Previous work hasn't examined ice cores to identify or report crystallographic changes. Please revisit. L4: see earlier comments on ice structure. L8: see earlier point about water budget equation, and the ice lenses being a negative 'storage' value, as indicated here. However, a stronger physical discussion of the potential formation processes for the ice lenses is needed – with comparison to ice structure and any alternative explanations too. L10: GrIS – either define and use as acronym throughout or use Greenland Ice Sheet as elsewhere. L11: Condense to a single paragraph section perhaps? L25: I'd suggest revisiting in view of the Munro (1990, AAAR) source.

P12: L24: Lutz reference focuses on ice algae, not cryoconite. Suggest Wientjes and Boggild references would be more appropriate here. Similarly, L25: Fountain discussed ice-lidded cryoconite in Antarctica which may physically be a little different – suggest a more cautious use of literature which refers to the types of feature and observations that are characteristic for Greenland (e.g. the older Gribbon, 1979, J Glac. or Gadja, 1958, Can Geogr. references for cryoconite holes in Greenland).

P13: L12: Does Hoffman's study relate to a temperate or polythermal ice mass – isn't it cold? Or just remove the thermal regime aspect here – "supraglacial environments elsewhere…" L15: You define the symbology, no need to repeat the definition here in L16, after its use on L15. L19: For impact, suggest you rephrase as "if these observations are representative of the Greenland Ice Sheet ablation zone, then wider implications are..., and future work should...."

Fig.3.: Consistency with (..) or [..] on axes labels. The captions seems to be overly long – focus on the content and remove superfluous text. Label 1 -10 in the figure. Hatched areas are "no data" not "core depth" are they not? Can you not include the snow-shovel data here for the uppermost 20cm – albeit in a different colour, for comparison and completeness? Might inclusion of potential ablation here be helpful given from the field campaign description, the cores were collected over one week during which time ablation would take place – and such that (for example) a refreezing event (if this is what the lenses are) might be more clearly identified if lenses appear at the same depth relative to a zero set for the period of coring?

Fig.4.: Is the lower image for the core in the upper? Perhaps use arrows to indicate where ice lenses are on the core.

Fig.5.: y-axis should be phi-eff. The equation given should be phi-hat-eff (inconsistent symbology). Surely "observations" not "data"? Caption – is "measured data" needed here?

Fig.6.: (b) there is a lot of information here, and I just wonder if two panels here would be helpful – one to give clearer indication of the water level in holes with a simple zero as ice surface, and then the detrended plot with the unsaturated crust estimate? The two grey tones are hard to differentiate. If detrended, surely the data should be scattered around zero – so did you offset this to a maximum positive deviation - one presumes so, but clarification would be appropriate? Have you compared distance or elevation against any of the variables – are there any other patterns to explore – as these don't seem to have been mentioned in the main text – even if to confirm there is no elevation dependency.

Table 1: could you include a column of mean phi-eff for each core here, for ease of direct comparison?

[Figure]

---

## Author Comment (AC1) · 18 Oct 2017

Dear reviewer,

Thank you for the thorough and supportive comments. The suggestion to contextualize the study in terms of the seasonal cycle and to quantify the effect of sub-surface porosity on ice mass were particularly fruitful. I provide a line by line reply to each comment below, and include revised figures at the end of the document. A revised manuscript will be uploaded following receipt of instructions from the handling editor. Thank you kindly,

Matthew Cooper

**Anonymous Referee #1**

**Received and published: 27 July 2017**

The paper presents the results of a recent field survey conducted to assess the potential for subsurface meltwater storage in porous ice in the ablation zone of the Greenland ice sheet. Focusing on a small, internally draining, hydrological catchment in the much-studied South West, the authors find that the subsurface 'reservoir' consists of two layers. These consist of a thin, light, unsaturated layer atop a thicker (<1 m), denser, saturated layer. Between them, these layers provide storage potential for up to 20 cm of meltwater which, integrated over the catchment, is the equivalent of one hour's proglacial discharge from this sector.

The methods employed in the study are sensible, and their results are very interesting, however the manuscript is a little confused in places and I would have liked to see the field data discussed in more detail. That said, this is a good paper and I expect will make a solid contribution to the literature subject to editing as follows.

**Major comments**

**1.** The paper is lacking in analysis of spatial variability along the transect studied. For example, Fig 3 reveals that ice lenses are not always common between adjacent cores. Further investigation into these features and why they arise would be both interesting and serve to strengthen the manuscript.

*Author response:* We have added substantial interpretation of the ice lenses that will help explain the lack of consistent stratigraphy in adjacent cores. We believe the ice lenses are structural ice features, not refrozen meltwater. For the crust to develop, the ice must be temperate, which should prevent substantial refreezing. Diurnal refreezing could occur, but if so it seems unlikely that weathering would proceed at depths beneath refrozen ice lenses without also weathering the lenses, thus preventing a progressive stratigraphy from developing. We think it is more likely the lenses are remnant solid ice that has undergone less weathering owing to structural heterogeneity in ice grain size, bubble size/content, and impurities. We also speculate that meltwater advection along micro seams or cracks in the ice may promote differential weathering, similar to joint block weathering of terrestrial lithology. Finally, ice lens stratigraphy in firm is highly variable and discontinuous over spatial scales as short as 1.5 m (Brown et al.,

2011; Machguth et al., 2016), thus spatial analysis of lens features is unlikely to prove fruitful. Each of these would suggest lenses are local features, helping to explain the lack of consistent stratigraphy among cores. In the revised, we have added substantive descriptions of these ideas to support our hypothesis that the lenses are structural features, not refrozen meltwater.

To address spatial variability in cryoconite holes along the transect we tested for statistically significant linear relationships between distance and 1) depth of cryoconite holes, and 2) depth to water in cryoconite holes. No relationship was found for depth to water though there was a slight trend toward shallower holes (-0.014 cm m-1, p<0.004). We report these findings in the revised Sect. 3.3, along with a strengthened discussion of variability along the transect.

**2.** While you do not have seasonal data, you should contextualize your findings in terms of the seasonal cycle. For example, what is the ablation rate in this location?

*Author response:* Thank you for the excellent suggestion to contextualize our study in terms of the seasonal cycle. As requested, we obtained daily measurements of ablation recorded by the KAN-M automatic weather station (AWS) during our field study. KAN-M is located ~8.3 km ENE of our field site at ~1270 m a.s.l. and is the most proximal AWS to our field site (1215 m a.s.l.). Sonic ranging data recorded by KAN-M indicate the maximum spring snow depth was ~50 cm and the snow disappearance date was ~21 June, which suggests the conditions we document developed over an ~21-day period between snow disappearance and the collection of the ice cores on 11-12 July. Following snow disappearance, AWS data indicate a cumulative ice surface lowering of ~55 cm prior to collection of the shallow ice cores on 11-12 July. The average ablation rate during this time was 2.65 cm d-1. The mean annual ablation rate at KAN-M is ~1.25 m a-1 (van As et al., 2017). These statistics are reported in the revised Sect 3.4 paragraphs 3-4.

To supplement these data we added a comparison with a recent publication that examines dark ice dynamics in the study region (Tedstone et al., 2017). Their analysis of regional meteorology from the Modèle Atmosphérique Régional (MAR) regional climate model (Fettweis et al., 2017) suggests ~50 cm average snow depth and mid-June snow disappearance date (see Figure 3 in Tedstone et al., 2017), consistent with the AWS data. Further, their analysis suggests that meteorological conditions during summer 2016 were ideal conditions for weathering crust development. These include below average cloud cover and rainfall, and above average downward shortwave radiation (e.g. compare to Figures 1–4 in Tedstone et al., 2017). From these data and the AWS data, we conclude it is not surprising that a well developed weathering crust was present in the study area at the time of observation. We have added several discussion points throughout the manuscript to emphasize the seasonal context.

2. (continued) Is your weathering crust likely to be the product of one, or more melt seasons?

*Author response:* We interpret the literature to suggest the weathering crust is a seasonal phenomenon with no interannual carryover except in the case of stagnant ice (e.g. Fountain and Walder, 1998). On sub-seasonal timescales, the weathering crust can rapidly decay when the surface energy balance is dominated by longwave or turbulent heat fluxes that melt the surface, removing the crust and exposing solid ice. However, it is conceivable that a deep weathering crust could persist to the end of the melt season if meteorological conditions allow. In this case,

we would expect interstitial and surficial meltwater to refreeze following snowfall. Annual snowfall in this part of Greenland is typically <1.0 m (Tedstone et al. 2017) and therefore meltwater that does not drain from the crust likely refreezes during winter and/or at night. If snow cover is absent or ephemeral, the crust may sublimate over winter. However, interannual variability in these conditions is likely substantial and we prefer not to comment on interannual carryover without a thorough analysis. That said, we have added a note in the revised that the mean annual ablation rate is ~1.25 m a-1 at KAN-M, thus given that we document weathered ice at depths > 1.5 m it is conceivable that near-surface, low density ice persists on interannual timescales.

**2. (continued)** You attribute ice lenses to remnant glacier ice but to me it seems that these are evidence of meltwater refreezing at depth. Can you correlate the incidences of ice lenses to e.g. the annual cycle or specific weather events?

*Author response:* As noted above, we do not think the ice lenses are refrozen meltwater but instead we think they are structural features, as suggested by reviewer #3. Further, based on similar analyses of meltwater lenses in firn we do not think the annual cycle or specific weather events will prove fruitful (Brown et al., 2011; Machguth et al., 2016).

**3.** While it is good that you discuss the implications of your findings for SMB and surface hydrology studies, I would like to see a bit more qualitative information here. For example, what would the difference in ice mass be in your catchment a) at the density of ice and b) when you account for sub-surface porosity?

*Author response:* The difference in ice mass in the catchment can be evaluated from the difference between solid ice density and in situ measured density integrated across the depth of porous ice:

$$\Delta m = \int_0^{h_o} \rho(h, t_o) \cdot dh - \int_0^{h_o} \rho(h, t_1) \cdot dh$$
(1)

where *m* is mass, *h* is ice thickness,  $\rho$  is ice density, and *t* is time. Assuming  $\rho(t_o) = 917$  kg m-3, integrating Eq. (1) across the depth-density profiles from our shallow cores yields  $\Delta m = 254$  kg m-2 or 25.4 cm water equivalent. Integrating this across the 63.1 km2 catchment yields  $1.6 \times 10^{10}$  kg or 0.016 Gt.

However, the mass of stored water must also be accounted for. Subtracting the  $17.3\pm4.3$  cm mass of stored water we document from the above estimate yields  $\Delta m = 81\pm43$  kg m-2 or  $8.1\pm4.3$  cm water equivalent. Integrating this across the 63.1 km2 catchment yields  $5.11\times10^9$  kg or  $0.0051\pm0.0027$  Gt. The first estimate (0.016 Gt) can be considered a maximum plausible mass difference as it assumes there is no subsurface stored meltwater, whereas the latter (0.0051 Gt) estimate is a minimum based on the water storage we document. In reality the mass of stored meltwater is time-variant, thus we can provide a snapshot estimate at best.

The effect of subsurface porosity on mass, therefore, depends on the timescale considered, the initial conditions, and, critically, the role of meltwater drainage. We omitted this discussion from the original manuscript to keep it focused on the instantaneous characterization of density, porosity, storage, and weathering crust structure, owing to incomplete knowledge of the initial

ice density profile, the density profile below the depth of the shallow cores, and the fate of the stored meltwater. We are happy to include the above exercise in the revised if requested.

**2. (continued)** Similarly, hydrological studies use estimates of snow permeability for water routing according to Darcy's Law. Can you provide an updated estimate of sub-surface permeability in your study area for future use by such studies?

*Author response:* We agree the results of our study point to the importance of sub-surface permeability but unfortunately, we cannot provide an estimate from our data. We do, however, discuss the topic in the discussion where we cite the four studies we are aware of that provide estimates of ice permeability, and we compare these to estimates of supraglacial channel flow velocities. Our hope is that an interested reader will use these citations as a resource for this topic.

**Minor comments:**

Line 29-30: I'm not sure that meltwater throughput 'reinforces concerns about ... sea level rise'

Author response: The statement is removed, as requested.

Line 29: Greenland ice sheet not Greenland Ice Sheet.

*Author response:* We respectfully submit that our capitalization is correct for a proper noun referring to a geographical place name. Reviewer 3 also expressed a preference for Greenland Ice Sheet.

**Page 2**

Line 3: Mention that these models assume that runoff is instantaneously lost to sea here.

*Author response:* As requested, we have mentioned that these models assume runoff is instantaneously lost to sea. To supplement this, we have added reference to several works that demonstrate substantial time lags and possible meltwater retention in the ablation zone, motivating the study of ablation zone hydrologic processes and near-surface porous ice.

Line 8: Sentence structure is odd here; implies that the stored meltwater is the substrate. Is that what you mean?

*Author response:* Thank you for catching this error. The "weathering crust" is the substrate. As requested, "meltwater storage" has been deleted.

Line 16: Maybe mention melting due to friction from the flow of meltwater.

*Author response:* As requested, this additional source of melting has been noted, but we added this to the interpretation of the ice lenses in results Sect. 3.1.

**Author changes in manuscript:**

Line 31: 'could potentially' rather than 'would'

Author response: 'would' has been changed to 'could', as requested.

**Page 3**

Line 2: Logical disconnect here, add an explanatory line.

Author response: We are not sure what logical disconnect the reviewer is referring to but the introduction has been substantially revised and we hope the problem has been corrected.

Line 6: Phrasing of 'near surface ablating' seems strange.

*Author response:* We have hyphenated 'near-surface' which we hope improves the phrasing. The phrasing is used to emphasize that the study is focused on bare, ablating ice, to avoid possible confusion about firn, snow, or superimposed ice.

Line 12: Add melt zones onto map in Fig 1

Author response: The entire area in Fig 1 was actively melting during the study period.

Line 16: Delete 'study area'

Author response: deleted, as requested

Line 27: Example of logical jump; you assign and uncertainty then say where you got it. It would be better to say, 'we consider 1.3 cm (10%) accuracy to be conservative'.

Author response: We have revised the sentence to read as requested.

Line 29: Just measurement uncertainty or a combination of measurement and instrument uncertainty?

*Author response:* Just measurement uncertainty. We used a pair of calipers and digital scale to make the measurements and we assume the instrumental uncertainty is substantially less than the measurement uncertainty. We have, however, improved our discussion of measurement uncertainty, specifically describing the two primary sources of error we expect are important 1) ice core volume measurement error owing to loss of material near the irregular ends of the individual ice core segments, and 2) interstitial meltwater retention errors owing to capillary water retention and incomplete free water drainage. The volumetric error would tend to result in underestimated ice density, the water retention in overestimated density. Hence, the two would tend to cancel to an unknown extent. Estimates of temperate ice water content range from 0-9%, though most estimates (including all based on in situ calorimetric methods) are in the range 0-3% (Pettersson et al., 2004). Recognizing that both sources of error are poorly constrained, we think our original 10% estimate is sufficiently conservative without giving undue confidence to either the measurements or the error estimate. We have also added a physical constraint that density

cannot exceed solid ice density (917 kg m-3) and effective porosity cannot exceed total porosity  $(1 - \rho_m / \rho_{ice})$ . These issues are presented in the revised methods Sect. 2.1 paragraph 3.

Line 12: What determined the maximum depth?

*Author response:* The maximum depth was determined based on the 1 m drill barrel length. A drill extension is required to retrieve cores deeper than 1 m. We did not expect weathered ice to extend below 1 m depth, so we designed our field methods to remove 1 m cores using the standard drill barrel without any extensions. After drilling the first few cores, we realized the ice was weathered to at least 1 m depth, but owing to time limitations, and for consistency with the first few cores, we chose to drill each of the ten cores to 1 m depth. For additional context, we drilled two 1.8 m cores near camp (described in the text), but we were not able to systematically return to each core site and drill deeper.

Line 28: Be consistent with units; you used cm3 before.

Author response: The units have been changed, as requested.

Line 6: Delete 'these'

Author response: 'these' has been deleted, as requested

Line 7: Justify this given the difference in structure between the two layers.

*Author response:* Thank you for pointing out this important caveat to our study. We have added substantial discussion of structural differences between the two layers in the revised results Sect. 3.3 paragraphs 3 and 4.

Line 20: Nearest cryoconite hole? Nearest x cryoconite holes?

*Author response:* The nearest cryoconite hole within a 1 m radius of the posting. We have moved this description further up so the methods are clear to the reader right away.

Line 27: Another logical jump re: transition!

*Author response:* Thank you for pointing out the logical disconnect with respect to the unsaturated/saturated layer transition. We have removed the a priori characterization of the saturated/unsaturated transition in the methods and instead report the transition in the results where it belongs. We use the depth to water below the ice surface in cryoconite holes as an estimate of the depth to saturation, whereas the depth probe measurements are used as a qualitative characterization of the weathering crust structure. We hope this addresses the logical disconnect.

We are also working with a graphics specialist in our department to design a conceptual diagram

for the introduction that we hope will improve the visual communication of the weathering crust structure to the reader. The diagram will merge the conceptual diagram from Müller and Keeler, (1969) and the characteristic subsurface depth-density profile for weathering crust from LaChapelle, (1959):

**Figure 1:** Conceptual schematic of the weathering crust (Müller and Keeler, 1969) and the ideal subsurface ice density curve for the near-surface of an ablating glacier, adapted from La Chapelle (1959).

In the text, we have clarified the following:

- 1) The unsaturated depth is inferred from the depth to water in cryoconite holes
- 2) The depth probe measurements are used as a qualitative description of weathering crust structure with reference to the sub-surface density profile and conceptual schematic
- 3) The apparent transition from very low density to higher density material is reported in the results

Line 30: To what height? It would have been good to measure the water table at the drilled holes as well as at Cryoconite holes.

*Author response:* We agree these measurements would have been good to measure, but we think the cryoconite hole water levels provide a better (non-destructive, equilibrium) estimate of the water table height. We were also severely limited by time. For example, it wasn't clear how much time was required for the water levels in the drilled holes to equilibrate. We planned to measure the water levels after all other science priorities were completed, but returning to (and locating) all 100 drilled holes was infeasible. Instead, we use the refilling of the drilled holes as a proxy for water saturation, and use the depth to water in the cryoconite holes as a measure of the water table height.

**Page 8**

Line 8: Impermeable yes, but how continuous? Are these highly localized features?

*Author response:* As noted above, we have substantially revised our interpretation of the ice lenses to suggest they are highly localized structural features.

**Page 9**

Line 7: I think the value below snow surface would be a better one to quote.

Author response: We agree and have added this value, as requested.

Line 10: Again, to what level? Completely full?

*Author response:* As with the 1 m drilled holes (N=100), the water levels are used here as an indication of sub surface saturation, and the depth to water in cryoconite holes is used as an estimate of the depth to saturation.

Line 17: Did you see any evidence of flow in any of the holes?

*Author response:* We did not test for evidence of flow in the holes, other than our observations of refilling, which implies subsurface permeability and flow.

Line 25: Quantify 'often'

*Author response:* We agree this sentence is somewhat vague and unqualified and we have removed the sentence altogether. The observations were not systematic and hence should not be reported.

Line 29: Ok so the water table is  $\leftarrow$  20 cm below the surface yes? Which is consistent with your statement that the bottom layer, from 20cm down is saturated.

*Author response:* The water levels in cryoconite holes were on average 15 cm below the ice sheet surface, which is our best estimate of the water table height. We apologize for the confusion in the original manuscript, resulting from the depth probe measurements. We use the average 15 cm depth as an estimate of the depth to saturation.

**Page 10**

Line 29-30: This is speculative without seasonal data. If the aquifer is perennially saturated then this is not necessarily the case.

*Author response:* The average annual maximum snow depth in this region of Greenland is

---

## Author Comment (AC2) · 19 Oct 2017

Dear reviewer,

Thank you for the thorough and supportive comments. The suggestion to contextualize the study in terms of the seasonal cycle was particularly fruitful (and was shared by all reviewers). To address this, we have reported the seasonal and annual ablation recorded by a nearby automatic weather station, and added a comparison with regional meteorology reported in a recent publication that is highly relevant to our study (Tedstone et al., 2017). These data reveal that summer 2016 was characterized by conditions particularly favorable to weathering crust development, and therefore should help contextualize our findings. I provide a line by line reply to each comment below, and include revised figures at the end of the document. A revised manuscript will be uploaded following receipt of instructions from the handling editor. Thank you kindly,

Matthew Cooper

**Anonymous Referee #2**

**Received and published: 29 August 2017**

This paper presents findings from a field campaign on the Western margin of the Greenland ice sheet concerning the nature of the 'weathering crust' on the bare ice in the ablation zone. The paper provides measurements from shallow ice cores of ice density and corresponding porosity as well as water content, finding surprisingly low density ice down to at least 1 meter depth. It is pointed out that the presence of this weathering crust means that (subsurface) melt does not necessarily correspond to 'surface' lowering, as might be measured by satellite altimetry.

I think this is an interesting set of measurements, and is a valuable contribution to understanding the supraglacial hydrology of the Greenland ice sheet. Meltwater storage in the percolation zone of the Greenland ice sheet (in firn) has been well documented over the last few years, and this study suggests that a non-negligible amount of water storage and transport may occur beneath the apparent ice surface in the ablation zone too.

The paper is well written and the figures are mostly clear. I have one main comment and a number of minor comments, mostly seeking clarification.

**Main comment**

The measurements represent a snap-shot of the weathering crust in mid-July 2016 and it is not clear how this relates to the behaviour over the course of the melt season. I appreciate that the field campaign was limited in length so it may not be known how the crust itself evolves, but I think there needs to be more discussion of the setting for these measurements. In particular, what is the annual ablation rate in this region? At what stage of the melt season are these taken (i.e. roughly how much melting has already occurred here)? What is the ice temperature in this region? These are important issues in understanding how reflective these results are of wider spatial scales but also larger time-scales.

*Author response:* We agree and thank the reviewer for recognizing this important weakness in our original submission. As requested, we obtained daily measurements of ablation recorded by the KAN-M automatic weather station (AWS) during our field study. KAN-M is located ~8.3 km ENE of our field site at ~1270 m a.s.l. and is the most proximal AWS to our field site (1215 m a.s.l.). Sonic ranging data recorded by KAN-M indicate the maximum spring snow depth was ~50 cm and the snow disappearance date was ~21 June, which suggests the conditions we document developed over an ~21-day period between snow disappearance and the collection of the ice cores on 11-12 July. Following snow disappearance, AWS data indicate a cumulative ice surface lowering of ~55 cm prior to collection of the shallow ice cores on 11-12 July. The average ablation rate during this time was 2.65 cm d-1. The mean annual ablation rate at KAN-M is ~1.25 m a-1 (van As et al., 2017). These statistics are reported in the revised Sect 3.4 paragraphs 3-4.

To supplement these data we added a comparison with a recent publication that examines the relationship between regional meteorology and remotely sensed surface reflectance in the study region (Tedstone et al., 2017). Their analysis of regional meteorology from the Modèle Atmosphérique Régional (MAR) regional climate model (Fettweis et al., 2017) suggests ~50 cm average snow depth and mid-June snow disappearance date in summer 2016 (see Figure 3 in Tedstone et al., 2017), consistent with the AWS data we analyze. Further, their analysis suggests that meteorological conditions during summer 2016 were ideal conditions for weathering crust development. These include below average cloud cover and rainfall, and above average downward shortwave radiation (e.g. compare to Figures 1–4 in Tedstone et al., 2017). From these data and the AWS data, we conclude it is not surprising that a well developed weathering crust was present in the study area at the time of observation. We have added several discussion points throughout the manuscript to emphasize the seasonal context.

**Main comment (continued)**

In particular, I think there could be more discussion of how the inferred stored water thickness (15-22cm) relates to the amount of melt that has so far been produced this season (roughly what fraction of it is this), and how the depth of the porous ice compares with the amount that melts each year. Eg. are the ice lenses that form the result of recent refreezing (i.e. earlier the same year) or some earlier time?

Author response: As noted above, KAN-M data indicate ~55 cm of ice surface ablation prior to collection of the shallow ice cores on 11-12 July, equivalent to 49.5 cm water equivalent assuming solid ice density of ~900 kg m-3. The inferred stored water thickness (revised to 13-21 cm) is therefore ~26–42% of the cumulative seasonal melt.

The mean annual ablation rate is  $\sim 1.25$  m a-1 at KAN-M (van As et al., 2017). The shallow cores were limited to less than <1.1 m depth, though we observed weathered ice at depths >1.6 m, suggesting the weathering crust depth at the time of observation was greater than the mean annual ablation rate.

Regarding ice lenses, we have substantially revised and (we hope) improved our discussion of their nature. We do not think they are refrozen meltwater. For the crust to develop permeability, the ice must be temperate, which should prevent substantial refreezing (Schuster, 2001). Diurnal

refreezing could occur, but if so it seems unlikely that weathering would proceed at depths beneath refrozen ice lenses without also weathering the lenses, thus preventing a progressive stratigraphy from developing. The situation is different in firn, where the medium in inherently permeable, allowing meltwater to penetrate along preferential flow paths forming refrozen lens horizons. The revised discussion now reads:

"Previous analyses of weathering crusts have not reported this pattern of alternating clear, solid ice and fractured, granular ice (e.g. Hoffman et al., 2014; Müller and Keeler, 1969; Schuster, 2001). It seems likely these lenses are structural features, as refrozen meltwater in unlikely in a thermally temperate weathering crust (Schuster, 2001), and even less so weathered ice at depths below refrozen meltwater. Stratified distribution of grain size, crystal structure, bubbles, and/or impurities with depth (Hudleston, 2015) could each influence the rate of subsurface radiative heating and hence weathering (Brandt and Warren, 1993; Liston et al., 1999). Meltwater advection along micro seams or cracks may also promote differential weathering (Hambrey, 1977; Hambrey and Lawson, 2000), similar to joint block weathering of terrestrial lithology. Surface expression of differential weathering is certainly evident along the transect (Figure 1), and at broader scales in the region is associated with outcropping of impurities (Wientjes et al., 2012). The ice lenses, then, may represent structural resistance to weathering, and/or result from heterogeneity in subsurface flow paths that enhance differential "rotting" of subsurface ice (Nye, 1991). We would thus expect lenses to be localized features, which helps explain the lack of consistent stratigraphy among cores."

**Main comment (continued)**

There could then be some discussion of how the porous ice evolves over the course of the year. Presumably all the water freezes again in winter? In which case the ice is mostly solid at the start of the melt season (except perhaps the unsaturated surface layer, which seems to have a similar porosity to snow)? If the saturated subsurface water doesn't ever run-off, it may simply be changing the quantity of runoff rather than delaying it.

Author response: We interpret the literature to suggest the weathering crust is a seasonal phenomenon with no interannual carryover except in the case of stagnant ice (e.g. Fountain and Walder, 1998). On sub-seasonal timescales, the weathering crust can rapidly decay when the surface energy balance is dominated by longwave or turbulent heat fluxes that melt the surface, removing the crust and exposing solid ice. Common examples would include heavy rain, very warm winds, or warm cloudy conditions. However, it is conceivable that a deep weathering crust could persist to the end of the melt season if meteorological conditions allow. In this case, we would expect interstitial and surficial meltwater to refreeze following snowfall. Annual snowfall in this part of Greenland is typically <1.0 m (Tedstone et al. 2017) and therefore meltwater that does not drain from the crust likely refreezes during winter and/or at night. If snow cover is absent or ephemeral, the crust may sublimate over winter. However, interannual variability in these conditions is substantial. Though we agree the annual progression of the porous ice is a logical next step for research on the topic, we prefer not to comment on interannual carryover without a physical model to support the analysis. That said, to touch on this topic we note in the revised that the mean annual ablation rate is ~1.25 m a-1 at KAN-M. Given that we document weathered ice at depths >1.5 m it is conceivable that near-surface, low density ice persists on interannual timescales.

**Specific comments**

Why are the findings frequently referred to as 'preliminary'? What are they preliminary to? If they are really preliminary, it begs the question why they are being published. I'd suggest that if the authors think the results are worth publishing they should not refer to them as preliminary (which does not preclude doing more work on the topic).

*Author response:* We appreciate this point and have removed our use of 'preliminary' throughout the text. Our original intent was to highlight the relatively immature status of the research topic in general, and more so its application in Greenland, not our specific findings.

**Page 1**

Line 29: Why does the routing of surface water to the ocean 'reinforce concerns' about contribution to global sea level rise? Isn't such melting part of the 'normal' operating cycle of an ice sheet?

Author response: We have removed the statement, as requested. Our original intent was to highlight the "efficient drainage" hypothesis, which is the assumption that ablation zone meltwater is transported rapidly, in its entirety, to surrounding oceans. To develop this idea more clearly, the revised introduction references works that demonstrate substantial time lags and possible meltwater retention in the ablation zone as motivation for the study of ablation zone hydrologic processes and near-surface porous ice

The revised introduction reads as follows:

"Each summer a vast hydrologic network of lakes and rivers forms on the surface of the southwest Greenland Ice Sheet ablation zone in response to surface melting (Chu, 2014; Smith et al., 2015). Evidence suggests that most or all of this water is efficiently delivered via supraglacial rivers to moulins, crevasses, and, ultimately, to proglacial rivers and surrounding oceans (van As et al., 2017a; Lindbäck et al., 2015; Rennermalm et al., 2013; Smith et al., 2015). The assumption of efficient meltwater delivery is reflected in regional climate and surface mass balance models of Greenland that instantaneously credit ablation zone surface runoff to the ocean with no physical representation of hydrologic processes or meltwater runoff retention taking place on the ablation zone bare ice surface (Smith et al., 2015). On daily to monthly timescales, however, field studies and satellite remote sensing have found evidence of substantial meltwater runoff delays in the Greenland Ice Sheet ablation zone (van As et al., 2017a; Karlstrom and Yang, 2016; Koenig et al., 2015; Lindbäck et al., 2015; Overeem et al., 2015; Rennermalm et al., 2013; Smith et al., 2015). Similar runoff delays are observed in supraglacial environments elsewhere (Karlstrom et al., 2014; Munro, 1990), owing to the presence of a degraded, porous "weathering crust" (Müller and Keeler, 1969) on the bare ice surface of glaciers and ice sheets that stores meltwater, delaying its delivery to supraglacial channels via porous subsurface flow (Irvine-Fynn et al., 2011; Karlstrom et al., 2014; Munro, 2011). The porous weathering crust may also provide a substrate for internal and/or surficial refreezing of meltwater (Hoffman et al., 2014; Paterson, 1972; Willis et al., 2002), similar to meltwater

transport, storage, and refreezing in snow and firn (Cox et al., 2015; Forster et al., 2014; Harper et al., 2012; Machguth et al., 2016). The presence of weathering crust in the Greenland Ice Sheet bare ice ablation zone, however, has gone largely undocumented, and little is known about the effect of weathering crust meltwater storage on hydrologic efficiency in the bare ice ablation zone, where >85% of Greenland ice sheet meltwater runoff is generated (Machguth et al., 2016)."

Section 2.1: It is not clear from this description how liquid water in the core is dealt with. Is it allowed to drain out? Presumably there is still quite a lot of water trapped in the core samples (due to capillary forces) and this contributes to the measured mass?

*Author response:* Thank you for this important observation. We did not explain this adequately in the manuscript. The drill barrel was held vertically and allowed to drain when cores were removed from the boreholes prior to weighing. After removal from the borehole, the drill was laid at a slight angle and the core was carefully removed from the drill barrel and immediately analyzed, providing additional time for drainage. Though our aim was to drain the cores completely, it is correct that some water remained owing to capillary forces. It is also possible that some non-capillary water remained owing to incomplete free-drainage. These water retention errors would result in overestimated ice density.

In adding a more thorough discussion of this issue to the methods section, we also provide more detail about the measurement uncertainty noted in the original manuscript. Namely, the natural breaks of the ice cores were irregular and some material was inevitably lost near the ends of the core segments. The 10% error estimate we provided in the original manuscript was meant to account primarily for this loss of material at the irregular ends of the ice core segments, which would tend to result in underestimated ice density.

To summarize, there are two primary sources of error we expect are important 1) ice core volume measurement error owing to loss of material near the irregular ends of the individual ice core segments, and 2) interstitial meltwater retention errors owing to capillary water retention and incomplete free water drainage. The volumetric error would tend to result in underestimated ice density, the water retention in overestimated density. Hence, the two would tend to cancel, though to an unknown extent as both errors are poorly constrained.

In the revised methods, we describe these error sources in greater detail, and we cite estimates of temperate ice water content ranging from 0-9%, though most estimates (15 of 18) are <3.4%, including all estimates made from in situ calorimetric methods (Pettersson et al., 2004). The uppermost 9% estimate is thus well within our  $\pm 20\%$  specific storage uncertainty estimate. We think this is sufficiently conservative without giving undue confidence to either the measurements or the error estimate.

It may also be worth noting this same issue is present for studies of firn density. For such studies, a physical model can be used to establish a theoretical dry-firn density that can be compared with in situ measured density to estimate liquid and/or refrozen water content. While subsurface weathering crust density in Antarctica has been modeled (Hoffman et al., 2014), such an exercise is well beyond the scope of this paper.

Finally, we corrected the error estimate by adding a physical constraint that density cannot exceed solid ice density (917 kg m-3) and effective porosity cannot exceed total porosity (1 –  $\rho_m/\rho_{ice}$ ). These issues are presented in the revised methods Sect. 2.1 paragraph 3.

**Page 4**

Line 9: This is a bit awkward wording, since this statement presumably assumes that the density is uniform (independent of depth). Perhaps better just to say that the geometry of the sampler means that the near-surface ice is disproportionately weighted in this average, rather than quantifying the 'center of mass'.

Author response: As requested, we have removed the 'center of mass' statistic and replaced with the following brief description " $\dots$  the density measurements may be more representative of the uppermost ~6 cm of material because of the shape of the sampler  $\dots$ "

Section 2.3: There is some confusion here about the 'unsaturated weathering crust depth', and how it relates to how penetrable the ice is. Reading further, it seems that the 'water table' (which I would interpret as the unsaturated depth) roughly coincides with a change in the strength of the ice that is presumably what the depth probe is detecting. It does not seem obvious to me why these two surfaces (the impenetrable ice surface and the water table) should happen to coincide - perhaps the presence of air in the pores above this allows the surface ice to 'rot' more rapidly. Or perhaps the permeability of the upper layer is sufficiently large that water in this layer readily runs off horizontally keeping it unsaturated. Perhaps the qualitative description of the surface given on page 9, line 20, could be moved forward to the method section to help explain these issues. In any case it would help to be clearer precisely what is meant by unsaturated - does this mean there is no liquid water, only residually-trapped water, or that water does not fill the pore space (all of which are different)?

Author response: We apologize for the confusing presentation of the steel rod measurements and the unsaturated depth in the original manuscript. In the revised, we have removed the a priori characterization of the saturated/unsaturated transition in the methods and instead report the transition in the results, where it belongs. The unsaturated depth is estimated from the depth to water below the ice surface in cryoconite holes, whereas the depth probe measurements are used as a qualitative characterization of the weathering crust structure.

To contextualize the depth probe measurements and the weathering crust structure for the reader, we are working with a graphics specialist in our department to create a diagram for the introduction that merges the weathering crust conceptual diagram of Müller and Keeler, (1969) with the characteristic depth-density decay curve from LaChapelle, (1959). A crude representation is shown below for reference as the diagram is in production. Near surface weathered ice tends to exhibit a characteristic increase in density from a very low-density surface layer to a higher density subsurface that approaches solid ice density. The very low density surface layer is demonstrated by the coarse material above the water table in the conceptual diagram of Müller and Keeler, (1969) (Figure 1, left). This is the material the depth probe penetrates, which we suggest may be indicative of the "shoulder" on the density decay curve where density increase non-linearly (Figure 1, right). We hope the diagram will clarify the depth-variable nature of the crust for the reader and provide context for the depth probe measurements.

**Figure 1:** Conceptual schematic of the weathering crust (Müller and Keeler, 1969) and the ideal subsurface ice density curve for the near-surface of an ablating glacier, adapted from La Chapelle (1959).

**Page 6**

Line 5: Why is refrozen meltwater included as water storage? If it has refrozen it is ice again and should be thought of as storage (it requires melting again - with the associated energy implications - before it could run off).

*Author response:* We have removed refrozen meltwater from the definition of storage, as requested.

**Page 6**

Line 19: Is the 'potential' liquid storage capacity not just the effective porosity multiplied by depth and total area (i.e. including the currently unsaturated pore space too)?

*Author response:* Our use of the word "potential" was incorrect and misleading. Your characterization is correct but we present the actual (instantaneous) specific storage estimated from the shallow cores, and then scale that to the study catchment by multiplying the average storage depth by the total area to estimate a storage volume.

L19 has been revised to read "Finally, for illustrative purposes we scale our storage estimate to the study catchment by multiplying the mean  $S_P$  estimated from the shallow ice cores by the bare ice surface area of the study catchment ...".

**Page 8**

Line 7: There seems to be some subjectivity involved here. Why is estimating the value wrong in one direction deemed 'not problematic'? If the densities were measured including the ice lenses, would it not make sense to use the volume including the ice lenses when converting to water content using the effective porosity? Or otherwise use a solid ice density of the ice lens to infer the density of the non-ice lens part of each segment?

*Author response:* Thank you for this important critique. Upon consideration, we agree it makes sense to use the volume including the ice lenses when converting to water content since they are included in the density. This was an oversight on our part. We have removed the various references to the ice lens density bias, which we think will remove unnecessary confusion for the reader. Moreover, given our interpretation of the lenses as structural features, it makes sense to include their volume in the storage estimate.

**Page 10**

Line 4: The drill did not go below 1.8m for fear of freezing. Did you make any measurements of temperature in the porous crust? It would be helpful to know if the ice is all at the bulk melting temperature or if it goes below this at depth.

Author response: We absolutely agree that temperature measurements would be invaluable. The reason we note the "risk of freezing" is to suggest, albeit indirectly, that the ice may be sub-freezing at this depth. Unfortunately, measuring subsurface ice temperature is very difficult and error prone unless done with considerable care, and we were not able to undertake such measurements. The best we can do is to suggest the ice may have been freezing based on our observations of the ice core drill seizing up. Co-author Miege has extensive experience drilling firn cores and he suggested based on his observations of the drill behavior at these depths that the ice was freezing.

Line 20: What is meant by a storage 'rate'? I could not work out what this number means.

*Author response:* This should have been referred to as 'specific storage rate' and is the specific storage (i.e. storage depth) divided by the time over which the meltwater storage accumulated. In either case, we have removed the comparison with these rates because they were estimated using water budgets and reviewer 3 objected to the comparison with our core-density method.

**Page 11**

Line 15: The 'lower and upper mean' is a strange concept; perhaps the 'mean lower and upper values' would be better wording.

*Author response:* As requested (with slight modification), we have replaced 'lower and upper mean Sp' with 'lower and upper estimates of Sp'.

Line 21 (and conclusions): Why are the results not considered representative of the rest of the ablation zone? I understand the desire for caution given that this is only one location, but without other evidence (perhaps you have it?) wouldn't the default assumption be that the results do apply more widely? What do you think is special about your field site that means the results would not apply more widely? Perhaps you could just say 'We do not know whether these findings represent typical conditions....' rather than 'not proposing' it.

*Author response:* We do not have any evidence that this location was unique. However, weathering crust growth and decay is strongly controlled by local meteorology and therefore can be highly variable over short distance and time. To our knowledge, there are no studies of seasonal weathering crust formation in Greenland, but subsurface melting in the study region has

been modeled and shown to depend on snow cover, which varies with elevation (van den Broeke et al., 2008). Lacking spatial data, we were trying to be cautious, but upon consideration we agree there is no need to over emphasize this speculative (albeit cautionary) assumption.

As requested, we have removed the statement 'we do not propose' and replaced with 'may not be representative'.

**Updated figures/table:**

---

## Author Comment (AC3) · 19 Oct 2017

Dear reviewer,

First, thank you for your thorough and tremendously helpful review. It is clear you invested considerable time into this review and your comments and suggestions have greatly improved the quality of the manuscript. We hope we have adequately addressed the various suggestions and concerns raised throughout. I provide a line by line reply to each comment below, and include revised figures at the end of the document. A revised manuscript will be uploaded following receipt of instructions from the handling editor. Thank you kindly,

Matthew Cooper

**Anonymous Referee #3**

Received and published: 8 September 2017

This paper reports on the findings from a field campaign on the ablation zone in southwestern Greenland. The focus of the paper is the so-called "weathering crust" that characterises glacier and ice sheet surfaces, and its potential hydrological storage role. The authors use a set of shallow ice cores (n=10) to describe the variability in near-surface ice density over depths of < 2 m. From these observations, the authors explore an effective porosity of the near-surface ice, and examine a potential water storage based on observations of a water table evident within the weathering crust. A specific storage of ~0.2 m is derived, suggesting that at the time of observations a water volume equivalent to 1 hours' worth of discharge from the local supraglacial catchment was essentially stored within the weathering crust.

The findings are a useful demonstration that this weathering crust exists on the Greenland Ice Sheet, and provides a sensible trigger for future work assessing the supraglacial drainage system and its functionality. Although some recent focus in Greenland has included the firn aquifer at higher elevations, it is an interesting insight to an overlooked hydrology of the ablating bare ice sector of the ice sheet. The growing recognition of the supraglacial realm as an ecosystem, and the potential importance of water storage on biogeochemical cycling at the ice sheet surface ensures this is a timely contribution and serves as a useful benchmark in this type of work.

Overall, the paper is well written, sensibly referenced, and the figures are clear. The methods are intelligible and could be repeated, and the calculations utilised are sensible within the limits of the data available/presented.

However, major comments would include:

A stronger description of how the weathering crust forms, and the subtlety of its growth and decay would be beneficial both in the introduction and in the later discussion. Specifically, would you expect a deep weathering crust at the time of your observations? Does the timing of snow melt, dominance of shortwave radiation, absence of rainfall give reason to consider the weathering crust (and ice lenses) you describe?

*Author response:* We have added a stronger description of how the weathering crust forms in the introduction, as requested. We draw attention to the depth dependency of radiative heating and the depth-density profile from LaChapelle (1959). We are working with a digital graphics specialist in our department to create a diagram that merges elements of the weathering crust conceptual schematic from Müller and Keeler, (1969) with the depth-density curve that we think will greatly improve the weathering crust description.

To address the temporal context of our findings, we obtained daily measurements of ablation recorded by the KAN-M automatic weather station (AWS) during our field study. KAN-M is located ~8.3 km ENE of our field site at ~1270 m a.s.l. and is the most proximal AWS to our field site (1215 m a.s.l.). Sonic ranging data recorded by KAN-M indicate the maximum spring snow depth was ~50 cm and the snow disappearance date was ~21 June, which suggests the conditions we document developed over an ~21-day period between snow disappearance and the collection of the ice cores on 11-12 July. Following snow disappearance, AWS data indicate a cumulative ice surface lowering of ~55 cm prior to collection of the shallow ice cores on 11-12 July. The average ablation rate during this time was 2.65 cm d-1. The mean annual ablation rate at KAN-M is ~1.25 m a-1 (van As et al., 2017). These statistics are reported in the revised Sect 3.4 paragraphs 3-4.

To supplement these data we added a comparison with a recent publication that examines the relationship between regional meteorology and remotely sensed surface reflectance in the study region (Tedstone et al., 2017). Their analysis of regional meteorology from the Modèle Atmosphérique Régional (MAR) regional climate model (Fettweis et al., 2017) suggests ~50 cm average snow depth and mid-June snow disappearance date in summer 2016 (see Figure 3 in Tedstone et al., 2017), consistent with the AWS data we analyze. Further, their analysis suggests that meteorological conditions during summer 2016 were ideal conditions for weathering crust development. These include below average cloud cover and rainfall, and above average downward shortwave radiation (e.g. compare to Figures 1–4 in Tedstone et al., 2017). From these data and the AWS data, we conclude it is not surprising that a well developed weathering crust was present in the study area at the time of observation. We have added several discussion points throughout the manuscript to emphasize the seasonal context.

**Major comments (continued):**

Is this a glacier ice weathering crust or one that perhaps is superimposed ice derived from snow and refrozen lenses forming therein? If this is glacier ice, then you should at least mention ice structure in addition to refreezing processes (particularly given the evidence of marked structure in the locality).

Author response: It was a glacier ice weathering crust, as data from KAN-M and from Tedstone et al. (2017) suggest the maximum snow depth in the region was ~0.5 m and had completely melted by 21 June. We observed a few remnant snow patches in the field. The snow was heavily metamorphosed into uniform spherical grains and was easily distinguished from the pervasive coarse bubbly weathering crust ice. We do, however, think superimposed snowmelt ice is an important part of the annual surface ablation cycle in the study area and probably contributes to the initial formation and structure of the crust, though we do not have data to support this presently and the influence of superimposed snowmelt on our results is likely negligible owing to

the  $\sim$ 55 cm of ice surface lowering. Regarding ice structure, we thank you for encouraging us to explore this more thoroughly and we have added considerable discussion of ice structure in the revised results, copied below in response to specific comments.

**Major comments (continued):**

A clearer emphasis regarding the results being a snapshot which reveals something about the supraglacial hydrology of the Greenland Ice Sheet could be beneficial. In your discussion, albeit subjectively, are you able to comment on the likelihood of greater or less storage to be seen at other times of the summer, is this a seasonally progressive hydrological feature or is the observation just that, a discrete observation – there are climate records for the locality which might allow some extrapolation of these ideas.

Appreciably, it is not possible to go beyond perhaps a statement on this, given the limitations of the data set, but it would be helpful.

*Author response:* As noted above, we have added a comparison with seasonal climate records from the nearby KAN-M AWS and with Tedstone et al. (2017) which together demonstrate climate conditions prior to the field study were favorable for weathering crust development. Regarding the seasonal progression, we report the annual ablation rate is ~1.25 m a-1, therefore it is conceivable there could be some interannual persistence given we document a >1.5 m deep crust. We note this is dependent on meteorological conditions favorable to crust growth through the end of summer. Additionally, we emphasize in several places, including a new closing paragraph in the discussion, the transient nature of the crust growth and removal. We are hesitant to comment further without field data or a physical model of crust development.

**Major comments (continued):**

A slightly strengthened assessment of the uncertainties would be important I feel – this should include an assessment of the water content of the cores themselves as it is not clear if all interstitial water was removed prior to evaluating mass for density estimates. Temperate ice can have interstitial water content of up to ~10% (see Petterson et al., 2004, JGR), and certainly the saturated lower-most ice in the developing weathering crust may exhibit such water content if this is a seasonally temperate ice layer. Can you perhaps try to assess uncertainties associated with this water content, and the resultant impact on other estimates presented here. The 10% and 10% quoted seemed a little arbitrary when slightly more detailed and thorough approaches could have been taken. Furthermore, can you account for the ablation of the ice surface if cores were not all taken on one day – can the core profile figure be corrected/adjusted for surface ablation – making crude assumption that ablation over transect broadly similar, or using a point estimate from the energy balance? Correcting for the 7 days ablation period might be informative and aid inferences – such that for example, ice lenses may be better aligned perhaps if representative of refreezing events or local thermal conditions.

Author response: Regarding ablation, the cores were collected over a period of two days (11-12 July, noted in the revised). The ablation rate was  $\sim 2.6 \text{ mm d}^{-1}$  during the field campaign. As such we have not adjusted the cores to account for this.

Regarding water content, thank you for raising this important point that was not adequately addressed in the initial manuscript. The drill barrel was held vertically and allowed to drain when cores were removed from the boreholes prior to weighing. After removal from the borehole, the drill was laid at a slight angle and the core was carefully removed from the drill barrel and immediately analyzed, providing additional time for drainage. Though our aim was to drain the cores completely, it is correct that some water remained owing to capillary forces. It is also possible that some non-capillary water remained owing to incomplete free-drainage. These water retention errors would result in overestimated ice density.

In adding a more thorough discussion of this issue to the methods section, we also provide more detail about the measurement uncertainty noted in the original manuscript. Namely, the natural breaks of the ice cores were irregular and some material was inevitably lost near the ends of the core segments. The 10% error estimate we provided in the original manuscript was meant to account primarily for this loss of material at the irregular ends of the ice core segments, which would to tend to result in underestimated ice density.

To summarize, there are two primary sources of error we expect are important 1) ice core volume measurement error owing to loss of material near the irregular ends of the individual ice core segments, and 2) interstitial meltwater retention errors owing to capillary water retention and incomplete free water drainage. The volumetric error would tend to result in underestimated ice density, the water retention in overestimated density. Hence, the two would tend to cancel, though to an unknown extent as both errors are poorly constrained.

In the revised methods, we describe these error sources in greater detail, and as requested, we cite estimates of temperate ice water content ranging from 0-9%, though most estimates (15 of 18) are <3.4%, including all estimates made from in situ calorimetric methods (Pettersson et al., 2004). We find no estimate of depth-dependent water content for near-surface <2m deep ice, hence a uniform water content error seems sensible, and the uppermost 9% estimate is well within our  $\pm 20\%$  error estimate for specific storage. We think this is sufficiently conservative without giving undue confidence to either the measurements or the error estimate. Perhaps more important is to interpret the error and provide expectations. For example, we think the storage is more likely to have been underestimated owing to interstitial meltwater retention than overestimated owing to poor volumetric measurements, but we don't have a rigorous way to demonstrate this.

It may also be worth noting this same issue is present for studies of firn density. For such studies, a physical model can be used to establish a theoretical dry-firn density that can be compared with in situ measured density to estimate liquid and/or refrozen water content. While subsurface weathering crust density in Antarctica has been modeled (Hoffman et al., 2014), such an exercise is well beyond the scope of this paper.

Finally, to improve the error estimate we have added a physical constraint that density cannot exceed solid ice density (917 kg m-3) and effective porosity cannot exceed total porosity (1 –  $\rho_m/\rho_{ice}$ ). These issues are presented in the revised methods Sect. 2.1 paragraph 3.

**Major comments (continued):**

Could more analysis of the data presented in Figure 6 be made available here? There are opportunities to examine patterns over elevation (small range though that is) and in relation to the detrended surface and 'roughness'. Similarly, it would be interesting to see if there is from the profiles (e.g. what are the patterns of phi-eff at, say, 33cm and 87cm depth, where it looks weathering crust (not ice lens) data is available across all cores – assuming these positions remain if adjusted for ablation over the 7-day sampling) – is there anything to be gained from a slightly deeper examination of the density and porosity data over depth and along the transect?

*Author response:* Regarding spatial variability along the transect, we tested for statistically significant linear relationships between distance/elevation and 1) depth of cryoconite holes, and 2) depth to water in cryoconite holes. No relationship was found for depth to water though there was a slight trend toward shallower holes (-0.014 cm m-1, p<0.004). We report these findings in the revised Sect. 3.3. The same trends were found for elevation owing to the gradual increase in elevation along the transect, but the elevation distribution was strongly skewed toward higher values. The lack of any trend in depth to water with distance confirms that cryoconite water levels generally mirrored the small-scale roughness, which was noted in the original manuscript but was difficult to see in Figure 6. In the revised, the grey filled area has been removed to improve interpretation of Figure 6.

Regarding density and porosity depth-variability, we have reframed portions of our analysis in terms of the theoretical depth-density curve presented in LaChapelle (1959), as suggested. We appreciate this suggestion and we hope it has improved the framing of the depth-density results. As requested further down, we gap-filled the missing data in Figure 3 so the depth-density profiles are complete, which provides a better comparison with the depth-density curve of LaChapelle.

Regarding ice lens stratigraphy, we do not think they are controlled by temporal meteorological variability. As suggested, we think they are structural features possibly controlled by stratification in ice grain size, crystal structure, bubble size and/or content, or impurities, each of which could influence radiative heating. We also speculate that meltwater advection along micro seams or cracks may promote differential weathering, similar to joint block weathering of terrestrial lithology. We hope this general discussion of ice structure provides guidance to the reader, but we were not able to find patterns in density or porosity across cores at specific depths. This may not be surprising. For example, ice lens stratigraphy in firn is highly variable and discontinuous over spatial scales as short as 1.5 m (Brown et al., 2011; Machguth et al., 2016). Each of these would suggest lenses are highly local features, helping to explain the lack of consistent stratigraphy among cores.

The new discussion of the ice lenses is provided further down in response to specific comments.

**Major comments (continued):**

In places the writing style became less clear, or seemed to have a slightly reduced scientific quality. Similarly, a couple of key references seemed to be absent or choices of references seems a little misplaced, while in other places there was a proliferation of sources when perhaps just

one or two examples would suffice. Some further editing and subtle revisions would likely be beneficial, to strengthen this paper, in addition to perhaps examining a few more relevant publications that would be of help in supporting these results and findings and their significance.

*Author response:* Thank you for these suggestions. We carefully reviewed the citations and found several instances where, we agree, the chosen citations were misplaced, especially in the introduction. To address this, we have carefully (we hope) separated key citations that refer to theoretical ice permeability studies (Lliboutry, 1996; Mader, 1992; Nye, 1991; Nye and Frank, 1973) from those that deal with in-situ glacier ice studies (Cook et al., 2016; Fountain and Walder, 1998; Irvine-Fynn, 2008; Müller and Keeler, 1969), from those that deal with subsurface radiative properties of glacier ice (Brandt and Warren, 1993; Liston et al., 1999; Liston and Winther, 2005), and finally, we made effort to support statements focused strictly on cryoconite holes with relevant publications (Boggild et al., 2010; Cook et al., 2016) from those that deal with adal with microbial communities in glacier ice more generally (Cook et al., 2012; Irvine-Fynn and Edwards, 2014). We hope the writing style has been improved.

**Minor comments and suggestions (some touching on points above) would include:**

Line 1: Suggest hyphenate "near-surface" throughout. (There are some variations, e.g. P3 L13 and L15).

Author response: As requested, 'near-surface' has been hyphenated throughout.

Line 2: "Greenland Ice Sheet", as used throughout the manuscript. It is refreshing to see authors correctly use the appropriate capitalisation for proper nouns (it shouldn't be the Greenland ice sheet, given it is a specific location and entity) and at times I wish publishers would adhere to grammatical correctness – but that is another discussion altogether. L2: "Meltwater storage in low-density near-surface bare ice in the ablation zone of the Greenland Ice Sheet" might read a little better perhaps?

*Author response:* We agree the title reads better (with slight modification) as "Meltwater storage in low-density near-surface bare ice in the Greenland Ice Sheet ablation zone"

Line 16: Suggest referring to this as "specific storage".

*Author response:* "liquid meltwater storage" has been changed to "specific storage" here and throughout the manuscript, as requested.

Line 17: Clarify the water level is depth from the surface or from the base of the auger holes, and is "recharge" a more preferable term than infilling (given this is a hydrology paper).

*Author response:* Here, water level is the depth from base of the holes (i.e. height of water in holes). We report these water levels as they are suggestive of water storage in cryoconite holes. Elsewhere, we report water levels relative to the surface. "Infilling" is replaced with "refilling" here and throughout the manuscript, as requested.

Line 18: "These observations are consistent. . ." given you present results and discuss them. Analysis might be provisional with clear directions to follow, but don't negate the potential utility of these observations.

*Author response:* Thank you for this suggestion, we agree and have removed "Though preliminary ... ".

Line 21: "supraglacial catchment"

Author response: "catchment" has been changed to "supraglacial catchment", as requested.

Line 25: A longer opening paragraph would be stronger as an opening. Can there be a clearer link from mass balance or ablation to runoff models for Greenland, and the assumptions regarding the efficient delivery. The sea level aspect here seems misplaced, as the study looks at in-season delays or reductions in discharge. Surely, noting the assumed efficient drainage is now being examined more closely with reference to the firn aquifer and so on would allow for a stronger introduction paragraph here.

*Author response:* We have lengthened the opening paragraph, as requested. To clarify the efficient delivery to surrounding oceans statement, we added reference to several works that demonstrate substantial time lags and possible meltwater retention in the ablation zone as motivation for the study of ablation zone hydrologic processes and near-surface porous ice. We first note the evidence for, and assumption of efficient runoff delivery, then note evidence of time lags and possible retention. Finally, we suggest the weathering crust as a possible mechanism for runoff delays based on evidence from other supraglacial environments. Weathering crust formation, ablation, and density are then discussed in the second paragraph.

The revised introduction reads as follows:

"Each summer a vast hydrologic network of lakes and rivers forms on the surface of the southwest Greenland Ice Sheet ablation zone in response to surface melting (Chu, 2014; Smith et al., 2015). Evidence suggests that most or all of this water is efficiently delivered via supraglacial rivers to moulins, crevasses, and, ultimately, to proglacial rivers and surrounding oceans (van As et al., 2017a; Lindbäck et al., 2015; Rennermalm et al., 2013; Smith et al., 2015). The assumption of efficient meltwater delivery is reflected in regional climate and surface mass balance models of Greenland that instantaneously credit ablation zone surface runoff to the ocean with no physical representation of hydrologic processes or meltwater runoff retention taking place on the ablation zone bare ice surface (Smith et al., 2015). On daily to monthly timescales, however, field studies and satellite remote sensing have found evidence of substantial meltwater runoff delays in the Greenland Ice Sheet ablation zone (van As et al., 2017a; Karlstrom and Yang, 2016; Koenig et al., 2015; Lindbäck et al., 2015; Overeem et al., 2015; Rennermalm et al., 2013; Smith et al., 2015). Similar runoff delays are observed in supraglacial environments elsewhere (Karlstrom et al., 2014; Munro, 1990), owing to the presence of a degraded, porous "weathering crust" (Müller and Keeler, 1969) on the bare ice surface of glaciers and ice sheets that stores meltwater, delaying its delivery to supraglacial channels via porous subsurface flow (Irvine-Fynn et al., 2011; Karlstrom et al., 2014; Munro, 2011). The porous weathering crust may also provide a substrate for internal and/or surficial refreezing of

meltwater (Hoffman et al., 2014; Paterson, 1972; Willis et al., 2002), similar to meltwater transport, storage, and refreezing in snow and firn (Cox et al., 2015; Forster et al., 2014; Harper et al., 2012; Machguth et al., 2016). The presence of weathering crust in the Greenland Ice Sheet bare ice ablation zone, however, has gone largely undocumented, and little is known about the effect of weathering crust meltwater storage on hydrologic efficiency in the bare ice ablation zone, where >85% of Greenland ice sheet meltwater runoff is generated (Machguth et al., 2016)."

Line 28: what is a "terminal moulin"? And not all runoff goes to moulins – there are supraglacial routes to proglacial regions, and lakes and crevasses. Suggest more circumspect and/clarified text here.

*Author response:* A "terminal moulin" is a moulin that exists at the terminal drainage point of a supraglacial catchment. Analysis of sub-meter resolution WorldView-1/2 satellite imagery suggests that every supraglacial river in the study region drains to a moulin before reaching the ice edge (Smith et al., 2015). To avoid confusion, we have dropped the word "terminal".

**Page 2**

Line 5: Cite Muller and Keeler (1969) for the introduction of the term "weathering crust". Might an additional diagram be helpful here to conceptually illustrate what you are focused on here for the less familiar reader?

*Author response:* As requested, Muller and Keeler (1969) have been cited following the first mention of 'weathering crust'. We agree a diagram would be useful and are currently working with a digital graphics specialist at our university to produce an updated diagram.

Line 6: Fountain and Walder (1998) also note minimal delay to supraglacial runoff and so text and citation here, given phrasing, might be slightly inappropriate. Suggest check the source again.

Author response: Fountain and Walder (1998) has been removed, as requested.

Line 9: What is a "seasonally temperate glacier"? Poor terminology, please revise. Seasonally temperate surface ice perhaps, but glacier thermal regime is a very different thing.

Author response: Thank you for this important correction. We were referring to the seasonally temperate near-surface, not the broad glacier thermal regime. We replaced our various usages of "temperate", "polythermal", etc. with "thermally transient ice surface" as in Irvine-Fynn et al., (2011). Elsewhere, we have removed mention of thermal regime altogether and replaced with "supraglacial environment" etc.

Line 11: Surely the depth is ice-type dependent, and stating "~2m" is not strictly correct. Consider rephrasing (see Cook et al., 2016). It was also surprising that at no point is Munro (1990, AAAR) cited here, a source confirming the subsurface melt and bulk ice density variations leading to uncertainty in runoff volumes at Peyto Glacier. Suggest consideration of this source, especially with regard L19. *Author response:* We agree "~2m" is unnecessarily specific and we have replaced with "typically <2 m thick, owing to the exponential attenuation of radiation penetration with depth (Brandt and Warren, 1993; Irvine-Fynn and Edwards, 2014; Müller and Keeler, 1969)."

Regarding Munro (1990), thank you for suggesting this highly relevant reference. Though we cited Munro (2011) in the original manuscript, this additional citation strengthens the literature review for the reader. We have added this citation to L19, as requested.

Line 17: Doesn't lateral meltwater motion result in sensible and frictional heat transfers, contributing to further removal of ice mass. Also suggest clarification over the vertical extension of the weathering crust, and how this influences mass for any given vertical position. The process described by Muller and Keeler (1969) is a little more complex than perhaps is given credence here, and perhaps a more careful description could be afforded. See also Cook et al., 2016.

*Author response:* Regarding sensible and frictional heat transfers, we have added this to our interpretation of the weathering crust structure in Sect. 3.1, where we suggest meltwater advection along micro seams and cracks may enhance subsurface weathering.

To address weathering crust vertical structure, we reference the characteristic depth-density profile from LaChapelle, (1959), and note the exponential attenuation of solar radiation in the upper few meters of ice (e.g. Brandt and Warren, 1993). We are working with a graphics specialist in our department to create a diagram that merges the weathering crust conceptual diagram (Müller and Keeler, 1969) with the characteristic depth-density profile (LaChapelle, 1959) that we hope will further clarify the depth-variable nature of the crust.

Line 22: The opening of this paragraph is not entirely appropriate, the structure and the content seems slightly superficial and/or repetitive (e.g. mention of delay in runoff is already in L6). Suggest revisiting this text through to L26 and P3.

*Author response:* Considering this and the next several comments, we have substantially revised the introduction. The revised introduction has the following structure:

- 1. Broad motivation ablation zone hydrology is poorly represented in models, particularly with respect to seasonal runoff delays and retention
- 2. Definition / description of weathering crust formation and relevance to supraglacial hydrology and surface mass balance
- 3. Description of broader relevance with respect to microbial habitat and albedo
- 4. Justification and purpose for this study in Greenland

Line 22: Is subsurface melting in Antarctic contexts the same as the definition provided of weathering crusts on "temperate ice" (see L9)? Strongly advise some differentiation between subsurface melting and weathering crust terminology. This sentence could be removed at no loss to the paper.

Author response: We agree and have removed the reference to Antarctic contexts, as requested.

Line 25: Slightly unconvincing use of the literature here: some references focus on cryoconite

holes, others on the weathering crust as a habitat. Recommend revisiting, with perhaps consideration of recent messages regarding glacier ecohydrology (e.g. Dubnick et al., 2017a,b, JGR and Hydro Proc.; Hotaling et al., 2017, Env Mic.; Milner et al., 2017, PNAS). Yes, the weathering crust is a substrate for cryoconite holes (see Muller and Keeler's 1969 diagram), but the focus here should be the hydrological aspects and for example disturbance to cryoconite holes that might influence their ecology (Edwards et al., 2011, ISME J; Mieczan et al., 2013, PPR) or distribution (e.g. Hodson et al., 2007, JGR). Then develop the 'undescribed in Greenland to date' message and the guide of what is to follow (subsequent paragraph). If you do touch on the biogeochemical cycling aspects, it might be helpful to touch on these again in the discussion section.

*Author response:* In the revised introduction, we discuss weathering crust relevance to microbial habitat in a standalone paragraph. As suggested, we emphasize the relevance of weathering crust hydrology (Cook et al., 2016) as a control on glacier ecology via 1) cryoconite hole distribution (Hodson et al., 2007; Takeuchi et al., 2000), and 2) saturated interstitial void space (Irvine-Fynn and Edwards, 2014). We then develop the 'undescribed in Greenland to date' message in the subsequent paragraph. The biogeochemical message is revisited in the discussion, as suggested, where we provide slightly more detailed discussion and additional references.

Line 1: Remove "In sum"

Author response: Removed, as requested.

Line 2-4: Consider revisiting (see L25 above), and bringing in energy balance and ablation (see again Munro, 1990, AAAR) and describing the reasons for weathering crust relevance. Then have a single paragraph giving the justification for the study in Greenland. I just found these two paragraphs jump around a little and felt that a more logical progression through material could be achieved.

*Author response:* Thank you again for these helpful suggestions. We hope the revised introduction provides a clear progression and justification for the present study.

Line 20: delete "mechanical" – not necessary. Be consistent with hyperlinks/formatting if used for www sources.

Author response: Deleted, as requested.

Line 21: "drilling" in glaciology typically implies more than shallow coring, might just talking about "coring" and "core sites" be sufficient? (e.g. P4 L1 "core sites" seems more appropriate).

*Author response:* "drilling" and "drilling sites" are changed to "coring" and "core sites", as requested.

Line 23: Issue of mass for calculation of density is relevant here. Are you measuring water and ice? If so, are not the estimates of density in error. This issue needs to be addressed and accounted for; ice density has to be properly estimated given the depth variable water content.

Author response: The analysis assumes we are measuring dry ice density. Though water retention errors are inevitable, we are not aware of an explicit estimate of depth-dependent water content for the very near-surface weathering crust we document in this study. As suggested, we cite estimates of temperate ice water content ranging from 0-9%, though most estimates (15 of 18) are <3.4%, including all estimates made from in situ calorimetric methods (Pettersson et al., 2004). The uppermost 9% estimate is well within our  $\pm 20\%$  storage error. We provide a thorough reporting of this error source in the methods.

Line 29: Suggest "This uncertainty is incorporated into calculations of ice porosity and water content (see Sections 2.2 and 2.4)". In places, as here, writing clarity and conciseness could be tightened up.

Author response: As requested, the sentence has been revised to read "This uncertainty is incorporated into calculations of ice density and porosity, propagating into  $\pm 20\%$  uncertainty in specific storage (see Sect. 2.2 and Sect. 2.4)."

**Page 4**

Line 8-9: Clarify the relevance of the centre of mass, if you are using the method to estimate the upper 14-30cm ice, just indicate that the upper 20cm is used, but the sampler geometry results in bias toward the uppermost ice and so leads to an underestimate of density. This just seems to be introducing terms which could be seen as confusing.

Author response: As requested, the reference to "centre of mass" is removed and the sentence now reads: "... the density measurements may be more representative of the uppermost  $\sim$ 6 cm of material because of the shape of the sampler (see Fig. 2)".

**Author changes in manuscript:**

Line 11: Issue of water content in core sections and uncertainties in density measurements remains problematic.

*Author response:* Thank you for drawing attention to this important source of uncertainty. We hope our clarified discussion of the two primary sources of expected error (water retention and loss of material) are sufficiently addressed in the revised methods.

Line 12-15: This could be condensed: e.g. "for context, two 1.8m cores were extracted but ice density measurements were not undertaken, these cores are described further in Section 3.3".

Author response: The sentence now reads as suggested.

Line 15: Estimates of porosity will be affected by ice core sections that were weighed still containing interstitial water. Removal of this water is not a trivial problem, as exposing the core to positive air temperatures will initiate further melt, and methods of forcing water out via centrifugal force is similarly challenging. An estimate of uncertainty is needed here, and this needs to be incorporated into the data derived from these potentially erroneous ice mass measurements.

Author response: Most estimates of temperate ice water content are less than <3.4%, well below the assumed  $\pm 10\%$  uncertainty we prescribe for density and porosity. Water retention errors would tend to cancel (to an unknown extent) with the expected volumetric errors due to loss of material at the irregular ends of the core segments. We appreciate the point and do not take it lightly, but we do not think there is a reasonable way to construct a physically based error estimate beyond the provided  $\pm 10\%$  (density and porosity) and  $\pm 20\%$  (specific storage). We highlight these sources of error in the methods and remind the reader as they are presented in the results and discussion.

Line 20: rationale for the change in ( ) to [ ]?

Author response: Thank you for noticing, [] has been changed to () throughout.

Line 25: Might combining the equations here to A = B > C (as Eq 1) seem a neater and more consistent presentation of the equations? Would allow a slightly smoother explanation.

Author response: The equations are combined, as requested.

Line 30: Does the time-frame and temperature of the water present any issue here? Given the thermal potential of supraglacial water (which I presume was used?), could you estimate and mass loss (or confirm this is negligible). The size of the weathering crust crystals might be important here – were the samples used representative of the upper 20cm for all sites?

*Author response:* Supraglacial water was used for the reason you mention. Water was immediately applied and, as noted, we carefully observed ice crystal structure and air bubbles and saw no evidence of melt. To some extent, the focus on the quality of the linear regression (noted below in a separate comment) was motivated by this concern. The linear relationship suggests 0% effective porosity for solid ice density (917 kg m-3), which provides some confidence in the measurements, and the estimates made from the relationship. Regarding representativeness, we collected samples at every core site as well as 15 additional sites along the transect to increase the sample size (N=25). This has been noted in the revised methods.

**Page 5**

Line 9: While it is good practice to cite, do we need more than one of two examples here? Just considering journal space.

Author response: We reduced the citations to two, as requested.

Line 19: "8m intervals"?

*Author response:* We changed to "8m posting", to distinguish them as physical locations, e.g. "cryoconite holes were measured within 1 m radius of the posting …"

Line 20 & Page 6 Line 2: were cryoconite holes ubiquitous features, or did you measure those proximate to the sample point? Clarify here. If one hole was measured – is this representative of local water table – might measuring 4 holes in at each site have provided more robust estimates?

Author response: They were ubiquitous, but for expediency we measured the nearest cryoconite hole within a 1 m radius of the posting. We note this in the revised text. Regarding representativeness, we were not able to confirm this via direct measurements at each posting (again, owing to severe time limitations). Measuring 4 holes would certainly provide more robust estimates of local variability but we were more concerned with obtaining an adequate sample size to establish conditions along the transect. We have added the following statement where we report water levels: "The height of water in these holes likely varied diurnally and could have steadily drained or filled during the study period (Cook et al., 2016). The 15.5 cm average depth to water thus likely represents a snapshot of the transient water table height. Further, owing to severe time limitations we measured a single hole at each 8 m posting and thus cannot quantify local variability."

Line 21: The steel rod measurements are not entirely convincing, can you justify this a little more clearly. Furthermore, as above, a conceptual model might help here. Perhaps you need to consider the density decay curve (LaChappelle, 1959) and clarify your reasoning here, or use some alternative term in terms of a qualitative measure of "weathering crust mechanical resistivity" to the steel probe to indicate perhaps the shoulder on the density decay curve? There is also the issue of capillary draw in the weathering crust, are you able to confirm the water table in the crust is the same as that in the ice matrix? Does the water table truly exist as a broadly consistent level? If not, is this an uncertainty you can at least note if not estimate.

*Author response:* Thank you for this important critique. We agree the steel rod measurements were not presented clearly. We have removed the a priori characterization of the saturated/unsaturated transition where the depth probe measurements are described in the methods. In the revised manuscript, we use the depth to water below the ice surface in cryoconite holes as an estimate of the depth to saturation, whereas the depth probe measurements are used as a qualitative characterization of the weathering crust structure, drawing on the characteristic subsurface depth-density profile for weathering crust (LaChapelle, 1959) as suggested. Thank you for this suggestion, we think it will substantially improve the communication of our results to the reader.

**Page 6**

Line 5: Refrozen water while a component of storage in an overarching sense, is not the liquid storage, and is likely to be a proportion of the total available liquid water following a freezing event or water drainage to a cold front in the ice. If you are talking about liquid "water storage" then surely it is a negative value/term in that it is water lost to freezing? I'd also caution here given the inference is that the ice lenses are refrozen water – which may or may not be the case (see comment below regarding ice structure) and so a clearer definition of water storage might be helpful here.

Author response: We agree and have removed the term, as requested.

Line 10: see L19, but are you exploring a total storage potential or just the saturated ice. Can you remove "saturated" here, and discuss both the observed water storage volume and the potential storage volume?

Author response: We are exploring the actual storage within saturated ice. Our use of the word "potential" was incorrect and misleading. L19 has been revised to read "Finally, for illustrative purposes we scale our storage estimate to the study catchment by multiplying the mean  $S_P$  estimated from the shallow ice cores by the bare ice surface area of the study catchment ...".

Line 11: do you not just "extract" cores, rather than excavate them?

Author response: We have changed "excavated to "extracted", as requested.

**Author changes in manuscript:**

Line 19: I think you need to better define the "potential total storage volume" (i.e. the entire weathering crust) vs. the estimated snapshot of water storage yielded by your observations – given the weathering crust storage potential will be time-variant given the nature of the weathering crusts formation. You could then discuss the total storage potential (under the conditions at the time of measurement), and the proportion of that which did indeed hold liquid water (the stored volume), and compare those to melt production or runoff volumes.

*Author response:* We present the actual (instantaneous) specific storage estimated from the shallow cores, and then scale that to the study catchment to estimate a storage volume. Our use of the word 'potential' was incorrect.

**Page 7**

Line 24: "metre rule"?

Author response: The sentence has been removed altogether as it is redundant with the methods.

Line 32: Given the strong surface expression of structural features in the sector of the ice sheet studies here, you might give an indication that ice structure might underlie this (and perhaps cite papers that note the strong evidence of structural glaciology at the locality, and its theoretical background – for example Hambrey et al, 2000, Geol Soc; Hudleston, 2015, J Struc Geol.). You could at least provide a hypothesis here as a potential guide for future work. Given ice lenses are discussed, and given the ice sheet surface is ablating, these lens features must be emergent – and while it is possible refreezing of meltwater may contribute, do ice temperatures or meteorological conditions support this given the prevalence of these lens features? Were the lenses truly horizontal in formation or exhibit slight orientation?

*Author response:* We agree these lenses must be emergent and have substantially revised our discussion, including the suggested references. Regrettably we did not make careful observation of their orientation. The revised discussion now reads:

"Previous analyses of weathering crusts have not reported this pattern of alternating clear, solid ice and fractured, granular ice (e.g. Hoffman et al., 2014; Müller and Keeler, 1969; Schuster, 2001). It seems likely these lenses are structural features, as refrozen meltwater in unlikely in a thermally temperate weathering crust (Schuster, 2001), and even less so weathered ice at depths below refrozen meltwater. Stratified distribution of grain size, crystal structure, bubbles, and/or impurities with depth (Hudleston, 2015) could each influence the rate of subsurface radiative

heating and hence weathering (Brandt and Warren, 1993; Liston et al., 1999). Meltwater advection along micro seams or cracks may also promote differential weathering (Hambrey, 1977; Hambrey and Lawson, 2000), similar to joint block weathering of terrestrial lithology. Surface expression of differential weathering is certainly evident along the transect (Figure 1), and at broader scales in the region is associated with outcropping of impurities (Wientjes et al., 2012). The ice lenses, then, may represent structural resistance to weathering, and/or result from heterogeneity in subsurface flow paths that enhance differential "rotting" of subsurface ice (Nye, 1991). We would thus expect lenses to be localized features, which helps explain the lack of consistent stratigraphy among cores."

Line 3: "The reported pM values therefore"? Missing word.

Author response: We added "values", thank you.

Line 4: Perhaps use "lens ice" to clarify your meaning here.

Author response: We have substituted 'lens ice', as requested.

Line 6: Surely if lenses are refrozen water their density will likely approach that of pure ice, and if structural features, their persistence would suggest higher glacier ice density values. As such, can you not include and quantify potential uncertainties here?

*Author response:* In keeping with our interpretation of ice lenses as structural features, we now include the ice lens volume in our estimates of density, volume, and porosity. Either way, as pointed out by another reviewer, it was incorrect to omit the ice lens volume from the original estimate as the mass was incorporated into the density calculation. As such, we have removed the various references to this source of error, which should reduce confusion for the reader.

Line 12: You have two data points above the theoretical limit, so "consistently smaller" is not strictly correct. You also refer to Figure 5 here three times in as many lines – consider using just one reference to the graphic.

*Author response:* We have removed all but one reference to Figure 5, and replaced "consistently smaller" with "generally smaller".

Line 15-16: "unphysical"? Perhaps use "physically implausible".

Author response: "Unphysical" is replaced with "physically implausible", as requested.

Line 15 & 18: There seems to be an overemphasis on the quality of the data here, please recall the equation used to derive the porosity value means there is circularity here – the porosity is a function of the two densities. Avoid overstating something here. You can simply report the observed relationship, the fact that how robust this is beyond the bounds of observations is equivocal, and that the relationship was used to estimate porosity.

Author response: We removed the overemphasis on the quality of the data here, following the

suggested progression of reporting the relationship, the uncertainty beyond the range of observations, and the application of the relationship to the shallow core densities.

Line 28: This section seems a little less flowing than others, and is characterized by short paragraphs. Can the core holes and the water levels noted in these be described further? The first paragraph and third surely belong together? But there is repetition here. Consider revisiting this section.

*Author response:* As requested, elements of the first and third paragraphs have been combined and the section has been condensed to four individual paragraphs with the following progression:

- 1) Description of ice surface topography, cryoconite hole depths, and depth to water along the transect
- 2) Description of refilling of drilled holes and water filled cryoconite holes as evidence of saturation
- 3) Description of the two-layer structure referencing the LaChapelle depth-density profile
- 4) Description of higher density material structure and possible lower bound on permeable ice from the two 1.8 m cores

**Page 9**

Line 5: So do dry holes indicate the water table is more complex and not a level surface?

*Author response:* Perhaps, but dry holes were on average 8 cm shallower than wet holes, suggesting this result may be due to random sampling of shallower holes. More importantly, we think, is the lack of a trend in depth to water below the surface, suggesting the water table generally mirrored the local topography.

Line 7: not sure you need to use caption detail in the figure reference here.

Author response: Caption detail was removed, as suggested.

Line 11: I think you need to define where the ice is saturated – it isn't the full depth of the weathering crust, or is it? Just feel a little more clarity in needed here to ensure the observations and inferences are clearly described.

*Author response:* As requested, we state clearly that depth to saturation is inferred from the depth to water in cryoconite holes.

Line 21: You don't really have a handle on the "transient" nature of the weathering crust here – yes, you can conceptualise this as a two-layered feature. But although you show spatial variability, you have no detail on temporal change. I would focus on the message relating to the snapshot of water storage – and the volume that represents. And only in your discussion, mention the processes of weathering crust formation and how this would mean the depths of the porous and saturated ice would potentially vary.

*Author response:* As requested, 'transient' has been removed and the paragraph/section now focuses on the snapshot of water storage by reporting the measurements we made, namely, the

average depth of cryoconite holes, the average depth to water, the trend toward shallower holes with distance  $(-0.014 \text{ cm m}^{-1})$ , and the lack of trend in depth to water, suggesting the water table generally mirrored the local topography.

We continue to think the two-layer description is helpful, simply because there was such a distinct transition in density/structure observed in the field, and because it was remarkably consistent with the structure implied by the diagram in Müller and Keeler, (1969). The primary evidence for the two-layer transition comes from the depth probe and shovel, since the corer was unable to remove the upper material intact and sampling resolution of density was limited to ~13 cm. As you suggest, the depth probe and shovel provide indirect evidence of a certain material strength, and is merely qualitative. To clarify this for the reader, we reference the density decay curve as suggested. We think the conceptual schematic we are preparing for the introduction will also greatly improve the communication of this idea to the reader. That said, we have debated the usefulness of the two-layer description and can remove it if you think it detracts from the quality of the presentation.

The sentence now reads "Based on these observations, we characterize the near surface ice as consisting of two bulk, though in principle continuous, layers."

Line 34: Further evidence for structural controls on the ice crystallography?

**Author response: We think so, yes.**

**Page 10**

Line 4: Repetition of freezing leading to cessation of coring from method section, un-helpful here as a result section.

Author response: The repetitive statement was removed, as requested.

Line 6: It would be nice to see a little more result reporting here – not solely the reference to the table and the mean for all sites. Perhaps expand a little.

*Author response:* This section has been expanded, as requested. We report the specific storage depth (referencing the table), and then use this as a transition to discuss how the depth of weathered ice and storage relate to the seasonal context. Here, we report the maximum spring snow depth, date of snow disappearance, cumulative ice surface ablation following snow disappearance, and comparison with regional meteorology reported in Tedstone et. al., (2017).

Line 13: Just wondered if a clearer summary section leading to discussion might be helpful – in following with the results. For example, open with the lacking recognition of the weathering crust, and how here, observations of ice density revealed X Y and Z on a portion of the Greenland Ice Sheet, then move to how the two-layered structure matches previous work, and how density and water storage values compare to the other limited reports.

*Author response:* The opening paragraph has been combined with the second paragraph and reorganized according to the recommended structure i.e. 1) lack of recognition of weathering crust storage volumes, 2) broad restatement of our findings, 3) two-layered structure consistent

with previous work, and 4) density and water storage consistent with previous limited reports.

Line 14: Cite the Larson reports and Irvine-Fynn et al review that discuss near-surface surface storage here. Would citing the Jansson et al (2003, J Hydro) review also be useful here?

Author response: We have added the Larson, Irvine-Fynn, and Jansson citations, as requested.

Line 15: Why the specifics on polythermal ice sheets here? The references cited discuss temperate glaciers and a polythermal glacier, respectively.

Author response: References to thermal regime have been removed, as requested.

Line 21: "stagnating"??

Author response: "stagnate" has been replaced with "stagnating", thank you.

Line 1: Recall the reports you cite simply modelled water storage via water budgets – and so you can't compare core observations to hydrological models. Previous work hasn't examined ice cores to identify or report crystallographic changes. Please revisit.

Author response: The references to specific storage rates have been removed and replaced with a general statement of consistency with previous findings of substantial water storage: "Moreover, these results are consistent with observations of substantial water storage within the weathering crust of supraglacial environments worldwide (Irvine-Fynn, 2008; Larson, 1978; Munro, 1990)."

Line 4: See earlier comments on ice structure.

Line 8: See earlier point about water budget equation, and the ice lenses being a negative 'storage' value, as indicated here. However, a stronger physical discussion of the potential formation processes for the ice lenses is needed – with comparison to ice structure and any alternative explanations too.

*Author response:* This paragraph has been substantially revised to emphasize the interpretation of the lenses as structural features. However we do use this paragraph as an opportunity to provide a more nuanced view of refreezing, highlighting work that suggests negligible refreezing in porous near-surface ice (Schuster, 2001; Wheler and Flowers, 2011) as well as non-negligible refreezing (Hoffman et al., 2014a).

Line 10: GrIS – either define and use as acronym throughout or use Greenland Ice Sheet as elsewhere.

Author response: GrIS has been changed to Greenland Ice Sheet, as requested.

Line 11: Condense to a single paragraph section perhaps?

Author response: The two paragraphs are condensed to a single paragraph, as requested.

Line 25: I'd suggest revisiting in view of the Munro (1990, AAAR) source.

*Author response:* We are not sure exactly what the reviewer is asking us to revisit in view of Munro (1990). We suspect it may be the finding that discrepancies between ablation and runoff were reconciled for that experiment by combining a detailed energy balance model with a specially designed ablatometer. If so, we agree the Munro experiment presents a useful demonstration this is possible, though we think it equally highlights the difficulty required to reconcile the effect of weathering crust on runoff and mass balance. We cite Munro (1990) throughout the revised discussion where relevant.

**Page 12**

Line 24: Lutz reference focuses on ice algae, not cryoconite. Suggest Wientjes and Boggild references would be more appropriate here. Similarly, L25: Fountain discussed ice-lidded cryoconite in Antarctica which may physically be a little different – suggest a more cautious use of literature which refers to the types of feature and observations that are characteristic for Greenland (e.g. the older Gribbon, 1979, J Glac. or Gadja, 1958, Can Geogr. references for cryoconite holes in Greenland).

*Author response:* The sentence has been revised to discuss microbial communities in general, with references added and revised as suggested.

**Page 13**

Line 12: Does Hoffman's study relate to a temperate or polythermal ice mass – isn't it cold? Or just remove the thermal regime aspect here – "supraglacial environments elsewhere. . ."

Author response: The reference to thermal regime has been removed and replaced with "supraglacial environments worldwide", as suggested.

Line 15: You define the symbology, no need to repeat the definition here in L16, after its use on L15.

Author response: The symbology has been removed, as requested.

Line 19: For impact, suggest you rephrase as "if these observations are representative of the Greenland Ice Sheet ablation zone, then wider implications are. . ., and future work should. . .."

Author response: The conclusion has been rephrased as suggested: "If these findings are representative of broader areas of the Greenland Ice Sheet ablation zone, they suggest the potential for substantial sub-seasonal meltwater storage within porous low density ice on the Greenland Ice Sheet ablation zone bare ice surface. Future work should examine how the weathering crust evolves in response to spatio-temporal changes in the surface energy balance, and quantify potential errors in sub-seasonal SMB and surface elevation change estimates derived from surface energy balance models and altimetry, as most currently neglect removal of mass via subsurface melting in the weathering crust."

Fig.3.: Consistency with (..) or [..] on axes labels. The captions seem to be overly long – focus on the content and remove superfluous text. Label 1 -10 in the figure. Hatched areas are "no data" not "core depth" are they not? Can you not include the snow- shovel data here for the uppermost 20cm – albeit in a different colour, for comparison and completeness? Might inclusion of potential ablation here be helpful given from the field campaign description, the cores were collected over one week during which time ablation would take place – and such that (for example) a refreezing event (if this is what the lenses are) might be more clearly identified if lenses appear at the same depth relative to a zero set for the period of coring?

*Author response:* The cores were collected on 11-12 July. We state this clearly in the revised methods. We have updated axis labels with consistent use of (), labeled the cores 1-10 in Figure 3, changed 'core depth' to 'no data', and gap filled the upper 20 cm with the snow cutter density measurements, as requested. Revised figures are included at the end of this documented.

Fig.4.: Is the lower image for the core in the upper? Perhaps use arrows to indicate where ice lenses are on the core.

*Author response:* The photos were taken from different cores but are the best photos we have. The caption has been updated to note the core location.

Fig.5.: y-axis should be phi-eff. The equation given should be phi-hat-eff (inconsistent symbology). Surely "observations" not "data"? Caption – is "measured data" needed here?

*Author response:* The equation has been corrected to read phi-hat-eff, 'data' has been changed to 'observations', and 'measured data' has been removed from the caption, as requested. Regarding the y-axis label, since the axis is used for phi-hat-eff, measured phi-eff, and phi-total, we think it is better to leave the label as generic phi [-] and let the legend distinguish, though if requested we are happy to change to all three relevant symbols.

Fig.6.: (b) there is a lot of information here, and I just wonder if two panels here would be helpful – one to give clearer indication of the water level in holes with a simple zero as ice surface, and then the detrended plot with the unsaturated crust estimate? The two grey tones are hard to differentiate. If detrended, surely the data should be scattered around zero – so did you offset this to a maximum positive deviation - one presumes so, but clarification would be appropriate? Have you compared distance or elevation against any of the variables – are there any other patterns to explore – as these don't seem to have been mentioned in the main text – even if to confirm there is no elevation dependency.

*Author response:* The grey shaded area has been removed to improve the clarity, as requested. We also added some empty space at either end and added the depths of the shallow cores as per a request from another reviewer.

Regarding the offset, yes, they were all offset to the maximum positive deviation such that the datum is 0. This was done for consistency with the conceptual diagrams of Muller and Keeler (1969) and Irvine-Fynn and Edwards (2014).

Regarding distance relations, as noted elsewhere we did find a slight trend toward shallower

holes  $(-0.014 \text{ cm m}^{-1})$  but no significant trend (or apparent trend whatsoever) in depth to water below the surface.

Table 1: could you include a column of mean phi-eff for each core here, for ease of direct comparison?

*Author response:* The mean phi-eff has been added as well as mean density. The porous ice and lens ice depths have been removed, but can be added back if requested.

 $\phi_{\rm eff}(-)$ #5 #2 ÷ Depth (cm)  $s^{0}$  ,  $v^{0}$  ,  $v^{0$

Updated figures/table:

Figure 3: Subsurface measured ice density ( $\rho_M$ ) and corresponding calculated effective porosity ( $\phi_{eff}$ ), and stratigraphy profiles from 10 shallow ice cores (#10-1, left to right) extracted at 80 m postings along the study transect (see Figure 1 for ice core locations). Horizontal blue shading represents solid ice layers. Vertical dashed line at solid ice density 917 kg m-3. Assumed ±10% measurement uncertainty represented by shaded grey bars. Hatched areas are no data.

---

## Author Response (AR1)

Dear referee #1,

Thank you for the thorough and supportive comments. The suggestion to contextualize the study in terms of the seasonal cycle was particularly fruitful (and was shared by all reviewers). To address this, we have reported the seasonal and annual ablation recorded by a nearby automatic weather station, and added a comparison with regional meteorology reported in a recent publication that is highly relevant to our study (Tedstone et al., 2017). These data reveal that spring and summer 2016 was characterized by conditions particularly favorable to weathering crust development, and therefore should help contextualize our findings. We have also substantially revised our interpretation of the ice lenses, emphasizing the role of ice structure throughout the revised manuscript. I provide a line by line reply to each comment below, and include revised figures at the end of the document. Thank you kindly,

Matthew Cooper

**Anonymous Referee #1**

The paper presents the results of a recent field survey conducted to assess the potential for subsurface meltwater storage in porous ice in the ablation zone of the Greenland ice sheet. Focusing on a small, internally draining, hydrological catchment in the much-studied South West, the authors find that the subsurface 'reservoir' consists of two layers. These consist of a thin, light, unsaturated layer atop a thicker ($\leftarrow$1 m), denser, saturated layer. Between them, these layers provide storage potential for up to 20 cm of meltwater which, integrated over the catchment, is the equivalent of one hour's proglacial discharge from this sector.

The methods employed in the study are sensible, and their results are very interesting, however the manuscript is a little confused in places and I would have liked to see the field data discussed in more detail. That said, this is a good paper and I expect will make a solid contribution to the literature subject to editing as follows.

**Major comments**

**1.** The paper is lacking in analysis of spatial variability along the transect studied. For example, Fig 3 reveals that ice lenses are not always common between adjacent cores. Further investigation into these features and why they arise would be both interesting and serve to strengthen the manuscript.

*Author response:* We have added substantial interpretation of the ice lenses that will help explain the lack of consistent stratigraphy in adjacent cores. We believe the ice lenses are structural ice features, not refrozen meltwater. For the crust to develop, the ice must be temperate, which should prevent substantial refreezing (Schuster, 2001). We think it is more likely the lenses are remnant solid ice that has undergone less weathering owing to structural heterogeneity in ice grain size, bubble size/content, and impurities. We also speculate that

meltwater advection along micro seams or cracks in the ice may promote differential weathering, similar to joint block weathering of terrestrial lithology. Also, ice lens stratigraphy in firn is highly variable and discontinuous over spatial scales as short as 1.5 m (Brown et al., 2011; Machguth et al., 2016), thus spatial analysis of lens features is unlikely to prove fruitful. Each of these would suggest lenses are local features, helping to explain the lack of consistent stratigraphy among cores. In the revised, we have added substantive descriptions of these ideas to support our hypothesis that the lenses are structural features, not refrozen meltwater.

To address spatial variability in cryoconite holes along the transect we tested for statistically significant linear relationships between distance and 1) depth of cryoconite holes, and 2) depth to water in cryoconite holes. No relationship was found for depth to water though there was a slight trend toward shallower holes (-0.012 cm m$^{-1}$, p<0.004). We report these findings in the revised Sect. 3.3, along with a strengthened discussion of variability along the transect.

*Author changes in manuscript:* P8 L19-34, revised presentation of ice lens results with alternative hypothesis regarding ice structural controls, including the role of preferential flow paths and ice foliation.

P9 L15-28, added discussion of spatial variability in cryoconite holes and water table height along the transect.

P12 L8-24, expanded discussion of ice lens features, their possible relationship with underlying ice structure, and expanded discussion of meltwater refreezing independent of the ice lenses.

**2.** While you do not have seasonal data, you should contextualize your findings in terms of the seasonal cycle. For example, what is the ablation rate in this location?

*Author response:* Thank you for the excellent suggestion to contextualize our study in terms of the seasonal cycle. As requested, we obtained daily measurements of ablation recorded by the KAN-M automatic weather station (AWS) during our field study. KAN-M is located ~8.3 km ENE of our field site at ~1270 m a.s.l. and is the most proximal AWS to our field site (1215 m a.s.l.). Sonic ranging data recorded by KAN-M indicate the maximum spring snow depth was ~50 cm and the snow disappearance date was ~21 June, which suggests the conditions we document developed over an ~21-day period between snow disappearance and the collection of the ice cores on 11-12 July, or otherwise persisted to some extent through the previous winter. Following snow disappearance, AWS data indicate a cumulative ice surface lowering of ~55 cm prior to collection of the shallow ice cores on 11-12 July. The average ablation rate during this time was 2.62 cm d$^{-1}$. The mean annual ablation rate at KAN-M is ~1.25 m a$^{-1}$ (van As et al., 2017). These statistics are reported in the revised Sect 3.4 paragraphs 2-3.

To supplement these AWS data we added a comparison with a recent publication that examines the relationship between regional meteorology and remotely sensed surface reflectance in the study region (Tedstone et al., 2017). Their analysis of regional meteorology from the Modèle Atmosphérique Régional (MAR) regional climate model (Fettweis et al., 2017) suggests ~50 cm average snow depth and mid-June snow disappearance date in summer 2016 (see Figure 3 in Tedstone et al., 2017), consistent with the AWS data we analyze. Further, their analysis suggests that meteorological conditions during summer 2016 were ideal conditions for weathering crust

development. These include below average cloud cover and rainfall, and above average downward shortwave radiation (e.g. compare to Figures 1–4 in Tedstone et al., 2017). From these data and the AWS data, we conclude it is not surprising that a well developed weathering crust was present in the study area at the time of observation. We have added several discussion points throughout the manuscript to emphasize the seasonal context.

*Author changes in manuscript:* P11 L5-24, added comparison with antecedent seasonal meteorology, report daily ablation rate prior to the study and the mean annual ablation rate in the region.

**2. (continued)** Is your weathering crust likely to be the product of one, or more melt seasons?

*Author response:* We interpret the literature to suggest the weathering crust is a seasonal phenomenon with no interannual carryover except in the case of stagnating ice (e.g. Fountain and Walder, 1998). On sub-seasonal timescales, the weathering crust can rapidly decay when the surface energy balance is dominated by longwave or turbulent heat fluxes that melt the surface, removing the crust and exposing solid ice. Common examples would include heavy rain, very warm winds, or warm cloudy conditions. However, it is conceivable that a deep weathering crust could persist to the end of the melt season if meteorological conditions allow. In this case, we would expect interstitial and surficial meltwater to refreeze following snowfall. Annual snowfall in this part of Greenland is typically <1.0 m (Tedstone et al. 2017) and therefore meltwater that does not drain from the crust likely refreezes during winter and/or at night. If snow cover is absent or ephemeral, the crust may sublimate over winter. However, interannual variability in these conditions is likely substantial. Though we agree the annual progression of the porous ice is a logical next step for research on the topic, we prefer not to comment extensively on interannual carryover without a physical model to support the analysis. That said, we have added several points to the revised that address seasonal and interannual evolution of the crust, including a note that the mean annual ablation rate is ~1.25 m a$^{-1}$ at KAN-M. Given that we document weathered ice at depths > 1.6 m, it is conceivable that near-surface, low density ice persists on interannual timescales.

*Author changes in manuscript:* P11 L5-24, added comparison with antecedent seasonal meteorology, and report the mean annual ablation rate in the region.

P11 L20-24, we note the >1.6 m thickness of weathered ice suggests ice structure may underlie the observed weathering crust and hence interannual carryover is conceivable.

P14 L32 – P15 L6, added a concluding statement about the importance of understanding the seasonal and interannual variability in order to determine the net effect of the conditions we document on mass balance and hydrologic processes.

**2. (continued)** You attribute ice lenses to remnant glacier ice but to me it seems that these are evidence of meltwater refreezing at depth. Can you correlate the incidences of ice lenses to e.g. the annual cycle or specific weather events?

*Author response:* As noted above, we do not think the ice lenses are refrozen meltwater, rather we think they are structural features, as was suggested in the original manuscript and also by

reviewer #3. Further, based on similar analyses of meltwater lenses in firn we do not think the annual cycle or specific weather events will prove fruitful (Brown et al., 2011; Machguth et al., 2016).

*Author changes in manuscript:* P8 L19-34, revised presentation of ice lens results with alternative hypothesis regarding ice structural controls.

P12 L8-24, expanded discussion of ice lens features, their possible relationship with underlying ice structure, and expanded discussion of meltwater refreezing independent of the ice lenses in subsequent paragraph (with reference to new Fig. 8).

**3.** While it is good that you discuss the implications of your findings for SMB and surface hydrology studies, I would like to see a bit more qualitative information here. For example, what would the difference in ice mass be in your catchment a) at the density of ice and b) when you account for sub-surface porosity?

*Author response:* The difference in ice mass in the catchment can be evaluated from the difference between solid ice density and in situ measured density integrated across the depth of porous ice:

$$\Delta m = \int_0^{h_o} \rho(h, t_o) \cdot dh - \int_0^{h_o} \rho(h, t_1) \cdot dh \qquad (1)$$

where $m$ is mass, $h$ is ice thickness, $\rho$ is ice density, and $t$ is time. Assuming $\rho(t_o) = 917$ kg m$^{-3}$, integrating Eq. (1) across the depth-density profiles from our shallow cores yields $\Delta m = 254$ kg m$^{-2}$ or 25.4 cm water equivalent. Integrating this across the 63.1 km$^2$ catchment yields $1.6\times10^{10}$ kg or 0.016 Gt.

However, the mass of stored water must also be accounted for. Subtracting the 17.3±4.3 cm mass of stored water we document from the above estimate yields $\Delta m = 81\pm43$ kg m$^{-2}$ or 8.1±4.3 cm water equivalent. Integrating this across the 63.1 km$^2$ catchment yields $5.11\times10^9$ kg or 0.0051±0.0027 Gt. The first estimate (0.016 Gt) can be considered a maximum plausible mass difference as it assumes there is no subsurface stored meltwater, whereas the latter (0.0051 Gt) estimate is a minimum based on the water storage we document. In reality, the mass of stored meltwater is time-variant, thus we can provide a snapshot estimate at best.

The effect of subsurface porosity on mass, therefore, depends on the timescale considered, the initial conditions, and, critically, the role of meltwater drainage. It is very much a moving target. We omitted this discussion from the original manuscript to keep it focused on the instantaneous characterization of density, porosity, storage, and weathering crust structure, owing to incomplete knowledge of the initial ice density profile, the density profile below the depth of the shallow cores, and the fate of the stored meltwater. Though we recognize the interest in the above demonstration, we feel strongly that the above analysis is very preliminary and a careful examination of the effect of our findings on ice mass requires a physical model of weathering crust development, meriting a standalone investigation. Instead, we report the instantaneous specific storage we document, and compare this to the seasonal and annual ablation rates.

*Author changes in manuscript:* P11 L12-14, added comparison of our specific storage estimate

with the seasonal and annual ablation rates in the study region.

**2. (continued)** Similarly, hydrological studies use estimates of snow permeability for water routing according to Darcy's Law. Can you provide an updated estimate of sub-surface permeability in your study area for future use by such studies?

*Author response:* We agree the results of our study point to the importance of sub-surface permeability but unfortunately, we cannot provide an estimate from our data. We do, however, discuss the topic in the discussion where we cite the four studies we are aware of that provide estimates of ice permeability, and we compare these to estimates of supraglacial channel flow velocities. Our hope is that an interested reader will use these citations as a resource for this topic.

*Author changes in manuscript:* Discussion of ice permeability with references to prior literature on P14 L9-18.

**Minor comments:**

Line 29-30: I'm not sure that meltwater throughput 'reinforces concerns about ... sea level rise'

*Author changes in manuscript:* P1 L28, the statement is removed, as requested.

Line 29: Greenland ice sheet not Greenland Ice Sheet.

*Author response:* We respectfully submit that our capitalization is correct for a proper noun referring to a geographical place name. Reviewer 3 also expressed a preference for Greenland Ice Sheet, so we decided to keep it as is.

Line 3: Mention that these models assume that runoff is instantaneously lost to sea here.

*Author response:* As requested, we mention these models assume runoff is instantaneously lost to sea. To supplement this, we add reference to several works that demonstrate substantial time lags and possible meltwater retention in the ablation zone, motivating the study of near-surface porous ice and ablation zone hydrologic processes.

*Author changes in manuscript:* P1 L30, P2 L1-7, added discussion of model assumption of instantaneous loss to sea, and references to works that document substantial meltwater runoff time lags.

Line 8: Sentence structure is odd here; implies that the stored meltwater is the substrate. Is that what you mean?

*Author response:* Thank you for catching this error. The "weathering crust" is the substrate. As requested, "meltwater storage" has been deleted.

*Author changes in manuscript:* P2 L7, sentence is rewritten.

Line 16: Maybe mention melting due to friction from the flow of meltwater.

*Author response:* As requested, this additional source of melting has been noted, but we added this to the interpretation of the ice lenses in the discussion.

*Author changes in manuscript:* P12 L16, added discussion of meltwater friction as a possible source of sub-surface ice weathering.

Line 31: 'could potentially' rather than 'would'

*Author changes in manuscript:* P3 L11 'would' is changed to 'could', as requested.

Line 2: Logical disconnect here, add an explanatory line.

*Author response:* We are not sure what logical disconnect the reviewer is referring to but the introduction has been substantially revised and we hope the problem has been corrected.

Line 6: Phrasing of 'near surface ablating' seems strange.

*Author response:* The phrase is removed, as requested, and we hyphenated 'near-surface' throughout to improve phrasing elsewhere (the phrasing is used to emphasize that the study is focused on bare, ablating ice, to avoid possible confusion about firn, snow, or superimposed ice).

Line 12: Add melt zones onto map in Fig 1

*Author response:* The entire area in Fig 1 was actively melting during the study period.

Line 16: Delete 'study area'

*Author changes in manuscript:* P3 L17, 'study area' deleted, as requested

Line 27: Example of logical jump; you assign and uncertainty then say where you got it. It would be better to say, 'we consider 1.3 cm (10%) accuracy to be conservative'.

*Author changes in manuscript:* P4 L26, sentence revised as requested.

Line 29: Just measurement uncertainty or a combination of measurement and instrument uncertainty?

*Author response:* Just measurement uncertainty. We used a pair of calipers and digital scale to make the measurements and we assume the instrumental uncertainty is substantially less than the measurement uncertainty. We have, however, improved our discussion of measurement uncertainty, specifically describing the two primary sources of error we expect are important 1) ice core volume measurement error owing to loss of material near the irregular ends of the individual ice core segments, and 2) interstitial meltwater retention errors owing to capillary

water retention and incomplete free water drainage. The volumetric error would tend to result in underestimated ice density, the water retention in overestimated density. Hence, the two would tend to cancel to an unknown extent. Estimates of temperate ice water content range from 0-9%, though most estimates (including all based on in situ calorimetric methods) are in the range 0-3% (Pettersson et al., 2004). Recognizing that both sources of error are poorly constrained, we think our original 10% estimate is sufficiently conservative without giving undue confidence to either the measurements or the error estimate.

*Author changes in manuscript:* P4 L25 – P5L3, revised discussion of error sources.

P7 L6-10, revised presentation of density and porosity uncertainty propagation into specific storage uncertainty.

Line 12: What determined the maximum depth?

*Author response:* The maximum depth was determined based on the 1 m drill barrel length. A drill extension is required to retrieve cores deeper than 1 m. We did not expect weathered ice to extend below 1 m depth, so we designed our field methods to remove 1 m cores using the standard drill barrel without any extensions. After drilling the first few cores, we realized the ice was weathered to at least 1 m depth, but owing to time limitations, and for consistency with the first few cores, we chose to drill each of the ten cores to 1 m depth. For additional context, we drilled two 1.8 m cores near camp (described in the text), but we were not able to systematically return to each core site and drill deeper.

*Author changes in manuscript:* P4 L21-24, note the 1 m drill barrel length and explain two additional 1.8 m cores collected for context.

Line 28: Be consistent with units; you used cm3 before.

*Author response:* The units have been changed throughout the text, as requested.

Line 6: Delete 'these'

*Author changes in manuscript:* P5 L24 'these' is deleted, as requested

Line 7: Justify this given the difference in structure between the two layers.

*Author response:* Thank you for pointing out this important caveat to our study. We have added substantial discussion of structural differences between the two layers in the revised results Sect. 3.3 and discussion Sect. 4, and we acknowledge this source of uncertainty in the results Sect.

*Author changes in manuscript:* P9 L12, we acknowledge this source of uncertainty. P10 L7-25, expanded discussion of structural differences between the two layers. P12 L8-24, additional discussion of structural differences between layers.

Line 20: Nearest cryoconite hole? Nearest x cryoconite holes?

*Author response:* The nearest cryoconite hole within a 1 m radius of the posting. We have moved this description further up so the methods are clear to the reader right away.

*Author changes in manuscript:* P6 L11, specified the nearest cryoconite hole within a ~1 m radius of the posting was measured. P9 L24, we note this limits our ability to quantify local variability in cryoconite hole water levels.

Line 27: Another logical jump re: transition!

*Author response:* Thank you for pointing out the logical disconnect with respect to the unsaturated/saturated layer transition. We have removed the a priori characterization of the saturated/unsaturated transition in the methods and instead report the transition in the results (where it belongs). We use the depth to water below the ice surface in cryoconite holes as an estimate of the depth to saturation, whereas the depth probe measurements are used as a qualitative characterization of the weathering crust structure. We hope this addresses the logical disconnect.

We designed a conceptual diagram for the revised introduction that we hope will improve the visual communication of the weathering crust structure to the reader. The diagram merges elements of the conceptual diagram from Müller and Keeler, (1969), and Irvine Fynn and Edwards (2014), with the characteristic subsurface depth-density profile for weathering crust from LaChapelle, (1959):

[Figure]

**Fig. 1. (a) Conceptual diagram of weathering crust structure, highlighting the porous ice layers, cryoconite holes, and saturated water table (adapted from Irvine-Fynn and Edwards, 2014 and Müller and Keeler, 1969). (b) Theoretical sub-surface depth-density profile showing the non-linear increase in ice density from the highly porous, low density near-surface ice to a higher density substrate (adapted from LaChapelle, 1959).**

In the text, we have clarified the following:

1) The unsaturated depth is inferred from the depth to water in cryoconite holes

2) The depth probe measurements are used as a qualitative description of weathering crust structure with reference to the sub-surface density profile and conceptual schematic
3) The apparent transition from very low density to higher density material is reported in the results

*Author changes in manuscript:* P6 Sect. 2.3, a priori characterization of saturated/unsaturated transition is removed, as requested.

P6 L13-14, we specify the depth to water in cryoconite holes is used as estimate of water table height (hence depth to saturation).

P6 L17-23, we clarify the depth probe is used as a qualitative check on the weathering crust structure with reference to new Fig. 1 conceptual diagram.

P9 L22-27, we expand the discussion of water table height variability inferred from depth to water in cryoconite.

P10 L7-16, we report the depth probe measurements and clarify they provide qualitative inference about the depth of weakly bonded, disintegrated ice as per revised Fig. 1.

Line 30: To what height? It would have been good to measure the water table at the drilled holes as well as at Cryoconite holes.

*Author response:* We agree these measurements would have been good to measure, but we think the cryoconite hole water levels provide a better (non-destructive, equilibrium) estimate of the water table height. We were also severely limited by time. For example, it wasn't clear how much time was required for the water levels in the drilled holes to equilibrate. We planned to measure the water levels after all other science priorities were completed, but returning to (and locating) all 100 drilled holes was infeasible. Instead, we use the refilling of the drilled holes as a proxy for water saturation, and use the depth to water in the cryoconite holes as a measure of the water table height.

Line 8: Impermeable yes, but how continuous? Are these highly localized features?

*Author response:* As noted above, we have substantially revised our interpretation of the ice lenses to suggest they are highly localized structural features.

*Author changes in manuscript:* P8 L19-34, revised presentation of ice lens results with alternative hypothesis regarding ice structural controls.

P12 L8-24, expanded discussion of ice lens features, their possible relationship with underlying ice structure, and expanded discussion of meltwater refreezing independent of the ice lenses.

Line 7: I think the value below snow surface would be a better one to quote.

*Author changes in manuscript:* P9 L21, value below ice surface is added, as requested.

Line 10: Again, to what level? Completely full?

*Author response:* As with the 1 m drilled holes (N=100), the water levels are used here as an indication of sub surface saturation, and the depth to water in cryoconite holes is used as an estimate of the (instantaneous, equilibrium) depth to saturation. This decision was made because the time required for the water levels in the drilled holes to equilibrate was unclear and we were unable to systematically return to each site to observe and measure these water levels.

Line 17: Did you see any evidence of flow in any of the holes?

*Author response:* We did not test for evidence of flow in the holes, other than our observations of refilling, which implies subsurface permeability and flow.

Line 25: Quantify 'often'

*Author response:* We agree this sentence is somewhat vague and we have removed the sentence altogether. The observations were not systematic and hence should not be reported.

Line 29: Ok so the water table is ←20 cm below the surface yes? Which is consistent with your statement that the bottom layer, from 20cm down is saturated.

*Author response:* The water levels in cryoconite holes were on average 15 cm below the ice sheet surface, which is our best estimate of the water table height. We apologize for the confusion in the original manuscript, resulting from the depth probe measurements. We use the average 15 cm depth as an estimate of the depth to saturation.

*Author changes in manuscript:* P9 L21-27, depth to water in cryoconite holes (hence water table height) is reported.

P10 L10-16, depth of crust penetrated with depth probe reported, and increase in material strength below this depth discussed in relation to the 15 cm depth to water.

Line 29-30: This is speculative without seasonal data. If the aquifer is perennially saturated then this is not necessarily the case.

*Author response:* We appreciate that this is speculative without seasonal data, but the average annual maximum snow depth in this region of Greenland is <1.0 m, thus we think it is highly unlikely the aquifer is perennially saturated.

Fig 3, reorient so #10 is on the left as in fig1.

*Author response:* We have revised Figure 3 such that #10 is on the left, as requested. Additionally, as per a request from Reviewer 3, we have constructed continuous depth-density profiles by substituting the snow-cutter density measurements for the upper 20 cm at cores #1, 2,

4, 5, 9, and 10, and used linear interpolation to gap fill missing data between 20–30 cm depth for cores #1, 4, 5, and 9. The new figure is included below.

Fig 6 b add core depth.

*Author response:* Core depths have been added, as requested. Revised figures included below.

**Revised figures/tables:**

[Figure]

**Fig. 1. (a) Conceptual diagram of weathering crust structure, highlighting the porous ice layers, cryoconite holes, and saturated water table (adapted from Irvine-Fynn and Edwards, 2014 and Müller and Keeler, 1969). (b) Theoretical sub-surface depth-density profile showing the non-linear increase in ice density from the highly porous, low density near-surface ice to a higher density substrate (adapted from LaChapelle, 1959).**

[Figure]

**Fig. 2.** Ortho-rectified image mosaic of the study area at 6 cm ground resolution from RGB camera imagery collected 10 July 2016 on board a quad-copter drone. Background 30 m Landsat image collected same day. Shallow ice cores extracted at 80 m posting (blue circles) along the 800 m transect provide ice density measurements to depths of 1.1 m, with two additional shallow ice cores extracted to 1.8 m depth at posting 1. Insets (below) show the 63.1 km² supraglacial catchment extent (magenta outline), as delineated from WorldView satellite stereo-photogrammetric digital elevation model topography, and supraglacial river and moulin locations derived from Landsat 8 imagery (Yang and Smith, 2016).

[Figure]

Fig. 3. (a) A surface weathering crust was pervasive throughout the study area, characterized by small scale topographic variability and cryoconite holes. (b-c) A 1000 cm$^3$ steel snow density sampler was vertically inserted into the upper 20 cm weathered ice. (d) A shallow ice core drill was used to obtain ice samples to depths of 1.8 m.

[Figure]

**Fig. 4.** Sub-surface measured ice density ($\rho_M$) and corresponding calculated effective porosity ($\phi_{eff}$), and stratigraphy profiles from 10 shallow ice cores (#10-1, left to right) extracted at 80 m posting along the study transect (see Fig. 2 for ice core locations). Horizontal blue shading represents solid ice layers. Vertical dashed line at solid ice density 0.917 g cm$^{-3}$. Assumed ±10% measurement uncertainty represented by shaded grey bars. Hatched areas are no data.

[Figure]

Fig. 5. (a) Typical near-surface shallow ice core (core #6) prior to in situ analysis of density and stratigraphy. Clear, solid ice lenses alternate with granular, fractured ice. Approximate locations of ice lenses noted with white arrows (not all lenses are clearly visible). (b) Ice lenses removed and confirmed after completed core analysis (core #1).

[Figure]

**Fig. 6.** Linear relationship ($\hat{\phi}_{eff}$, solid line) between measured ice density ($\rho_M$) and effective porosity ($\phi_{eff}$) and assumed ±10% measurement error (whiskers). Dashed line is theoretical upper limit where effective porosity equals total porosity (i.e. $\phi_T = \rho_M/\rho_T$).

[Figure]

**Fig. 7. (a) Ice sheet surface topography along the 800 m study transect extracted from a 6 cm posting stereo-photogrammetric digital elevation model derived from RGB imagery collected 10 July 2016 from a quad-copter drone and the 2ⁿᵈ-order polynomial best fit. (b) Ice sheet surface topography detrended with the polynomial best fit, cryoconite hole depths (vertical grey bars), and cryoconite hole water levels (vertical blue bars) sampled along the 800 m study transect, adjusted to a common vertical reference. Locations of the 10 shallow boreholes and their depth relative to the detrended surface are labelled #1-10.**

**Table 1: Shallow ice core depth, mean core density, mean core porosity, and specific storage depth (S$_P$), for each shallow ice core.**

| Core | Ice Core Depth | Mean Core Density | Mean Core Porosity | $S_P$ |
|------|------|------|------|------|
| | (cm) | (g cm$^{-3}$) | (-) | (cm) |
| 1 | 100 | 0.72 | 0.19 | 12 – 16 |
| 2 | 100 | 0.72 | 0.19 | 11 – 15 |
| 3 | 100 | 0.76 | 0.15 | 10 – 13 |
| 4 | 90 | 0.63 | 0.28 | 15 – 21 |
| 5 | 89 | 0.63 | 0.27 | 16 – 21 |
| 6 | 97 | 0.74 | 0.17 | 15 – 20 |
| 7 | 90 | 0.65 | 0.26 | 15 – 20 |
| 8 | 102 | 0.72 | 0.19 | 15 – 20 |
| 9 | 90 | 0.64 | 0.26 | 16 – 21 |
| 10 | 82 | 0.64 | 0.27 | 14 – 18 |
| **μ** | **94** | **0.69** | **0.22** | **14 – 18** |

[Figure]

**Fig. 8. Night-time refreezing of meltwater at the surface of cryoconite holes and water tracks was frequently observed during the field study.**

Dear referee #2,

Thank you for the thorough and supportive comments. The suggestion to contextualize the study in terms of the seasonal cycle was particularly fruitful (and was shared by all reviewers). To address this, we have reported the seasonal and annual ablation recorded by a nearby automatic weather station, and added a comparison with regional meteorology reported in a recent publication that is highly relevant to our study (Tedstone et al., 2017). These data reveal that spring and summer 2016 was characterized by conditions particularly favorable to weathering crust development, and therefore should help contextualize our findings. We have also substantially revised our interpretation of the ice lenses, emphasizing the role of ice structure throughout the revised manuscript. I provide a line by line reply to each comment below, and include revised figures at the end of the document. Thank you kindly,

Matthew Cooper

**Anonymous Referee #2**

This paper presents findings from a field campaign on the Western margin of the Greenland ice sheet concerning the nature of the 'weathering crust' on the bare ice in the ablation zone. The paper provides measurements from shallow ice cores of ice density and corresponding porosity as well as water content, finding surprisingly low density ice down to at least 1 meter depth. It is pointed out that the presence of this weathering crust means that (sub-surface) melt does not necessarily correspond to 'surface' lowering, as might be measured by satellite altimetry.

I think this is an interesting set of measurements, and is a valuable contribution to understanding the supraglacial hydrology of the Greenland ice sheet. Meltwater storage in the percolation zone of the Greenland ice sheet (in firn) has been well documented over the last few years, and this study suggests that a non-negligible amount of water storage and transport may occur beneath the apparent ice surface in the ablation zone too.

The paper is well written and the figures are mostly clear. I have one main comment and a number of minor comments, mostly seeking clarification.

**Main comment**

The measurements represent a snap-shot of the weathering crust in mid-July 2016 and it is not clear how this relates to the behaviour over the course of the melt season. I appreciate that the field campaign was limited in length so it may not be known how the crust itself evolves, but I think there needs to be more discussion of the setting for these measurements. In particular, what is the annual ablation rate in this region? At what stage of the melt season are these taken (i.e. roughly how much melting has already occurred here)? What is the ice temperature in this region? These are important issues in understanding how reflective these results are of wider spatial scales but also larger time-scales.

*Author response:* Thank you for the excellent suggestion to contextualize our study in terms of the seasonal cycle. As requested, we obtained daily measurements of ablation recorded by the KAN-M automatic weather station (AWS) during our field study. KAN-M is located ~8.3 km ENE of our field site at ~1270 m a.s.l. and is the most proximal AWS to our field site (1215 m a.s.l.). Sonic ranging data recorded by KAN-M indicate the maximum spring snow depth was ~50 cm and the snow disappearance date was ~21 June, which suggests the conditions we document developed over an ~21-day period between snow disappearance and the collection of the ice cores on 11-12 July, or otherwise persisted to some extent through the previous winter. Following snow disappearance, AWS data indicate a cumulative ice surface lowering of ~55 cm prior to collection of the shallow ice cores on 11-12 July. The average ablation rate during this time was 2.65 cm d$^{-1}$. The mean annual ablation rate at KAN-M is ~1.25 m a$^{-1}$ (van As et al., 2017). These statistics are reported in the revised Sect 3.4 paragraphs 2-3.

To supplement these AWS data we added a comparison with a recent publication that examines the relationship between regional meteorology and remotely sensed surface reflectance in the study region (Tedstone et al., 2017). Their analysis of regional meteorology from the Modèle Atmosphérique Régional (MAR) regional climate model (Fettweis et al., 2017) suggests ~50 cm average snow depth and mid-June snow disappearance date in summer 2016 (see Figure 3 in Tedstone et al., 2017), consistent with the AWS data we analyze. Further, their analysis suggests that meteorological conditions during summer 2016 were ideal conditions for weathering crust development. These include below average cloud cover and rainfall, and above average downward shortwave radiation (e.g. compare to Figures 1–4 in Tedstone et al., 2017). From these data and the AWS data, we conclude it is not surprising that a well developed weathering crust was present in the study area at the time of observation. We have added several discussion points throughout the manuscript to emphasize the seasonal context.

*Author changes in manuscript:* P11 L5-24, added comparison with seasonal meteorology to emphasize the seasonal context, report the daily ablation rate prior to the study and the mean annual ablation rate in the region, and compare to Tedstone et al, (2017) to contextualize in terms of interannual variability.

P14 L32 – P15 L6, added a concluding statement about the importance of understanding the seasonal and interannual variability in order to determine the net effect of the conditions we document on mass balance and hydrologic processes.

**Main comment (continued)**

In particular, I think there could be more discussion of how the inferred stored water thickness (15-22cm) relates to the amount of melt that has so far been produced this season (roughly what fraction of it is this), and how the depth of the porous ice compares with the amount that melts each year. Eg. are the ice lenses that form the result of recent refreezing (i.e. earlier the same year) or some earlier time?

*Author response:* As noted above, KAN-M data indicate ~55 cm of ice surface ablation prior to collection of the shallow ice cores on 11-12 July, equivalent to 49.5 cm water equivalent assuming solid ice density of ~900 kg m$^{-3}$. The inferred stored water thickness (revised to 14-18 cm) is therefore ~28–36% of the cumulative seasonal melt.

The mean annual ablation rate is ~1.25 m a$^{-1}$ at KAN-M (van As et al., 2017). The shallow cores were limited to less than <1.1 m depth, though we observed weathered ice at depths >1.6 m, suggesting the weathering crust depth at the time of observation was greater than the mean annual ablation rate.

*Author changes in manuscript:* P11 L10-14, we report the amount of melt (ice surface ablation) produced prior to collection of the cores, and the fraction of this represented by the specific storage we document.

P11 L13, we report the mean annual ablation rate. P11 L20, we compare the mean annual ablation rate to the depth of porous ice, as requested.

*Author response (cont.):* Regarding ice lenses, we have substantially revised and (we hope) improved our discussion of their nature. We do not think they are refrozen meltwater but rather we think they are structural features, as was suggested in the original manuscript and also by reviewer #3. For the crust to develop permeability, the ice must be temperate, which should prevent substantial refreezing (Schuster, 2001). The situation is different in firn, where the medium in inherently permeable, allowing meltwater to penetrate along preferential flow paths forming refrozen lens horizons. Preferential flow paths are possible along fractures in weathered ice, though we consider it very unlikely that a progressive stratigraphy of ice lenses could develop in an ablating weathering crust. We discuss this throughout the revised results and discussion.

*Author changes in manuscript:* P8 L19-34, revised presentation of ice lens results with alternative hypothesis regarding ice structural controls, including the role of preferential flow paths and ice foliation.

P12 L8-24, expanded discussion of ice lens features, their possible relationship with underlying ice structure, and expanded discussion of meltwater refreezing independent of the ice lenses.

**Main comment (continued)**

There could then be some discussion of how the porous ice evolves over the course of the year. Presumably all the water freezes again in winter? In which case the ice is mostly solid at the start of the melt season (except perhaps the unsaturated surface layer, which seems to have a similar porosity to snow)? If the saturated sub-surface water doesn't ever run-off, it may simply be changing the quantity of runoff rather than delaying it.

*Author response:* We interpret the literature to suggest the weathering crust is a seasonal phenomenon with no interannual carryover except in the case of stagnating ice (e.g. Fountain and Walder, 1998). On sub-seasonal timescales, the weathering crust can rapidly decay when the surface energy balance is dominated by longwave or turbulent heat fluxes that melt the surface, removing the crust and exposing solid ice. Common examples would include heavy rain, very warm winds, or warm cloudy conditions. However, it is conceivable that a deep weathering crust could persist to the end of the melt season if meteorological conditions allow. In this case, we would expect interstitial and surficial meltwater to refreeze following snowfall. Annual snowfall in this part of Greenland is typically <1.0 m (Tedstone et al. 2017) and therefore meltwater that

does not drain from the crust likely refreezes during winter and/or at night. If snow cover is absent or ephemeral, the crust may sublimate over winter. However, interannual variability in these conditions is substantial. Though we agree the annual progression of the porous ice is a logical next step for research on the topic, we prefer not to comment extensively on interannual carryover without a physical model to support the analysis. That said, we have added several points to the revised that address seasonal and interannual evolution of the crust, including a note that the mean annual ablation rate is ~1.25 m a$^{-1}$ at KAN-M. Given that we document weathered ice at depths > 1.6 m, it is conceivable that near-surface, low density ice persists on interannual timescales.

*Author changes in manuscript:* P11 L5-24, added comparison with antecedent seasonal meteorology, and report the mean annual ablation rate in the region.

P11 L20-24, we note the >1.6 m thickness of weathered ice suggests ice structure may underlie the observed weathering crust and hence interannual carryover is conceivable.

P14 L32 – P15 L6, added a concluding statement about the importance of understanding the seasonal and interannual variability in order to determine the net effect of the conditions we document on mass balance and hydrologic processes.

**Specific comments**

Why are the findings frequently referred to as 'preliminary'? What are they preliminary to? If they are really preliminary, it begs the question why they are being published. I'd suggest that if the authors think the results are worth publishing they should not refer to them as preliminary (which does not preclude doing more work on the topic).

*Author response:* We have removed our use of 'preliminary' throughout the text. Our original intent was to highlight the relatively immature status of the research topic in general, and more so its application in Greenland, not our specific findings.

Line 29: Why does the routing of surface water to the ocean 'reinforce concerns' about contribution to global sea level rise? Isn't such melting part of the 'normal' operating cycle of an ice sheet?

*Author response:* We have removed the statement, as requested. Yes, it is part of the normal operating cycle. Our intent was to highlight the "efficient drainage" hypothesis, which is the assumption that ablation zone meltwater is transported rapidly, in its entirety, to surrounding oceans. To develop this idea more clearly, the revised introduction references works that demonstrate substantial time lags and possible meltwater retention in the ablation zone as motivation for the study of near-surface porous ice and ablation zone hydrologic processes.

*Author changes in manuscript:* P1 L30, P2 L1-7, revised discussion of efficient meltwater delivery assumption, and added references to works that document substantial meltwater runoff time lags in the ablation zone.

Section 2.1: It is not clear from this description how liquid water in the core is dealt with. Is it allowed to drain out? Presumably there is still quite a lot of water trapped in the core samples (due to capillary forces) and this contributes to the measured mass?

*Author response:* Thank you for this important observation. We did not explain this adequately in the manuscript. The drill barrel was held vertically and allowed to drain when cores were removed from the boreholes prior to weighing. After removal from the borehole, the drill was laid at a slight angle and the core was carefully removed from the drill barrel and immediately analyzed, providing additional time for drainage. Though our aim was to drain the cores completely, it is correct that some water remained owing to capillary forces. It is also possible that some non-capillary water remained owing to incomplete free-drainage. These water retention errors would result in overestimated ice density.

In adding a more thorough discussion of this issue to the methods section, we also provide more detail about the measurement uncertainty noted in the original manuscript. Namely, the natural breaks of the ice cores were irregular and some material was inevitably lost near the ends of the core segments. The 10% error estimate we provided in the original manuscript was meant to account primarily for this loss of material at the irregular ends of the ice core segments, which would tend to result in underestimated ice density.

To summarize, there are two primary sources of error we expect are important 1) ice core volume measurement error owing to loss of material near the irregular ends of the individual ice core segments, and 2) interstitial meltwater retention errors owing to capillary water retention and incomplete free water drainage. The volumetric error would tend to result in underestimated ice density, the water retention in overestimated density. Hence, the two would tend to cancel, though to an unknown extent as both errors are poorly constrained.

In the revised methods, we describe these error sources in greater detail, and we cite estimates of temperate ice water content ranging from 0-9%, though most estimates (15 of 18) are <3.4%, including all estimates made from in situ calorimetric methods (Pettersson et al., 2004). The uppermost 9% estimate is thus well within our ±20% (revised to ±14%, see P7 L6-8) specific storage uncertainty estimate. We think this is sufficiently conservative without giving undue confidence to either the measurements or the error estimate.

*Author changes in manuscript:* P4 L25 – P5L3, revised discussion of error sources as noted above.

P7 L6-8, revised calculation of density and porosity uncertainty propagation into specific storage uncertainty.

Line 9: This is a bit awkward wording, since this statement presumably assumes that the density is uniform (independent of depth). Perhaps better just to say that the geometry of the sampler means that the near-surface ice is disproportionately weighted in this average, rather than quantifying the 'center of mass'.

*Author changes in manuscript:* P4 L18, sentence is revised to read as requested.

Section 2.3: There is some confusion here about the 'unsaturated weathering crust depth', and how it relates to how penetrable the ice is. Reading further, it seems that the 'water table' (which I would interpret as the unsaturated depth) roughly coincides with a change in the strength of the ice that is presumably what the depth probe is detecting. It does not seem obvious to me why these two surfaces (the impenetrable ice surface and the water table) should happen to coincide - perhaps the presence of air in the pores above this allows the surface ice to 'rot' more rapidly. Or perhaps the permeability of the upper layer is sufficiently large that water in this layer readily runs off horizontally keeping it unsaturated. Perhaps the qualitative description of the surface given on page 9, line 20, could be moved forward to the method section to help explain these issues. In any case it would help to be clearer precisely what is meant by unsaturated - does this mean there is no liquid water, only residually-trapped water, or that water does not fill the pore space (all of which are different)?

*Author response:* We apologize for the confusing presentation of the steel rod measurements and the unsaturated depth in the original manuscript. In the revised, we have removed the a priori characterization of the saturated/unsaturated transition in the methods and instead report the transition in the results, where it belongs. The unsaturated depth is estimated from the depth to water below the ice surface in cryoconite holes, whereas the depth probe measurements are used as a qualitative characterization of the weathering crust structure. As you correctly note, the steel rod detects an increase in material strength, which tends to coincide with the water table, as suggested in the diagram from Müller and Keeler, (1969), however there is no obvious reason why these two surface coincide, though it may be related to ice density as you suggest.

To contextualize the depth probe measurements and the weathering crust structure for the reader, we designed a conceptual diagram for the revised introduction that we hope will improve the visual communication of the weathering crust structure to the reader. The diagram merges elements of the conceptual diagram from Müller and Keeler, (1969), and Irvine Fynn and Edwards (2014), with the characteristic sub-surface depth-density profile for weathering crust from LaChapelle, (1959):

[Figure]

**Fig. 1. (a)** Conceptual diagram of weathering crust structure, highlighting the porous ice layers, cryoconite holes, and saturated water table (adapted from Irvine-Fynn and Edwards, 2014 and Müller and Keeler, 1969). **(b)** Theoretical sub-surface depth-density profile showing the non-linear increase in ice density from the highly porous, low density near-surface ice to a higher density substrate (adapted from LaChapelle, 1959).

Near-surface weathered ice tends to exhibit a characteristic increase in density from a very low-density surface layer to a higher density sub-surface that approaches solid ice density (described in revised introduction P2 L24-30). The very low-density surface layer is demonstrated by the coarse material above the water table in the conceptual diagram. This is the material the depth probe penetrates, which we suggest may be indicative of the "shoulder" on the density decay curve where density increases non-linearly (Fig. 1b). We hope the diagram will clarify the depth-variable nature of the crust for the reader and provide context for the depth probe measurements.

In the text, we have clarified the following:

1) The unsaturated depth is inferred from the depth to water in cryoconite holes
2) The depth probe measurements are used as a qualitative description of weathering crust structure with reference to the sub-surface density profile and conceptual schematic
3) The apparent transition from very low density to higher density material is reported in the results

*Author changes in manuscript:* P6 Sect. 2.3, a priori characterization of saturated/unsaturated transition is removed, as requested.

P6 L13-14, we specify the depth to water in cryoconite holes is used as estimate of water table height (hence depth to saturation).

P6 L17-23, we clarify the depth probe is used as a qualitative check on the weathering crust structure with reference to new Fig. 1 conceptual diagram.

P9 L22-27, we expand discussion of water table height variability inferred from depth to water in cryoconite.

P10 L7-16, we report the depth probe measurements and clarify they provide qualitative inference about the depth of weakly bonded, disintegrated ice as per revised Fig. 1.

Line 5: Why is refrozen meltwater included as water storage? If it has refrozen it is ice again and should be thought of as storage (it requires melting again - with the associated energy implications - before it could run off).

*Author changes in manuscript:* P6 L3, refrozen meltwater removed from definition of storage, as requested.

Line 19: Is the 'potential' liquid storage capacity not just the effective porosity multiplied by depth and total area (i.e. including the currently unsaturated pore space too)?

*Author response:* Our use of the word "potential" was incorrect. Your characterization is correct but we present the actual (instantaneous) specific storage estimated from the shallow cores, and then scale that to the study catchment by multiplying the average storage depth by the total area to estimate a storage volume.

*Author changes in manuscript:* P7 L12-17, revised to read "Finally, for illustrative purposes we scale our $S_P$ estimate to the study catchment by multiplying the lower and upper values for $S_P$ estimated from the shallow ice cores by the bare ice surface area of the study catchment …".

Line 7: There seems to be some subjectivity involved here. Why is estimating the value wrong in one direction deemed 'not problematic'? If the densities were measured including the ice lenses, would it not make sense to use the volume including the ice lenses when converting to water content using the effective porosity? Or otherwise use a solid ice density of the ice lens to infer the density of the non-ice lens part of each segment?

*Author response:* Thank you for this important critique. We agree it makes sense to include the ice lens' volume when converting to water content since they are included in the density. This was an oversight on our part. We have removed the various references to the ice lens density bias, which will remove unnecessary confusion for the reader. Moreover, given our interpretation of the lenses as structural features, it makes sense to include their volume in the storage estimate.

*Author changes in manuscript:* The sentence has been removed altogether.

Line 4: The drill did not go below 1.8m for fear of freezing. Did you make any measurements of temperature in the porous crust? It would be helpful to know if the ice is all at the bulk melting temperature or if it goes below this at depth.

*Author response:* We agree that temperature measurements would be invaluable. The reason we note the "risk of freezing" is to suggest, albeit indirectly, that the ice may be sub-freezing at this depth. Unfortunately, measuring sub-surface ice temperature is very difficult and error prone unless done with considerable care, and we were not able to undertake such measurements. Co-author Miege has extensive experience drilling firn cores and he suggested based on his observations of the drill behavior at these depths that the ice was freezing. However, the sentence has been removed as per request from reviewer #3.

Line 20: What is meant by a storage 'rate'? I could not work out what this number means.

*Author response:* This should have been referred to as 'specific storage rate' and is the specific storage (i.e. storage depth) divided by the time over which the meltwater storage accumulated. Nevertheless, we have removed the comparison with these rates because they were estimated using water budgets and reviewer #3 objected to the comparison with our core-density method.

*Author changes in manuscript:* The sentence is removed altogether.

Line 15: The 'lower and upper mean' is a strange concept; perhaps the 'mean lower and upper values' would be better wording.

*Author changes in manuscript:* P7 L12-13, sentence has been revised as requested.

Line 21 (and conclusions): Why are the results not considered representative of the rest of the ablation zone? I understand the desire for caution given that this is only one location, but without other evidence (perhaps you have it?) wouldn't the default assumption be that the results do apply more widely? What do you think is special about your field site that means the results would not apply more widely? Perhaps you could just say 'We do not know whether these findings represent typical conditions....' rather than 'not proposing' it.

*Author response:* We do not have any evidence that this location was unique. However, weathering crust growth and decay is strongly controlled by local meteorology and therefore can be highly variable over short distance and time. To our knowledge, there are no studies of seasonal weathering crust formation in Greenland, but sub-surface melting in the study region has been modeled and shown to depend on snow cover, which varies with elevation (van den Broeke et al., 2008). Lacking spatial data, we were being cautious, but we agree we should not over-emphasize this speculative (albeit cautionary) assumption.

*Author changes in manuscript:* P13 L4, we have replaced 'we do not propose' with 'may not be representative'.

Throughout the text, we have removed statements suggesting our work is 'preliminary' or not representative of other locations.

**Revised figures/tables:**

[revised manuscript text omitted]

Dear referee #3,

First, thank you for your thorough and tremendously helpful review. It is clear you invested considerable time into this review and your comments and suggestions have greatly improved the quality of the manuscript. The suggestion to explore the role of ice structure was particularly fruitful. We have substantially revised our interpretation of the ice lenses, emphasizing the role of ice structure throughout the revised manuscript. We hope we have adequately addressed the various suggestions and concerns raised throughout. I provide a line by line reply to each comment below, and include revised figures at the end of the document. Thank you kindly,

Matthew Cooper

**Anonymous Referee #3**

This paper reports on the findings from a field campaign on the ablation zone in southwestern Greenland. The focus of the paper is the so-called "weathering crust" that characterises glacier and ice sheet surfaces, and its potential hydrological storage role. The authors use a set of shallow ice cores (n=10) to describe the variability in near-surface ice density over depths of < 2 m. From these observations, the authors explore an effective porosity of the near-surface ice, and examine a potential water storage based on observations of a water table evident within the weathering crust. A specific storage of ~0.2 m is derived, suggesting that at the time of observations a water volume equivalent to 1 hours' worth of discharge from the local supraglacial catchment was essentially stored within the weathering crust.

The findings are a useful demonstration that this weathering crust exists on the Greenland Ice Sheet, and provides a sensible trigger for future work assessing the supraglacial drainage system and its functionality. Although some recent focus in Greenland has included the firn aquifer at higher elevations, it is an interesting insight to an overlooked hydrology of the ablating bare ice sector of the ice sheet. The growing recognition of the supraglacial realm as an ecosystem, and the potential importance of water storage on biogeochemical cycling at the ice sheet surface ensures this is a timely contribution and serves as a useful benchmark in this type of work.

Overall, the paper is well written, sensibly referenced, and the figures are clear. The methods are intelligible and could be repeated, and the calculations utilised are sensible within the limits of the data available/presented.

However, major comments would include:

A stronger description of how the weathering crust forms, and the subtlety of its growth and decay would be beneficial both in the introduction and in the later discussion. Specifically, would you expect a deep weathering crust at the time of your observations? Does the timing of snow melt, dominance of shortwave radiation, absence of rainfall give reason to consider the weathering crust (and ice lenses) you describe?

*Author response:* We added a stronger description of how the weathering crust forms in the revised introduction, as requested. We draw attention to the depth dependency of radiative heating and the characteristic increase in ice density from the very porous upper layer to the higher density subsurface (Cook et al., 2016; LaChapelle, 1959a). To communicate this to the reader more effectively, we designed a conceptual diagram for the revised introduction that we hope will improve the visual communication of the weathering crust structure to the reader. The diagram merges elements of the conceptual diagram from Müller and Keeler, (1969), and Irvine Fynn and Edwards (2014), with the characteristic subsurface depth-density profile for weathering crust from LaChapelle, (1959):

[Figure]

**Fig. 1. (a) Conceptual diagram of weathering crust structure, highlighting the porous ice, cryoconite holes, and saturated water table (adapted from Irvine-Fynn and Edwards, 2014 and Müller and Keeler, 1969). (b) Theoretical sub-surface depth-density profile showing the non-linear increase in ice density from the highly porous, low density near-surface ice to a higher density substrate (adapted from LaChapelle, 1959).**

To address the temporal context of our findings, we obtained daily measurements of ablation recorded by the KAN-M automatic weather station (AWS) during our field study. KAN-M is located ~8.3 km ENE of our field site at ~1270 m a.s.l. and is the most proximal AWS to our field site (1215 m a.s.l.). Sonic ranging data recorded by KAN-M indicate the maximum spring snow depth was ~50 cm and the snow disappearance date was ~21 June, which suggests the conditions we document developed over an ~21-day period between snow disappearance and the collection of the ice cores on 11-12 July, or otherwise persisted to some extent through the previous winter. Following snow disappearance, AWS data indicate a cumulative ice surface lowering of ~55 cm prior to collection of the shallow ice cores on 11-12 July. The average ablation rate during this time was 2.65 cm d$^{-1}$. The mean annual ablation rate at KAN-M is ~1.25 m a$^{-1}$ (van As et al., 2017). These statistics are reported in the revised Sect 3.4 paragraphs 2-3.

To supplement these AWS data we added a comparison with a recent publication that examines the relationship between regional meteorology and remotely sensed surface reflectance in the study region (Tedstone et al., 2017). Their analysis of regional meteorology from the Modèle

Atmosphérique Régional (MAR) regional climate model (Fettweis et al., 2017) suggests ~50 cm average snow depth and mid-June snow disappearance date in summer 2016 (see Fig. 3 in Tedstone et al., 2017), consistent with the AWS data we analyze. Further, their analysis suggests that meteorological conditions during summer 2016 were ideal conditions for weathering crust development. These include below average cloud cover and rainfall, and above average downward shortwave radiation (e.g. compare to figures 1–4 in Tedstone et al., 2017). From these data and the AWS data, we conclude it is not surprising that a well developed weathering crust was present in the study area at the time of observation. We have added several discussion points throughout the manuscript to emphasize the seasonal context.

*Author changes in manuscript:* P11 L5-24, added comparison with seasonal meteorology to emphasize the seasonal context, report the daily ablation rate prior to the study and the mean annual ablation rate in the region, and compare to Tedstone et al, (2017) to contextualize in terms of interannual variability.

P14 L32 – P15 L6, added a concluding statement about the importance of understanding the seasonal and interannual variability in order to determine the net effect of the conditions we document on mass balance and hydrologic processes.

**Major comments (continued):**

Is this a glacier ice weathering crust or one that perhaps is superimposed ice derived from snow and refrozen lenses forming therein? If this is glacier ice, then you should at least mention ice structure in addition to refreezing processes (particularly given the evidence of marked structure in the locality).

*Author response:* It was a glacier ice weathering crust, as data from KAN-M and from Tedstone et al. (2017) suggest the maximum snow depth in the region was ~50 cm and had completely melted by 21 June. We observed a few remnant snow patches in the field. The snow was heavily metamorphosed into uniform spherical grains and was easily distinguished from the pervasive coarse bubbly weathering crust ice. We do think superimposed snowmelt ice is an important part of the annual surface ablation cycle in the study area and probably contributes to the initial formation and structure of the crust, though we do not have data to support this presently. Regarding ice structure, we thank you for encouraging us to explore this more thoroughly. We have added considerable discussion of ice structure in the revised results and discussion.

*Author changes in manuscript:* P8 L19-34, revised presentation of ice lens results with alternative hypothesis regarding ice structural controls, including the role of preferential flow paths and ice foliation.

P12 L8-24, expanded discussion of ice lens features, their possible relationship with underlying ice structure, and separate discussion of meltwater refreezing independent of the ice lenses.

**Major comments (continued):**

A clearer emphasis regarding the results being a snapshot which reveals something about the supraglacial hydrology of the Greenland Ice Sheet could be beneficial. In your discussion, albeit

subjectively, are you able to comment on the likelihood of greater or less storage to be seen at other times of the summer, is this a seasonally progressive hydrological feature or is the observation just that, a discrete observation – there are climate records for the locality which might allow some extrapolation of these ideas.

Appreciably, it is not possible to go beyond perhaps a statement on this, given the limitations of the data set, but it would be helpful.

*Author response:* As noted above, we added a comparison with seasonal climate records from the nearby KAN-M AWS and with Tedstone et al. (2017) which together demonstrate climate conditions prior to the field study were favorable for weathering crust development. Regarding the seasonal progression, we report the annual ablation rate is ~1.25 m a$^{-1}$, therefore it is conceivable there could be some interannual persistence given we document a >1.6 m deep crust. We note this is dependent on meteorological conditions favorable to crust growth through the end of summer. Additionally, we emphasize in several places, including a new closing paragraph in the discussion, the transient nature of the crust growth and removal. We are hesitant to comment further without field data or a physical model of crust development.

*Author changes in manuscript:* P11 L5-24, added comparison with antecedent seasonal meteorology, report the daily ablation rate prior to the study and the mean annual ablation rate in the region.

P11 L12-14, added comparison of our specific storage estimate with the seasonal and annual ablation rates in the study region.

P11 L20-24, we note the >1.6 m thickness of weathered ice suggests ice structure may underlie the observed weathering crust and hence interannual carryover is conceivable.

P14 L32 – P15 L6, added a concluding statement about the importance of understanding the seasonal and interannual variability in order to determine the net effect of the conditions we document on mass balance and hydrologic processes.

**Major comments (continued):**

A slightly strengthened assessment of the uncertainties would be important I feel – this should include an assessment of the water content of the cores themselves as it is not clear if all interstitial water was removed prior to evaluating mass for density estimates. Temperate ice can have interstitial water content of up to ~10% (see Petterson et al., 2004, JGR), and certainly the saturated lower-most ice in the developing weathering crust may exhibit such water content if this is a seasonally temperate ice layer. Can you perhaps try to assess uncertainties associated with this water content, and the resultant impact on other estimates presented here. The 10% and 10% quoted seemed a little arbitrary when slightly more detailed and thorough approaches could have been taken. Furthermore, can you account for the ablation of the ice surface if cores were not all taken on one day – can the core profile Fig. be corrected/adjusted for surface ablation – making crude assumption that ablation over transect broadly similar, or using a point estimate from the energy balance? Correcting for the 7 days ablation period might be informative and aid inferences – such that for example, ice lenses may be better aligned perhaps if representative of

refreezing events or local thermal conditions.

*Author response:* Regarding ablation, the cores were collected over a period of two days (11-12 July, noted in the revised). The ablation rate was ~2.6 mm d$^{-1}$ during the field campaign. As such we have not adjusted the cores to account for this.

Regarding water content, thank you for raising this important point that was not adequately addressed in the initial manuscript. The drill barrel was held vertically and allowed to drain when cores were removed from the boreholes prior to weighing. After removal from the borehole, the drill was laid at a slight angle and the core was carefully removed from the drill barrel and immediately analyzed, providing additional time for drainage. Though our aim was to drain the cores completely, it is correct that some water remained owing to capillary forces. It is also possible that some non-capillary water remained owing to incomplete free-drainage. These water retention errors would result in overestimated ice density.

In adding a more thorough discussion of this issue to the methods section, we also provide more detail about the measurement uncertainty noted in the original manuscript. Namely, the natural breaks of the ice cores were irregular and some material was inevitably lost near the ends of the core segments. The 10% error estimate we provided in the original manuscript was meant to account primarily for this loss of material at the irregular ends of the ice core segments, which would to tend to result in underestimated ice density.

To summarize, there are two primary sources of error we expect are important 1) ice core volume measurement error owing to loss of material near the irregular ends of the individual ice core segments, and 2) interstitial meltwater retention errors owing to capillary water retention and incomplete free water drainage. The volumetric error would tend to result in underestimated ice density, the water retention in overestimated density. Hence, the two would tend to cancel, though to an unknown extent as both errors are poorly constrained.

In the revised methods, we describe these error sources in greater detail, and as requested, we cite estimates of temperate ice water content ranging from 0-9%, though most estimates (15 of 18) are <3.4%, including all estimates made from in situ calorimetric methods (Pettersson et al., 2004). We find no estimate of depth-dependent water content for near-surface <2m deep ice, hence a uniform water content error seems sensible, and the uppermost 9% estimate is well within our ±20% (revised to ±14%, see P7 L6-8) error estimate for specific storage. We think this is sufficiently conservative without giving undue confidence to either the measurements or the error estimate.

*Author changes in manuscript:* P4 L25 – P5L3, revised discussion of error sources as noted above.

P7 L6-8, revised calculation of density and porosity uncertainty propagation into specific storage uncertainty.

**Major comments (continued):**

Could more analysis of the data presented in Fig. 6 be made available here? There are

opportunities to examine patterns over elevation (small range though that is) and in relation to the detrended surface and 'roughness'. Similarly, it would be interesting to see if there is from the profiles (e.g. what are the patterns of phi-eff at, say, 33cm and 87cm depth, where it looks weathering crust (not ice lens) data is available across all cores – assuming these positions remain if adjusted for ablation over the 7-day sampling) – is there anything to be gained from a slightly deeper examination of the density and porosity data over depth and along the transect?

*Author response:* Regarding spatial variability along the transect, we tested for statistically significant linear relationships between distance/elevation and 1) depth of cryoconite holes, and 2) depth to water in cryoconite holes. No relationship was found for depth to water though there was a slight trend toward shallower holes (-0.012 cm m$^{-1}$, p<0.004). We report these findings in the revised Sect. 3.3. The same trends were found for elevation owing to the gradual increase in elevation along the transect (hence distance and elevation co-vary), but the distribution of elevation values was strongly skewed toward higher values. The lack of any trend in depth to water with distance confirms that the water table generally mirrored the small-scale roughness, which was noted in the original manuscript but was difficult to see in Fig. 6. In the revised, the grey filled area has been removed to improve interpretation of Fig. 6 (Fig. 7 in the revised, included below).

Regarding density and porosity depth-variability, we reframed portions of our analysis in terms of the theoretical depth-density curve presented in LaChapelle (1959), as suggested. We appreciate this suggestion and we hope it has improved the framing of the depth-density results. As requested further down, we gap-filled the missing data in Fig. 3 (Fig. 4 in the revised) so the depth-density profiles are complete, which provides a better comparison with the depth-density curve of LaChapelle (and the new Fig. 1 in the revised).

Regarding ice lens stratigraphy, we do not think they are controlled by temporal meteorological variability. As suggested, we think they are structural features possibly controlled by stratification in ice grain size, crystal structure, bubble size and/or content, or impurities. We also hypothesize that meltwater advection along micro seams or cracks may promote differential weathering, similar to joint block weathering of terrestrial lithology. We hope this general discussion of ice structure provides guidance to the reader, but we were not able to find patterns in density or porosity across cores at specific depths. This may not be surprising. For example, ice lens stratigraphy in firn is highly variable and discontinuous over spatial scales as short as 1.5 m (Brown et al., 2011; Machguth et al., 2016). Each of these would suggest lenses are highly local features, helping to explain the lack of consistent stratigraphy among cores.

*Author changes in manuscript:* P8 L19-34, revised presentation of ice lens results with alternative hypothesis regarding ice structural controls, including the role of preferential flow paths and ice foliation.

P9 L15-28, added discussion of spatial variability in cryoconite holes and water table height along the transect.

P12 L8-24, expanded discussion of ice lens features, their possible relationship with underlying ice structure, and expanded discussion of meltwater refreezing independent of the ice lenses.

**Major comments (continued):**

In places the writing style became less clear, or seemed to have a slightly reduced scientific quality. Similarly, a couple of key references seemed to be absent or choices of references seems a little misplaced, while in other places there was a proliferation of sources when perhaps just one or two examples would suffice. Some further editing and subtle revisions would likely be beneficial, to strengthen this paper, in addition to perhaps examining a few more relevant publications that would be of help in supporting these results and findings and their significance.

*Author response:* Thank you for these suggestions. We carefully reviewed the citations and found several instances where, we agree, the chosen citations were misplaced, especially in the introduction. To address this, we have carefully (we hope) separated citations that refer to theoretical ice permeability studies (Lliboutry, 1996; Mader, 1992; Nye, 1991; Nye and Frank, 1973) from those that deal with in-situ glacier ice studies (Cook et al., 2016; Fountain and Walder, 1998; Irvine-Fynn, 2008; Müller and Keeler, 1969), from those that deal with subsurface radiative properties of glacier ice (Brandt and Warren, 1993; Liston et al., 1999; Liston and Winther, 2005), and finally, we made effort to support statements focused strictly on cryoconite holes with relevant publications (Boggild et al., 2010; Cook et al., 2016) from those that deal with microbial communities in glacier ice more generally (Cook et al., 2012; Irvine-Fynn and Edwards, 2014). The paper has been substantially revised and we hope the writing style has been improved.

**Minor comments and suggestions (some touching on points above) would include:**

Line 1: Suggest hyphenate "near-surface" throughout. (There are some variations, e.g. P3 L13 and L15).

*Author response:* As requested, 'near-surface' has been hyphenated throughout.

Line 2: "Greenland Ice Sheet", as used throughout the manuscript. It is refreshing to see authors correctly use the appropriate capitalisation for proper nouns (it shouldn't be the Greenland ice sheet, given it is a specific location and entity) and at times I wish publishers would adhere to grammatical correctness – but that is another discussion altogether. L2: "Meltwater storage in low-density near-surface bare ice in the ablation zone of the Greenland Ice Sheet" might read a little better perhaps?

*Author response:* We agree the title reads better (with slight modification) as "Meltwater storage in low-density near-surface bare ice in the Greenland Ice Sheet ablation zone"

Line 16: Suggest referring to this as "specific storage".

*Author changes in manuscript:* P1 L15, replaced "liquid meltwater storage" with "specific meltwater storage" here, and throughout the manuscript, as requested.

Line 17: Clarify the water level is depth from the surface or from the base of the auger holes, and is "recharge" a more preferable term than infilling (given this is a hydrology paper).

*Author response:* Here, water level is the depth from base of the holes (i.e. height of water in holes). We report these water levels as they are suggestive of water storage in cryoconite holes. Elsewhere, we report water levels relative to the surface. "Infilling" is replaced with "refilling" here and throughout the manuscript, as requested.

*Author changes in manuscript:* P1 L16, we note water levels are "above hole bottoms". P9 L20-28 we clarify depth to water vs height of water above hole bottoms.

Line 18: "These observations are consistent. . ." given you present results and discuss them. Analysis might be provisional with clear directions to follow, but don't negate the potential utility of these observations.

*Author changes in manuscript:* P1 L17, "Though preliminary …" is removed, here and throughout the manuscript, as requested.

Line 21: "supraglacial catchment"

*Author changes in manuscript:* P1 L21 "catchment" is changed to "supraglacial catchment", as requested.

Line 25: A longer opening paragraph would be stronger as an opening. Can there be a clearer link from mass balance or ablation to runoff models for Greenland, and the assumptions regarding the efficient delivery. The sea level aspect here seems misplaced, as the study looks at in-season delays or reductions in discharge. Surely, noting the assumed efficient drainage is now being examined more closely with reference to the firn aquifer and so on would allow for a stronger introduction paragraph here.

*Author response:* We have lengthened the opening paragraph, as requested. To clarify the efficient delivery to surrounding oceans statement, we reference works that demonstrate substantial time lags and possible meltwater retention in the ablation zone as motivation for the study of near-surface porous ice and ablation zone hydrologic processes. We first note the evidence for, and assumption of efficient runoff delivery, then note evidence of time lags and possible retention. Finally, we suggest the weathering crust as a possible mechanism for runoff delays based on evidence from other supraglacial environments. We also draw a parallel with meltwater retention in snow and firn, as requested. Weathering crust formation, ablation, and density are then discussed in the second paragraph.

*Author changes in manuscript:* P1 L25 – P2 12, the opening paragraph has been rewritten, as requested.

Line 28: what is a "terminal moulin"? And not all runoff goes to moulins – there are supraglacial routes to proglacial regions, and lakes and crevasses. Suggest more circumspect and/clarified text here.

*Author response:* A "terminal moulin" is a moulin that exists at the terminal drainage point of a

supraglacial catchment. Analysis of sub-meter resolution WorldView-1/2 satellite imagery suggests that every supraglacial river in the study region drains to a moulin before reaching the ice edge (Smith et al., 2015). To avoid confusion, we have dropped the word "terminal".

*Author changes in manuscript:* P1 L26-28, we revised the statement and added reference to Colgan et al., (2011) to support drainage to crevasses, as requested.

Line 5: Cite Muller and Keeler (1969) for the introduction of the term "weathering crust". Might an additional diagram be helpful here to conceptually illustrate what you are focused on here for the less familiar reader?

*Author changes in manuscript:* P2 L5, Müller and Keeler, (1969) are cited following first mention of 'weathering crust'. Conceptual diagram (new Fig. 1) added, as requested.

Line 6: Fountain and Walder (1998) also note minimal delay to supraglacial runoff and so text and citation here, given phrasing, might be slightly inappropriate. Suggest check the source again.

*Author response:* Fountain and Walder (1998) is removed, as requested.

Line 9: What is a "seasonally temperate glacier"? Poor terminology, please revise. Seasonally temperate surface ice perhaps, but glacier thermal regime is a very different thing.

*Author response:* Thank you for this important correction. We were referring to the seasonally temperate near-surface, not the broad glacier thermal regime. Here and throughout the revised text, we replaced our various use of "temperate", "polythermal", etc. with "thermally transient ice surface" as in Irvine-Fynn et al., (2011). Elsewhere, we have removed mention of thermal regime altogether and replaced with "supraglacial environment" etc.

Line 11: Surely the depth is ice-type dependent, and stating "~2m" is not strictly correct. Consider rephrasing (see Cook et al., 2016). It was also surprising that at no point is Munro (1990, AAAR) cited here, a source confirming the subsurface melt and bulk ice density variations leading to uncertainty in runoff volumes at Peyto Glacier. Suggest consideration of this source, especially with regard L19.

*Author changes in manuscript:* P2 L21, "~2m" is replaced with "typically <2 m thick (Irvine-Fynn and Edwards, 2014; Müller and Keeler, 1969)".

Regarding Munro, (1990), thank you for suggesting this highly relevant reference. Though we cited Munro, (2011) in the original manuscript, this additional citation strengthens the literature review for the reader. We have added several references to this citation throughout the manuscript, as requested.

Line 17: Doesn't lateral meltwater motion result in sensible and frictional heat transfers, contributing to further removal of ice mass. Also suggest clarification over the vertical extension of the weathering crust, and how this influences mass for any given vertical position. The

process described by Muller and Keeler (1969) is a little more complex than perhaps is given credence here, and perhaps a more careful description could be afforded. See also Cook et al., 2016.

*Author response:* Regarding sensible and frictional heat transfers, we have added this to our interpretation of the weathering crust structure in Sect. 4, where we suggest meltwater advection along cracks may enhance subsurface weathering.

To address weathering crust vertical structure, we reference the characteristic depth-density profile from LaChapelle, (1959), and note the exponential attenuation of solar radiation in the upper few meters of ice (e.g. Brandt and Warren, 1993). We hope our new conceptual diagram (Fig. 1 in the revised) will further clarify the depth-variable nature of the crust for the reader.

*Author changes in manuscript:* P2 L14 – P3 L8, the introduction has been substantially revised to emphasize the depth-variable nature of the crust, including the new Fig. 1 conceptual diagram.

Line 22: The opening of this paragraph is not entirely appropriate, the structure and the content seems slightly superficial and/or repetitive (e.g. mention of delay in runoff is already in L6). Suggest revisiting this text through to L26 and P3.

*Author response:* Considering this and the next several comments, the revised introduction has the following structure:

1. Broad motivation – ablation zone hydrology is poorly represented in models, particularly with respect to seasonal runoff delays and retention
2. Definition / description of weathering crust formation and relevance to supraglacial hydrology and surface mass balance
3. Description of broader relevance with respect to microbial habitat and albedo
4. Justification and purpose for this study in Greenland

*Author changes in manuscript:* P1 L25 – P3 L23, the introduction is substantially revised and repetitive material is removed, as requested.

Line 22: Is subsurface melting in Antarctic contexts the same as the definition provided of weathering crusts on "temperate ice" (see L9)? Strongly advise some differentiation between subsurface melting and weathering crust terminology. This sentence could be removed at no loss to the paper.

*Author changes in manuscript:* Reference to Antarctic contexts is removed, as requested.

Line 25: Slightly unconvincing use of the literature here: some references focus on cryoconite holes, others on the weathering crust as a habitat. Recommend revisiting, with perhaps consideration of recent messages regarding glacier ecohydrology (e.g. Dubnick et al., 2017a,b, JGR and Hydro Proc.; Hotaling et al., 2017, Env Mic.; Milner et al., 2017, PNAS). Yes, the weathering crust is a substrate for cryoconite holes (see Muller and Keeler's 1969 diagram), but the focus here should be the hydrological aspects and for example disturbance to cryoconite holes that might influence their ecology (Edwards et al., 2011, ISME J; Mieczan et al., 2013, PPR) or distribution (e.g. Hodson et al., 2007, JGR). Then develop the 'undescribed in

Greenland to date' message and the guide of what is to follow (subsequent paragraph). If you do touch on the biogeochemical cycling aspects, it might be helpful to touch on these again in the discussion section.

*Author response:* In the revised introduction, we discuss weathering crust relevance to microbial habitat in a standalone paragraph. As suggested, we emphasize the relevance of weathering crust hydrology (Cook et al., 2016) as a control on glacier ecology via 1) cryoconite hole distribution (Hodson et al., 2007; Takeuchi et al., 2000), and 2) saturated interstitial void space (Irvine-Fynn and Edwards, 2014). We then develop the 'undescribed in Greenland to date' message in the subsequent paragraph. The biogeochemical message is revisited in the discussion, as suggested, where we provide slightly more detailed discussion and additional references.

*Author changes in manuscript:* P2 L31 – P3 L8, weathering crust relevance to microbial habitat is discussed in a standalone paragraph with reference to suggested messages regarding glacier ecohydrology. P14 L20-31, biogeochemical and surface albedo message is revisited in the discussion.

Line 1: Remove "In sum"

*Author changes in manuscript:* Removed, as requested.

Line 2-4: Consider revisiting (see L25 above), and bringing in energy balance and ablation (see again Munro, 1990, AAAR) and describing the reasons for weathering crust relevance. Then have a single paragraph giving the justification for the study in Greenland. I just found these two paragraphs jump around a little and felt that a more logical progression through material could be achieved.

*Author response:* Thank you again for these helpful suggestions. We hope the revised introduction provides a clear progression and justification for the present study.

*Author changes in manuscript:* P2 L29, Munro, (1990) is cited here and in several locations throughout the revised text.

Line 20: delete "mechanical" – not necessary. Be consistent with hyperlinks/formatting if used for www sources.

*Author changes in manuscript:* P4 L3, "mechanical" deleted, as requested. Hyperlink formatting corrected, as requested.

Line 21: "drilling" in glaciology typically implies more than shallow coring, might just talking about "coring" and "core sites" be sufficient? (e.g. P4 L1 "core sites" seems more appropriate).

*Author changes in manuscript:* P4 L3 "drilling" and "drilling sites" are changed to "coring" and "core sites", here and throughout the revised text, as requested.

Line 23: Issue of mass for calculation of density is relevant here. Are you measuring water and

ice? If so, are not the estimates of density in error. This issue needs to be addressed and accounted for; ice density has to be properly estimated given the depth variable water content.

*Author response:* The analysis assumes we are measuring dry ice density. Though water retention errors are inevitable, we are not aware of a specific estimate of depth-dependent water content for the very near-surface weathering crust we document in this study. As suggested, we cite estimates of temperate ice water content ranging from 0-9%, though most estimates (15 of 18) are <3.4%, including all estimates made from in situ calorimetric methods (Pettersson et al., 2004). The uppermost 9% estimate is well within our ±20% (revised to ±14%, see P7 L6-8) specific storage uncertainty estimate. We provide a thorough reporting of this error source in the revised methods.

*Author changes in manuscript:* P4 L25 – P5L3, revised discussion of error sources as noted above. P7 L6-8, revised calculation of density and porosity uncertainty propagation into specific storage uncertainty.

Line 29: Suggest "This uncertainty is incorporated into calculations of ice porosity and water content (see Sections 2.2 and 2.4)". In places, as here, writing clarity and conciseness could be tightened up.

*Author changes in manuscript:* P5 L2, revised as requested.

Line 8-9: Clarify the relevance of the centre of mass, if you are using the method to estimate the upper 14-30cm ice, just indicate that the upper 20cm is used, but the sampler geometry results in bias toward the uppermost ice and so leads to an underestimate of density. This just seems to be introducing terms which could be seen as confusing.

*Author changes in manuscript:* P4 L18, revised as requested.

Line 11: Issue of water content in core sections and uncertainties in density measurements remains problematic.

*Author response:* Thank you for drawing attention to this important source of uncertainty. We hope our clarified discussion of the two primary sources of expected error (water retention and loss of material) are sufficiently addressed in the revised methods.

*Author changes in manuscript:* P4 L25 – P5L3, revised discussion of error sources as noted above. P7 L6-8, revised calculation of density and porosity uncertainty propagation into specific storage uncertainty.

Line 12-15: This could be condensed: e.g. "for context, two 1.8m cores were extracted but ice density measurements were not undertaken, these cores are described further in Section 3.3".

*Author changes in manuscript:* P4 L22, revised as requested.

Line 15: Estimates of porosity will be affected by ice core sections that were weighed still

containing interstitial water. Removal of this water is not a trivial problem, as exposing the core to positive air temperatures will initiate further melt, and methods of forcing water out via centrifugal force is similarly challenging. An estimate of uncertainty is needed here, and this needs to be incorporated into the data derived from these potentially erroneous ice mass measurements.

*Author response:* We appreciate the point and do not take it lightly, but we do not think there is a reasonable way to construct a physically based error estimate beyond the provided ±10% (density and porosity) and ±14% (specific storage). As noted in response to previous requests, we highlight these sources of error in the revised methods.

Line 20: rationale for the change in ( ) to [ ]?

*Author response:* Thank you for noticing, [ ] has been changed to ( ) throughout.

Line 25: Might combining the equations here to A = B > C (as Eq 1) seem a neater and more consistent presentation of the equations? Would allow a slightly smoother explanation.

*Author changes in manuscript:* P5 L8, equation revised as requested.

Line 30: Does the time-frame and temperature of the water present any issue here? Given the thermal potential of supraglacial water (which I presume was used?), could you estimate and mass loss (or confirm this is negligible). The size of the weathering crust crystals might be important here – were the samples used representative of the upper 20cm for all sites?

*Author response:* Supraglacial water was used for the reason you mention. Water was immediately applied and, as noted, we carefully observed ice crystal structure and air bubbles and saw no evidence of melt. To some extent, the focus on the quality of the linear regression (noted below in a separate comment) was motivated by this concern. The linear relationship suggests 0% effective porosity for solid ice density (917 kg m$^{-3}$), which provides some confidence in the measurements, and the estimates made from the relationship. Regarding representativeness, we collected samples at every core site as well as 15 additional sites along the transect to increase the sample size (N=25). This has been noted in the revised methods.

*Author changes in manuscript:* P5 L15, we note 25 samples were collected in total, distributed along the transect. P5 L16, we note liquid water was sourced from nearby flowing rills and immediately applied.

Line 9: While it is good practice to cite, do we need more than one of two examples here? Just considering journal space.

*Author changes in manuscript:* P5 L27, citations reduced to two, as requested.

Line 19: "8m intervals"?

*Author response:* We changed to "8m posting", to distinguish them as physical locations, e.g.

"cryoconite holes were measured within 1 m radius of the posting …"

Line 20 & Page 6 Line 2: were cryoconite holes ubiquitous features, or did you measure those proximate to the sample point? Clarify here. If one hole was measured – is this representative of local water table – might measuring 4 holes in at each site have provided more robust estimates?

*Author response:* They were ubiquitous, but for expediency we measured the nearest cryoconite hole within a 1 m radius of the posting. We note this in the revised text. Regarding representativeness, we were not able to confirm this via direct measurements at each posting (again, owing to severe time limitations). Measuring 4 holes would certainly provide more robust estimates of local variability but we were more concerned with obtaining an adequate sample size to establish conditions along the transect. We have added the following statement where we report water levels: "The height of water in these holes likely varied diurnally and could have steadily drained or filled during the study period (Cook et al., 2016). The 15.5 cm average depth to water thus likely represents a snapshot of the transient water table height. Further, we measured a single hole at each 8 m posting and thus cannot quantify local variability."

*Author changes in manuscript:* P6 L11, we note the "nearest cryoconite hole within a 1 m radius of the posting …". P9 L15-27, we discuss the aforementioned issues of representativeness.

Line 21: The steel rod measurements are not entirely convincing, can you justify this a little more clearly. Furthermore, as above, a conceptual model might help here. Perhaps you need to consider the density decay curve (LaChappelle, 1959) and clarify your reasoning here, or use some alternative term in terms of a qualitative measure of "weathering crust mechanical resistivity" to the steel probe to indicate perhaps the shoulder on the density decay curve? There is also the issue of capillary draw in the weathering crust, are you able to confirm the water table in the crust is the same as that in the ice matrix? Does the water table truly exist as a broadly consistent level? If not, is this an uncertainty you can at least note if not estimate.

*Author response:* Thank you for this important critique. We agree the steel rod measurements were not presented clearly. We have removed the a priori characterization of the saturated/unsaturated transition where the depth probe measurements are described in the methods. In the revised manuscript, we use the depth to water below the ice surface in cryoconite holes as an estimate of the depth to saturation, whereas the depth probe measurements are used as a qualitative characterization of the weathering crust structure, drawing on the characteristic subsurface depth-density profile for weathering crust (LaChapelle, 1959), as suggested. Thank you for this suggestion, we think it will substantially improve the communication of our results to the reader.

In the text, we have clarified the following:

4) The unsaturated depth is inferred from the depth to water in cryoconite holes
5) The depth probe measurements are used as a qualitative description of weathering crust structure with reference to the sub-surface density profile and conceptual diagram (Fig. 1)
6) The apparent transition from very low density to higher density material is reported in the results

*Author changes in manuscript:* P6 Sect. 2.3, a priori characterization of saturated/unsaturated transition is removed, as requested.

P6 L13-14, we specify the depth to water in cryoconite holes is used as estimate of water table height (hence depth to saturation).

P6 L17-23, we clarify the depth probe is used as a qualitative check on the weathering crust structure with reference to new Fig. 1 conceptual diagram.

P9 L22-27, we expanded the discussion of water table height variability inferred from depth to water in cryoconite.

P10 L7-16, we report the depth probe measurements and clarify they provide qualitative inference about the depth of weakly bonded, disintegrated ice as per revised Fig. 1.

Line 5: Refrozen water while a component of storage in an overarching sense, is not the liquid storage, and is likely to be a proportion of the total available liquid water following a freezing event or water drainage to a cold front in the ice. If you are talking about liquid "water storage" then surely it is a negative value/term in that it is water lost to freezing? I'd also caution here given the inference is that the ice lenses are refrozen water – which may or may not be the case (see comment below regarding ice structure) and so a clearer definition of water storage might be helpful here.

*Author changes in manuscript:* P6 L26, refrozen term removed from equation, as requested.

Line 10: see L19, but are you exploring a total storage potential or just the saturated ice. Can you remove "saturated" here, and discuss both the observed water storage volume and the potential storage volume?

*Author response:* We are exploring the actual storage within saturated ice. Our use of the word "potential" was incorrect and misleading.

*Author changes in manuscript:* P7 L12 has been revised to read "Finally, for illustrative purposes we scale our $S_P$ estimate to the study catchment by multiplying the lower and upper values for $S_P$ estimated from the shallow ice cores by the bare ice surface area of the study catchment …".

Line 11: do you not just "extract" cores, rather than excavate them?

*Author changes in manuscript:* P7 L3, "excavated" is changed to "extracted", as requested.

Line 19: I think you need to better define the "potential total storage volume" (i.e. the entire weathering crust) vs. the estimated snapshot of water storage yielded by your observations – given the weathering crust storage potential will be time-variant given the nature of the weathering crusts formation. You could then discuss the total storage potential (under the conditions at the time of measurement), and the proportion of that which did indeed hold liquid

water (the stored volume), and compare those to melt production or runoff volumes.

*Author response:* We present the actual (instantaneous) specific storage estimated from the shallow cores, and then scale that to the study catchment to estimate a storage volume. Our use of the word 'potential' was incorrect and has been removed.

Line 24: "metre rule"?

*Author response:* The sentence has been removed altogether as it is redundant with the methods.

Line 32: Given the strong surface expression of structural features in the sector of the ice sheet studies here, you might give an indication that ice structure might underlie this (and perhaps cite papers that note the strong evidence of structural glaciology at the locality, and its theoretical background – for example Hambrey et al, 2000, Geol Soc; Hudleston, 2015, J Struc Geol.). You could at least provide a hypothesis here as a potential guide for future work. Given ice lenses are discussed, and given the ice sheet surface is ablating, these lens features must be emergent – and while it is possible refreezing of meltwater may contribute, do ice temperatures or meteorological conditions support this given the prevalence of these lens features? Were the lenses truly horizontal in formation or exhibit slight orientation?

*Author response:* We agree these lenses must be emergent and have substantially revised our discussion, including the suggested references. Regrettably we did not make careful observation of their orientation.

*Author changes in manuscript:* P8 L19-34, revised presentation of ice lens results with alternative hypothesis regarding ice structural controls, including the role of preferential flow paths and ice foliation.

P11 L20-23, further suggest ice structure may underlie the observed stratigraphy.

P12 L8-24, expanded discussion of ice lens features, their possible relationship with underlying ice structure, and expanded discussion of meltwater refreezing independent of the ice lenses.

Line 3: "The reported pM values therefore"? Missing word.

*Author changes in manuscript:* P8 L33, revised to read "the pM values reported in Fig. 4 …"

Line 4: Perhaps use "lens ice" to clarify your meaning here.

*Author changes in manuscript:* P9 L2, revised to read "weathered ice + lens ice".

Line 6: Surely if lenses are refrozen water their density will likely approach that of pure ice, and if structural features, their persistence would suggest higher glacier ice density values. As such, can you not include and quantify potential uncertainties here?

*Author response:* In keeping with our interpretation of ice lenses as structural features, we include the ice lens volume in our revised estimates of density, volume, and porosity. As pointed out by another reviewer, it was incorrect to omit the ice lens volume from the original estimate as the mass was incorporated into the density calculation. As such, we have removed the various references to this source of error, which should reduce confusion for the reader.

Line 12: You have two data points above the theoretical limit, so "consistently smaller" is not strictly correct. You also refer to Fig. 5 here three times in as many lines – consider using just one reference to the graphic.

*Author changes in manuscript:* P9 L4-5, "consistently" is replaced with "generally", and all but first reference to Fig. 5 is removed (now Fig. 6 in the revised).

Line 15-16: "unphysical"? Perhaps use "physically implausible".

*Author changes in manuscript:* P9 L7 "unphysical" is replaced with "physically implausible", as requested.

Line 15 & 18: There seems to be an overemphasis on the quality of the data here, please recall the equation used to derive the porosity value means there is circularity here – the porosity is a function of the two densities. Avoid overstating something here. You can simply report the observed relationship, the fact that how robust this is beyond the bounds of observations is equivocal, and that the relationship was used to estimate porosity.

*Author response:* We removed the overemphasis on the quality of the data here, following the suggested progression of reporting the relationship, the uncertainty beyond the range of observations, and the application of the relationship to the shallow core densities.

*Author changes in manuscript:* P9 L4-13, paragraph revised as requested.

Line 28: This section seems a little less flowing than others, and is characterized by short paragraphs. Can the core holes and the water levels noted in these be described further? The first paragraph and third surely belong together? But there is repetition here. Consider revisiting this section.

*Author response:* As requested, elements of the first and third paragraphs have been combined and the section has been condensed to four individual paragraphs with the following progression:

1) Description of ice surface topography, cryoconite hole depths, and depth to water along the transect
2) Description of refilling of drilled holes and water filled cryoconite holes as evidence of saturation
3) Description of the two-layer structure referencing the LaChapelle depth-density profile
4) Description of higher density material structure and possible lower bound on permeable ice from the two 1.8 m cores

*Author changes in manuscript:* P9 Sect 3.3 Paragraphs 1-4 rewritten, as requested.

Line 5: So do dry holes indicate the water table is more complex and not a level surface?

*Author response:* We do not think the water table is a level surface, but rather it seems to mirror the topography at the ~8m scale we sampled. This is supported by the lack of any trend in depth to water below the surface, relative to distance along the transect or relative to elevation.

Line 7: not sure you need to use caption detail in the Fig. reference here.

*Author changes in manuscript:* P9 L22, caption detail removed, as requested.

Line 11: I think you need to define where the ice is saturated – it isn't the full depth of the weathering crust, or is it? Just feel a little more clarity in needed here to ensure the observations and inferences are clearly described.

*Author response:* As requested, we state clearly that depth to saturation is inferred from the depth to water in cryoconite holes, and therefore the weathering crust is saturated from 15 cm down, on average.

*Author changes in manuscript:* P6 L13, we clarify in methods that depth to water in cryoconite holes is used to estimate depth to saturation i.e. the water table height. P7 L4, we clarify in methods that depth to water in cryoconite holes is used to estimate depth to saturation, and this depth is used to calculate storage from the borehole density/porosity. P9 L20-27, we clarify in the results that depth to water in cryoconite holes is used to estimate depth to saturation, and we note this is a snapshot estimate of what is likely a transient water table (Cook et al., 2016).

Line 21: You don't really have a handle on the "transient" nature of the weathering crust here – yes, you can conceptualise this as a two-layered feature. But although you show spatial variability, you have no detail on temporal change. I would focus on the message relating to the snapshot of water storage – and the volume that represents. And only in your discussion, mention the processes of weathering crust formation and how this would mean the depths of the porous and saturated ice would potentially vary.

*Author response:* As requested, 'transient' has been removed and the revised paragraph/section focuses on the snapshot of water storage and crust structure. We report the average depth of cryoconite holes, the average depth to water, the trend toward shallower holes with distance (-0.012 cm m$^{-1}$), and the lack of trend in depth to water. We reference work suggesting the water table is likely transient (Cook et al., 2016), acknowledging we provide a snapshot estimate.

Line 34: Further evidence for structural controls on the ice crystallography?

*Author response:* We agree, yes.

*Author changes in manuscript:* P8 L19-34, revised presentation of ice lens results with alternative hypothesis regarding ice structural controls, including the role of preferential flow paths and ice foliation. P11 L20-23, further suggest ice structure may underlie the observed stratigraphy. P12 L8-24, expanded discussion of ice lens features, their possible relationship with

underlying ice structure, and expanded discussion of meltwater refreezing independent of the ice lenses.

Line 4: Repetition of freezing leading to cessation of coring from method section, un-helpful here as a result section.

*Author changes in manuscript:* P10 L20-25, statement removed, as requested.

Line 6: It would be nice to see a little more result reporting here – not solely the reference to the table and the mean for all sites. Perhaps expand a little.

*Author response:* This section is expanded, as requested. We now report the mean density and porosity along with specific storage (referencing the table), and discuss the variability between cores. We then discuss antecedent meteorology to provide seasonal context. We report the maximum spring snow depth, date of snow disappearance, cumulative ice surface ablation following snow disappearance, and comparison with regional meteorology reported in Tedstone et. al., (2017).

*Author changes in manuscript:* P10 Sect 3.4 expanded, as requested.

Line 13: Just wondered if a clearer summary section leading to discussion might be helpful – in following with the results. For example, open with the lacking recognition of the weathering crust, and how here, observations of ice density revealed X Y and Z on a portion of the Greenland Ice Sheet, then move to how the two-layered structure matches previous work, and how density and water storage values compare to the other limited reports.

*Author response:* The opening paragraph has been combined with the second paragraph and reorganized as recommended i.e. 1) lack of recognition of weathering crust storage, 2) broad restatement of our findings, 3) two-layered structure consistent with previous work, and 4) density and water storage consistent with previous limited reports.

*Author changes in manuscript:* P11 L25, opening paragraph reorganized, as requested.

Line 14: Cite the Larson reports and Irvine-Fynn et al review that discuss near-surface surface storage here. Would citing the Jansson et al (2003, J Hydro) review also be useful here?

*Author changes in manuscript:* P11 L25-27, citations added, as requested.

Line 15: Why the specifics on polythermal ice sheets here? The references cited discuss temperate glaciers and a polythermal glacier, respectively.

*Author changes in manuscript:* P11 L25, reference to thermal regime removed, as requested.

Line 21: "stagnating"??

*Author changes in manuscript:* P11 L30 comparison to specific storage rates removed

altogether.

Line 1: Recall the reports you cite simply modelled water storage via water budgets – and so you can't compare core observations to hydrological models. Previous work hasn't examined ice cores to identify or report crystallographic changes. Please revisit.

*Author changes in manuscript:* P11 L30, references to specific storage rates have been removed and replaced with a general statement of consistency with previous findings.

Line 4: See earlier comments on ice structure.

*Author changes in manuscript:* P12 L8-24, we separated our discussion of ice lens features and their possible relationship with underlying ice structure from discussion of meltwater refreezing. New paragraph on meltwater refreezing follows, with reference to anecdotal observations of surface refreezing (see new Fig. 8, below).

Line 8: See earlier point about water budget equation, and the ice lenses being a negative 'storage' value, as indicated here. However, a stronger physical discussion of the potential formation processes for the ice lenses is needed – with comparison to ice structure and any alternative explanations too.

*Author response:* Thank you again for emphasizing the need to explain the lenses in greater detail.

Line 10: GrIS – either define and use as acronym throughout or use Greenland Ice Sheet as elsewhere.

*Author changes in manuscript:* GrIS has been changed to Greenland Ice Sheet, as requested.

Line 11: Condense to a single paragraph section perhaps?

*Author changes in manuscript:* P13 L2, the two paragraphs are condensed to a single paragraph, as requested.

Line 25: I'd suggest revisiting in view of the Munro (1990, AAAR) source.

*Author response:* We are not sure exactly what the reviewer is asking us to revisit in view of Munro (1990). We assume it is the finding that discrepancies between ablation and runoff were reconciled for that experiment by combining a detailed energy balance model with a specially designed ablatometer. If so, we agree the Munro experiment presents a useful demonstration this is possible, though we think it equally highlights the difficulty required to reconcile the effect of weathering crust on runoff and mass balance. We cite Munro (1990) throughout the revised text where relevant to emphasize it is possible to account for sub-surface melting (as well as van den Broeke et al., 2008).

*Author changes in manuscript:* P2 L29, we cite Munro with respect to accurately determining

mass change during periods of weathering crust development or removal. P13 L20-25, we emphasize sub-surface melting needs to be accounted for if surface elevation change is compared to modeled melt, citing Munro, (1990) and ven den Broeke, et al. (2008).

Line 24: Lutz reference focuses on ice algae, not cryoconite. Suggest Wientjes and Boggild references would be more appropriate here. Similarly, L25: Fountain discussed ice-lidded cryoconite in Antarctica which may physically be a little different – suggest a more cautious use of literature which refers to the types of feature and observations that are characteristic for Greenland (e.g. the older Gribbon, 1979, J Glac. or Gadja, 1958, Can Geogr. references for cryoconite holes in Greenland).

*Author changes in manuscript:* P14 L20, paragraph is broadened to discuss microbial communities generally, not just cryoconite holes. Thus, Lutz reference is retained, and several additional references are added. Fountain reference is replaced with Gribbon, (1979).

Line 12: Does Hoffman's study relate to a temperate or polythermal ice mass – isn't it cold? Or just remove the thermal regime aspect here – "supraglacial environments elsewhere. . ."

*Author changes in manuscript:* P15 L8, reference to thermal regime is removed and replaced with "supraglacial environments worldwide", as requested.

Line 15: You define the symbology, no need to repeat the definition here in L16, after its use on L15.

*Author changes in manuscript:* P15 L13, the symbology is removed, as requested.

Line 19: For impact, suggest you rephrase as "if these observations are representative of the Greenland Ice Sheet ablation zone, then wider implications are. . ., and future work should. . .."

*Author changes in manuscript:* P15 L15, the statement is rephrased, as requested

Fig.3.: Consistency with (..) or [..] on axes labels. The captions seem to be overly long – focus on the content and remove superfluous text. Label 1 -10 in the Fig.. Hatched areas are "no data" not "core depth" are they not? Can you not include the snow- shovel data here for the uppermost 20cm – albeit in a different colour, for comparison and completeness? Might inclusion of potential ablation here be helpful given from the field campaign description, the cores were collected over one week during which time ablation would take place – and such that (for example) a refreezing event (if this is what the lenses are) might be more clearly identified if lenses appear at the same depth relative to a zero set for the period of coring?

*Author response:* The cores were collected on 11-12 July. We state this clearly in the revised methods. Axis labels updated with consistent use of (). Cores 1-10 labeled in Fig. 3, changed 'core depth' to 'no data', and gap filled the upper 20 cm with the snow cutter density measurements, as requested. See revised figures below.

Fig.4.: Is the lower image for the core in the upper? Perhaps use arrows to indicate where ice lenses are on the core.

*Author response:* The photos were taken from different cores but are the best photos we have. Core locations noted in revised caption and arrows added to indicate ice lenses.

Fig.5.: y-axis should be phi-eff. The equation given should be phi-hat-eff (inconsistent symbology). Surely "observations" not "data"? Caption – is "measured data" needed here?

*Author response:* The equation is corrected to read phi-hat-eff, 'data' is changed to 'observations', and 'measured data' is removed from the caption, as requested. Regarding the y-axis label, since the axis is used for phi-hat-eff, measured phi-eff, and phi-total, we think it is better to leave the label as generic phi [-] and let the legend distinguish.

Fig.6.: (b) there is a lot of information here, and I just wonder if two panels here would be helpful – one to give clearer indication of the water level in holes with a simple zero as ice surface, and then the detrended plot with the unsaturated crust estimate? The two grey tones are hard to differentiate. If detrended, surely the data should be scattered around zero – so did you offset this to a maximum positive deviation - one presumes so, but clarification would be appropriate? Have you compared distance or elevation against any of the variables – are there any other patterns to explore – as these don't seem to have been mentioned in the main text – even if to confirm there is no elevation dependency.

*Author response:* The grey shaded area is removed to improve clarity, as requested. Empty space is added at either end to improve clarity, and shallow cores are labeled at their respective depth and location, as per a request from another reviewer.

Regarding the offset, yes, they were all offset to the maximum positive deviation such that the datum is 0. This was done for consistency with the new Fig. 1 conceptual diagram, consistent with Muller and Keeler (1969) and Irvine-Fynn and Edwards (2014).

Regarding trends, as noted above we find a slight trend toward shallower holes ($-0.012$ cm m$^{-1}$) but no significant trend (or trend whatsoever) in depth to water below the surface. Trends for elevation are same, but elevation values are not normally distributed and co-vary strongly with distance, so we report the distance trend.

Table 1: could you include a column of mean phi-eff for each core here, for ease of direct comparison?

*Author response:* The mean phi-eff is added as well as mean density

**Revised figures/tables:**

[revised manuscript text omitted]

---

## Referee Report (RR1)

Overall, there is a neat and generally well written paper here that reports on the

The authors have made a substantial effort to respond positively to the three Reviewers comments. Several of the major comments raised by the reviewers were common to those reviews. A read through the authors response document (while evidencing some repetition) highlights they have considered all points raised, and made positive modifications in the majority of the cases.

In consideration of the variety and wealth of revisions the authors have fully and effectively detailed, it seems prudent to simply examine this paper very much as a new submission, and consequently, the review that follows adopts this strategy.

There are a few more significant points to note here that are worthy of highlighting:

1. The authors still do not seem to fully describe the seasonal (and synoptic) dynamics of the weathering crust. This was a point raised by several of the reviewers previously, and now in later sections the authors refer to the "transient" nature of the weathering crust (hereafter, crust) – however, they do not seem to give the reader a clear indication early on as to the causes behind the transience. See next comment.

2. The authors, in response to commentary in previous reviews regarding the supraglacial conditions over the summer season, have included wording regarding "supplementing … measurements with … meteorological… [records]" (see Section 2, P3). However, it seems these data are not presented, nor reported or discussed in any meaningful way. This seems to be a missed opportunity as, for example, they would be able to discuss the refreezing potential or synoptic variability immediately around their period of observations. One can't help but feel that by placing the ablation season meteorology (short wave radiation, air temperature, rainfall, ablation) in a figure, indicating the end of snow cover and timing of their observations -  the authors could make a far more convincing argument relating to the refreezing, the transience of the crust and so on – points they are keen to expand upon in their results and discussion. It seems a very simple and effective way to really make the paper excellent with clear support for the arguments posited. And would demand relatively little material change to the main text, which is generally sound but could be bolstered by including those data to support arguments.

3. The general methods and calculations sections which were comment upon in the review process are now much more clear and robust. Similarly, the description and initial discussion of the core results are appropriate.

4.  The discussion could potentially benefit from some further minor tweaking – this section could be condensed to make a more punchy and impactful paper here.

For far more minor observational comments, some of which relate to the points above, see the listing below.

P1.

Title: Ice Sheet should be capitalised.

L11: Contradiction with L14, either the ice is saturated or it is partially saturated at depth, and unsaturated near the surface. Suggest clarification or removal of saturated in L11, and move to L13.

L29: suggest "for Greenland"

P2.

L2 and 4: Perhaps clarify that the former is for the ice sheet, while L4 refers to valley glaciers elsewhere.

L5: "inferred to relate to the presence" – none of the studies have definitively demonstrated the role of the crust, and so suggest change "owing".

L7: Yes, the crust is a substrate for the development and deepening of cryoconite holes, suggest reconsider use of "substrate" here – "locus"?

L8: Suggest splitting sentence relating to firm as its own. Refreezing is not transport, so there is disjunction here.

L14: suggest reference to Figure 1 here, or at least in this paragraph.

L15: The seasonal or synoptic development should be noted here. The crust forms in the melt season when glacier ice is exposed. And note that the as synoptic conditions vary over the ablation season, the depth and degree of evolution of the crust may vary.

L21: Not all the crust is saturated, is it?

L22: Isn't Irvine-Fynn et al (2011) a better citation here that describes the crust formation?

P3.

L1 or 7: For relevance, the Hotaling et al 2017 would be a sensible inclusion here, perhaps as an additional note as to the relevance the crust may have for glacier surface ecosystems. (Hotaling, S., Hood, E. and Hamilton, T.L., 2017. Microbial ecology of mountain glacier ecosystems: Biodiversity, ecological connections, and implications of a warming climate. Environmental Microbiology.)

L9: these? Unclear. For new paragraph. Surely better to phrase: "Despite the hydrological and ecological implications…"

L16: Delete "hydrologic system… balance" as repetitive of previous sentence.

L24-25: This seems to read awkwardly, consider condensing a little.

L28: You do not present any meteorological data here. This then becomes rather irrelevant as the reader can not access the rates of ablation, the radiation records or ascertain if night-time refreezing may occur.

P4.

L2: Not convinced "posting" is the correct word, and considering an international readership – a more standard word would be more prudent. Check dictionary – as "posting" typically refers to a singular location. Here it seems "interval(s)" is the more appropriate and accessible word to use. Please consider revising throughout. This was noted by previous reviewers.

L7: What is the manufacturers accuracy for the Acculab scale, perhaps cite this here?

P5:

L3: clarify "specific water storage".

P6.

L2: Suggest checking TC formatting for quotes or subjective terms – if using "…" for a quote, perhaps consider '…' for a more subjective term (as 'true' here). Not sure of formatting expectations, but may be worth considering or removing here.

L10: consider replacing "drill bit" with auger where appropriate, to better clarify between corer and auger.

L14: should i.e. be in brackets? Or preceded with comma.

L15: unclear – suggest revisting for clarity, issues with "posting" again.

P8.

L6: suggest rewording to: "to unweathered glacier ice" and strongly suggest citing Muller and Keeler (1969) and Irvine-Fynn et al., (2011) here too. Most of the examples show density only over a few decimetres at most, with glacier ice density typically being reached within this depth, whereas the deeper scenarios relate to stagnant ice where there is no replenishment (see Fountain and Walder, 1998).

L9: Tighten language here as not quite clear, and might using "respectively" be appropriate, rather that "vs.".

L10: Drilling, coring – unclear. Perhaps seek to clarify or note this is both for cores and augers.

L23: A reference that reviews ice fabric or structure here might be sensible. Support for the notion would be appropriate.

L24 and L27: should km and cm both be written in full here?

L31: Could be the optical properties of the crystals – if crystals are large and clear and devoid of air bubbles or other inclusions, then the internal melting may be reduced and shortwave radiation simply passes to the next more optically opaque layer. Suggest a minor clarification here, if appropriate. Or later.

P9.

L18: again postings is unclear – "locations" and reduce confusion with the systematic intervals used for the profile vs. the other sampling locations.

L19: These 'dry' observations were not all on one day while 'wet' were other days? Might be worth just noting that, if the case.

P10.

L7: "ice comprising two layers…" perhaps

P11.

L5: You don't really describe the transience in the introduction, and so perhaps need to note this there, so that this does not come in unexpected or unjustified.

L9: Comma not needed after Tedstone here. Check references throughout.

L24: Perhaps consider opening the discussion with a brief summary of what the core "take home" results you've presented are, then move on to the comparisons. Again, just short text addition here

which may be beneficial. Also suggest initial subheading for the overview section (crust form/structure?), that then leads to the melt storage volume and other implications.

L29: Repetitive with "storage" – consider using "transient reservoir in Greenland's bare ice ablation zone".

P12.

L6: Suggest include Munro (2011) here and remove Fountain ad Walder (1998).

L8: Previous work hasn't reported ice structure so "unlike" is not really correct. Consider noting that this study has also reported ice structure, which hasn't been reported in previous studies of the weathering crust. You used this assertion elsewhere, and it might be prudent to downplay this criticism.

L11-14: explain a little more, see earlier comment regarding optical properties of crystals and density of interfaces.

L15: centimetre-scale in full

L16: do you mean melt not weathering?

L15-20: Here, and in previous sentence(s), the use of and position references seems unusual. Consider revisiting and tidying up and condensing a little.

L17: delete "terrestrial lithology" phrase, it is unhelpful.

L26-28: Temperature conditions are very difference in Antarctic setting, are they not. And the complexity here is not well presented. The refreezing process releases latent heat, which may lead to melt that off-sets the 'sink' described. Consider being more physically accurate and detailed here.

P13.

L7: Think this could be more clearly written here. E.g. Our study catchment is equivalent to 2% of the 2800 km^3 contributing area draining to the Waston River. But not sure what this statement adds. You've highlighted the water volume, so why does the contributing area matter? Surely you can state that the water volume is equivalent to 1hr of peak Q in 2012, or is the same as Nhrs of Q (see Overeem's Q data series perhaps).

L9: Consider "density of porosity more widely over the GIS ablation zone" – vast is subjective.

L11: "en-, sub- and proglacial systems"

L12: While relevant and appropriate, this might be viewed as moving slightly away from the data itself and tending toward conjectural discussion, and could be condensed while maintaining the effectiveness. You are speaking beyond the data here, and it would perhaps be more effective to keep this to a shorter section which is more concise and focused. Fundamentally, you are simply saying that by finding a crust on GrIS, you now know that ablation measurements that fail to include subsurface melt may be problematic to use for validation, that water storage may vary over time due to the dynamic behaviour of the crust, that assuming rapid runoff may not be appropriate, and the hydraulics of the crust may have impacts on impurity and ecological mobility and residence time (ie. albedo and biogeochemistry). Can you not simplify the section?

L15: Neither LaChapelle or Muller & Keeler are ice sheet studies. Remove "sheet". And surely Munro (1990) here too.

L24: not sure dubious is the correct wording here. "less reliable" or "less robust"?

L32: But other models have included supraglacial ponds etc. so this seems to be a little misrepresentative.

P14.

L5: Doyle's work is for late summer when the crust is likely degraded, so perhaps not as appropriate as hoped. Could be removed entirely.

L20: Not sure entrainment is the correct word. Perhaps "retention".

P15:

L26: UAS – only use here, know this is drone imaging, but be clear throughout.

Tables and figures:

Figure 1b – the density curve given seems slightly at odds with the cores and previous estiamtes (LaChapelle 1959; Muller & Keeler, 1969). Suggest slight reduction in the exaggeration of the low density surface portion of the plot.

Figure 2 – consider a-d to clarify insets, and in caption. Could present lower panels at top, and site below (ie. reverse the figure). Suggest scale given either on image or in text for insets b and c.

Figure 8 – indicative scales would be beneficial. Explain the various images fully please.

Table 1 – define mu symbol

---

## Author Response (AR2)

Dear Dr. Wouters,

We are pleased to submit for your consideration a minor revision of our manuscript "Meltwater storage in low-density near-surface bare ice in the Greenland Ice Sheet ablation zone". We have addressed all comments made by the reviewers. These include a new figure showing the meteorological data collected by the KAN-M PROMICE/GAP automatic weather station and a revised description of these data in Section 3.4. To further address reviewer concerns regarding the temporal variability of the weathering crust, we separated the second paragraph of the introduction into two paragraphs. The first paragraph describes the physical process of weathering whereas the second paragraph describes controls on the temporal variability. The scope and conclusions of our paper has not changed but we hope these revisions will improve the clarity of our message for the readership.

A detailed, point-by-point description of all changes to the manuscript appears below. As well, a revised manuscript with all changes highlighted is included below the point-by-point description.

Respectfully submitted,

Matthew G. Cooper (lead author)

UCLA Department of Geography

**Reviewer comments**

1. The authors still do not seem to fully describe the seasonal (and synoptic) dynamics of the weathering crust. This was a point raised by several of the reviewers previously, and now in later sections the authors refer to the "transient" nature of the weathering crust (hereafter, crust) – however, they do not seem to give the reader a clear indication early on as to the causes behind the transience. See next comment.

**Author reply:** As requested, we describe controls on the seasonal dynamics of the weathering crust in the revised introduction. We have now separated the second paragraph of the introduction into two paragraphs. The first paragraph describes the physical process of weathering whereas the second paragraph describes controls on the temporal variability.

**Changes to the manuscript:** P2 L26 revised paragraph describing seasonal controls on weathering crust growth and decay.

2. The authors, in response to commentary in previous reviews regarding the supraglacial conditions over the summer season, have included wording regarding "supplementing ... measurements with ... meteorological... [records]" (see Section 2, P3). However, it seems these data are not presented, nor reported or discussed in any meaningful way. This seems to be a missed opportunity as, for example, they would be able to discuss the refreezing potential or synoptic variability immediately around their period of observations. One can't help but feel that by placing the ablation season meteorology (short wave radiation, air temperature, rainfall, ablation) in a figure, indicating the end of snow cover and timing of their observations - the authors could make a far more convincing argument relating to the refreezing, the transience of the crust and so on – points they are keen to expand upon in their results and discussion. It seems a very simple and effective way to really make the paper excellent with clear support for the arguments posited. And would demand relatively little material change to the main text, which is generally sound but could be bolstered by including those data to support arguments.

**Author reply:** As requested, we now include a figure (new Figure 8) that shows the meteorological records collected at the PROMICE/GAP KAN-M automatic weather station (AWS) during summer 2016. These data are discussed in Section 3.4 P12 L4. As requested, we show the daily and cumulative ice surface ablation, net shortwave radiation, and air temperature.

We contacted the providers of these data to confirm our interpretation of them. Following clarification of the methods used to measure snow ablation at KAN-M, we have revised our reporting of these data in two respects. First, in the prior submission we reported a snow disappearance date of ~21 June. This was determined by examining the sonic ranging data. The data providers helped us confirm the winter snowpack likely melted out on or around ~8 June, and there were two ephemeral snowfall events on ~16 June and ~25 June. These were determined by examining the albedo records along with the sonic ranging data. This albedo data is also included in the revised Figure 8. The ~25 June snowfall event was the 'snow disappearance date' we previously reported. The difference in the ~21 June date we reported and the ~25 June ephemeral snowfall event is likely due to small scale variability in snow cover and melt rates in the immediate vicinity of the weather station. The sonic ranger measures a different footprint than the albedo sensor.

The second change we made relates directly to the first. In the prior submission, we reported the cumulative ice surface ablation prior to collection of the shallow ice cores was ~55 cm. In the revised, we report a cumulative ice surface ablation of ~74 cm. The difference is due to our assumption that ablation prior to 21 June was snow ablation and needed to be subtracted from total ablation to determine solid ice surface ablation. The data providers confirmed that snow ablation does not need to be subtracted from the ice surface ablation.

These differences do not change our conclusions, but they do change the statistics we report in Section 3.4. Specifically, the cumulative ice surface ablation is now larger, and therefore the inferred 14–18 cm of specific meltwater storage is ~21–27% of the cumulative seasonal ice surface ablation instead of the 28–36% reported previously (P12 L16).

**Changes to the manuscript:** P12 Section 3.4; Figure 8.

3. The general methods and calculations sections which were comment upon in the review process are now much more clear and robust. Similarly, the description and initial discussion of the core results are appropriate.

4. The discussion could potentially benefit from some further minor tweaking – this section could be condensed to make a more punchy and impactful paper here.

**Author reply:** Thank you for these helpful suggestions. The discussion has been revised as requested in the comments below.

For far more minor observational comments, some of which relate to the points above, see the listing below.

P1. Title: Ice Sheet should be capitalised.

**Changes to the manuscript:** P1 L2, capitalized as requested.

L11: Contradiction with L14, either the ice is saturated or it is partially saturated at depth, and unsaturated near the surface. Suggest clarification or removal of saturated in L11, and move to L13.

**Changes to the manuscript:** P1 L13, saturated moved to L13, as requested.

L29: suggest "for Greenland" P2.

**Changes to the manuscript:** P2 L1, "for Greenland", as requested.

L2 and 4: Perhaps clarify that the former is for the ice sheet, while L4 refers to valley glaciers elsewhere.

**Changes to the manuscript:** P2 L4 and L6, clarified as requested.

L5: "inferred to relate to the presence" – none of the studies have definitively demonstrated the role of the crust, and so suggest change "owing".

**Changes to the manuscript:** P2 L6, "inferred to relate", as requested.

L7: Yes, the crust is a substrate for the development and deepening of cryoconite holes, suggest reconsider use of "substrate" here – "locus"?

**Changes to the manuscript:** P2 L9, 'locus', as requested.

L8: Suggest splitting sentence relating to firm as its own. Refreezing is not transport, so there is disjunction here.

**Changes to the manuscript:** P2 L10, split into two sentences, as requested.

L14: suggest reference to Figure 1 here, or at least in this paragraph.

**Changes to the manuscript:** P2 L24, reference to Fig. 1, as requested.

L15: The seasonal or synoptic development should be noted here. The crust forms in the melt season when glacier ice is exposed. And note that the as synoptic conditions vary over the ablation season, the depth and degree of evolution of the crust may vary.

**Changes to the manuscript:** P2 L26, seasonal development described in new paragraph, as requested.

L21: Not all the crust is saturated, is it? L22: Isn't Irvine-Fynn et al (2011) a better citation here that describes the crust formation?

**Changes to the manuscript:** P2 L24, 'saturated' deleted and Irvine-Fynn et al (2011) included.

L1 or 7: For relevance, the Hotaling et al 2017 would be a sensible inclusion here, perhaps as an additional note as to the relevance the crust may have for glacier surface ecosystems. (Hotaling, S., Hood, E. and Hamilton, T.L., 2017. Microbial ecology of mountain glacier ecosystems: Biodiversity, ecological connections, and implications of a warming climate. Environmental Microbiology.)

**Changes to the manuscript:** P3 L15, Hotaling et al. (2017) cited, as requested.

L9: these? Unclear. For new paragraph. Surely better to phrase: "Despite the hydrological and ecological implications..."

**Changes to the manuscript:** P3 L23, revised as requested.

L16: Delete "hydrologic system... balance" as repetitive of previous sentence. L24-25: This seems to read awkwardly, consider condensing a little.

**Changes to the manuscript:** P3 L31, 'hydrologic system … balance' deleted, as requested.

L28: You do not present any meteorological data here. This then becomes rather irrelevant as the reader cannot access the rates of ablation, the radiation records or ascertain if night-time refreezing may occur.

**Changes to the manuscript:** Section 3.4, meteorological data presented in new Figure 8 and discussed in Section 3.4, as requested.

P4.

L2: Not convinced "posting" is the correct word and considering an international readership – a more standard word would be more prudent. Check dictionary – as "posting" typically refers to a singular location. Here it seems "interval(s)" is the more appropriate and accessible word to use. Please consider revising throughout. This was noted by previous reviewers.

**Changes to the manuscript:** 'posting' changed to 'interval' throughout the text, as requested.

L7: What is the manufacturers accuracy for the Acculab scale, perhaps cite this here? P5: L3: clarify "specific water storage".

**Changes to the manuscript:** P5 L21, 'specific water storage', as requested. Regarding the scale, this model is no longer in production and the specifications are in storage in Greenland. It is a scientific grade scale.

L2: Suggest checking TC formatting for quotes or subjective terms – if using "..." for a quote, perhaps consider '...' for a more subjective term (as 'true' here). Not sure of formatting expectations, but may be worth considering or removing here.

**Changes to the manuscript:** Thank you.

L10: consider replacing "drill bit" with auger where appropriate, to better clarify between corer and auger.

**Changes to the manuscript:** P6 L28, 'drill bit' replaced with auger here and throughout.

L14: should i.e. be in brackets? Or preceded with comma. L15: unclear – suggest revisiting for clarity, issues with "posting" again. P8.

**Changes to the manuscript:** P7 L2, brackets used as requested.

L6: suggest rewording to: "to unweathered glacier ice" and strongly suggest citing Muller and Keeler (1969) and Irvine-Fynn et al., (2011) here too. Most of the examples show density only over a few decimetres at most, with glacier ice density typically being reached within this depth, whereas the deeper scenarios relate to stagnant ice where there is no replenishment (see Fountain and Walder, 1998).

**Changes to the manuscript:** P8 L25, reworded to 'unweathered glacier ice', and citations updated, as requested.

L9: Tighten language here as not quite clear, and might using "respectively" be appropriate, rather that "vs.".

**Changes to the manuscript:** P8 L27, 'respectively', as requested.

L10: Drilling, coring – unclear. Perhaps seek to clarify or note this is both for cores and augers.

**Changes to the manuscript:** P8 L31, 'coring', as requested.

L23: A reference that reviews ice fabric or structure here might be sensible. Support for the notion would be appropriate.

**Changes to the manuscript:** P9 L14, 'Hudleston (2015)', as requested.

L24 and L27: should km and cm both be written in full here?

**Changes to the manuscript:** P9 L15 and L19, as requested.

L31: Could be the optical properties of the crystals – if crystals are large and clear and devoid of air bubbles or other inclusions, then the internal melting may be reduced and shortwave radiation simply passes to the next more optically opaque layer. Suggest a minor clarification here, if appropriate. Or later.

**Changes to the manuscript:** P14 L1, optical properties discussed, as requested.

P9.

L18: again postings is unclear – "locations" and reduce confusion with the systematic intervals used for the profile vs. the other sampling locations.

**Changes to the manuscript:** 'postings' updated to 'locations' with respect to specific locations and 'intervals' with respect to the measurement strategy, as requested.

L19: These 'dry' observations were not all on one day while 'wet' were other days? Might be worth just noting that, if the case.

**Changes to the manuscript:** P10 L14, 'at the time of observation' and P10 L19 'As such …', as requested.

P10.

L7: "ice comprising two layers..." perhaps

**Changes to the manuscript:** No changes are made.

P11.

L5: You don't really describe the transience in the introduction, and so perhaps need to note this there, so that this does not come in unexpected or unjustified.

**Changes to the manuscript:** As noted, seasonal controls are presented in the revised introduction.

L9: Comma not needed after Tedstone here. Check references throughout.

**Changes to the manuscript:** P12 L9, comma removed, as requested.

L24: Perhaps consider opening the discussion with a brief summary of what the core "take home" results you've presented are, then move on to the comparisons. Again, just short text addition here which may be beneficial. Also suggest initial subheading for the overview section (crust form/structure?), that then leads to the melt storage volume and other implications.

**Changes to the manuscript:** P13 L1, brief summary of take home results is included, and a subheading is added

L29: Repetitive with "storage" – consider using "transient reservoir in Greenland's bare ice ablation zone".

**Changes to the manuscript:** P13 L15, revised as requested.

P12.  L6: Suggest include Munro (2011) here and remove Fountain ad Walder (1998).

**Changes to the manuscript:** P13 L24, Munro (2011) included and Fountain and Walder (1998) removed, as requested.

L8: Previous work hasn't reported ice structure so "unlike" is not really correct. Consider noting that this study has also reported ice structure, which hasn't been reported in previous studies of the weathering crust. You used this assertion elsewhere, and it might be prudent to downplay this criticism.

**Changes to the manuscript:** P13 L26, 'In addition to meltwater storage …', as requested. Also P8 L9.

L11-14: explain a little more, see earlier comment regarding optical properties of crystals and density of interfaces.

**Changes to the manuscript:** P14 L1, optical properties discussed, as requested.

L15: centimetre-scale in full  L16: do you mean melt not weathering?

**Changes to the manuscript:** P14 L5, 'melting', as requested.

L15-20: Here, and in previous sentence(s), the use of and position references seems unusual. Consider revisiting and tidying up and condensing a little.

**Changes to the manuscript:** Efforts were made to tidy up here, as requested. In other places we retain citations mid-sentence where specific statements require support.

L17: delete "terrestrial lithology" phrase, it is unhelpful.

**Changes to the manuscript:** removed, as requested.

L26-28: Temperature conditions are very difference in Antarctic setting, are they not. And the complexity here is not well presented. The refreezing process releases latent heat, which may lead to melt that off-sets the 'sink' described. Consider being more physically accurate and detailed here.

**Changes to the manuscript:** P14 L14, 'Although the climatic context is different …', as requested.

P13.

L7: Think this could be more clearly written here. E.g. Our study catchment is equivalent to 2% of the 2800 km^3 contributing area draining to the Waston River. But not sure what this statement adds. You've highlighted the water volume, so why does the contributing area matter? Surely you can state that the water volume is equivalent to 1hr of peak Q in 2012, or is the same as Nhrs of Q (see Overeem's Q data series perhaps).

**Changes to the manuscript:** P14 L31, revised as requested. Contributing area demonstrates that small portions of weathering crust can harbor surprisingly large volumes of meltwater.

L9: Consider "density of porosity more widely over the GIS ablation zone" – vast is subjective.

**Changes to the manuscript:** P15 L1, revised as requested.

L11: "en-, sub- and proglacial systems"

**Changes to the manuscript:** P15 L3, as requested.

L12: While relevant and appropriate, this might be viewed as moving slightly away from the data itself and tending toward conjectural discussion, and could be condensed while maintaining the effectiveness. You are speaking beyond the data here, and it would perhaps be more effective to keep this to a shorter section which is more concise and focused. Fundamentally, you are simply saying that by finding a crust on GrIS, you now know that ablation measurements that fail to include subsurface melt may be problematic to use for validation, that water storage may vary over time due to the dynamic behaviour of the crust, that assuming rapid runoff may not be appropriate, and the hydraulics of the crust may have impacts on impurity and ecological mobility and residence time (ie. albedo and biogeochemistry). Can you not simplify the section?

**Changes to the manuscript:** P15 L6, this section has been condensed from four to three paragraphs. The ideas relating to channel vs porous hydraulics are fundamentally similar to the ideas relating to SMB model meltwater runoff delays. The section now follows the following structure 1) importance for ice surface elevation changes, 2) importance for meltwater routing and runoff delays, and 3) importance for ecology and albedo.

L15: Neither LaChapelle or Muller & Keeler are ice sheet studies. Remove "sheet". And surely Munro (1990) here too.

**Changes to the manuscript:** P15 L9, 'sheet' removed, as requested. Citations updated, as requested.

L24: not sure dubious is the correct wording here. "less reliable" or "less robust"?

**Changes to the manuscript:** P15 L16, 'dubious' replaced with 'not reliable', as requested.

L32: But other models have included supraglacial ponds etc. so this seems to be a little misrepresentative.

**Changes to the manuscript:** Sentence removed.

P14.

L5: Doyle's work is for late summer when the crust is likely degraded, so perhaps not as appropriate as hoped. Could be removed entirely.

**Changes to the manuscript:** Removed, as requested.

L20: Not sure entrainment is the correct word. Perhaps "retention".

**Changes to the manuscript:** P15 L31, 'retention', as requested.

P15: L26: UAS – only use here, know this is drone imaging, but be clear throughout.

**Changes to the manuscript:** P17 L11, 'Unmanned Aerial System', as requested.

Tables and figures:

Figure 1b – the density curve given seems slightly at odds with the cores and previous estimates (LaChapelle 1959; Muller & Keeler, 1969). Suggest slight reduction in the exaggeration of the low density surface portion of the plot.

**Changes to the manuscript:** No changes made.

Figure 2 – consider a-d to clarify insets, and in caption. Could present lower panels at top, and site below (ie. reverse the figure). Suggest scale given either on image or in text for insets b and c.

**Changes to the manuscript:** 'Postings' changed to 'intervals', as requested. No other changes made.

Figure 8 – indicative scales would be beneficial. Explain the various images fully please. Table 1 – define mu symbol

**Changes to the manuscript:** P30, 'mu' changed to 'average', as requested.

P32 L5, Descriptions of photographs provided, as requested.

[revised manuscript text omitted]